# Mini-batch optimization enables training of ODE models on large-scale datasets

Paul Stapor [1,2], Leonard Schmiester [1,2], Christoph Wierling[3], Simon Merkt[4], Dilan Pathirana[4],
Bodo M. H. Lange[3], Daniel Weindl [1] & Jan Hasenauer [1,2,4 ✉]

Quantitative dynamic models are widely used to study cellular signal processing. A critical step in modelling is the estimation of unknown model parameters from experimental data. As model sizes and datasets are steadily growing, established parameter optimization approaches for mechanistic models become computationally extremely challenging. Mini-batch optimization methods, as employed in deep learning, have better scaling properties. In this work, we adapt, apply, and benchmark mini-batch optimization for ordinary differential equation (ODE) models, thereby establishing a direct link between dynamic modelling and machine learning. On our main application example, a large-scale model of cancer signaling, we benchmark mini-batch optimization against established methods, achieving better optimization results and reducing computation by more than an order of magnitude. We expect that our work will serve as a first step towards mini-batch optimization tailored to ODE models and enable modelling of even larger and more complex systems than what is currently possible.

[1] Helmholtz Zentrum München - German Research Center for Environmental Health, Institute of Computational Biology, 85764 Neuherberg, Germany. [2] Technische Universität München, Center for Mathematics, Chair of Mathematical Modeling of Biological Systems, 85748 Garching, Germany. [3] Alacris Theranostics GmbH, 12489 Berlin, Germany. [4] Universität Bonn, Faculty of Mathematics and Natural Sciences, 53115 Bonn, Germany. ✉email: jan.hasenauer@uni-bonn.de

Cellular signal processing controls key properties of diverse mechanisms such as cell division[1], growth[2], differentiation[3], or apoptosis[4]. Understanding its highly dynamic and complex nature is one of the major goals of systems biology[5]. A common approach is modelling signaling pathways using ordinary differential equations (ODEs)[6–10]. To account for the complex cross-talk between different pathways, recent models are growing increasingly large, reaching the boundaries of what is currently computationally feasible[10–14].

Most ODE models contain unknown parameters, e.g., reaction rate constants, which model training algorithms infer from measurement data such as immunoblotting[15], proteomics[12], quantitative PCR[16], or cell viability[17]. Larger models require more data to ensure the reliability of parameter estimates and model predictions[18]. For models of cancer signaling, public databases[19–22] can be exploited. However, data from different perturbation experiments correspond to different initial value problems[23], which we refer to as "experimental conditions". Each experimental condition is a different vector of input parameters to the ODE system, which requires independent simulation at each iteration during model training. Hence, the computation time scales linearly with the number of experimental conditions. For large-scale ODE models with several hundred chemical species and thousands of experimental conditions, this can take tens of thousands of computing hours, even with state-of-the-art methods, such as adjoint sensitivity analysis and hierarchical optimization[10,14].

For the training of ODE models, gradient-based approaches such as multi-start local optimization[23] or hybrid scatter search[24] are the best performing methods to date[23,24]. In multi-start local optimization, local optimization initializes at many random starting points in order to globally explore the parameter space. For small- to medium-scale models, these methods unravel the structure of local optima and recover the same global optimum reproducibly[23,25]. However, for large-scale models, where each local optimization is computationally expensive, only a small number of starts are feasible[10,12,14]. This is one of the main reasons why satisfactory parameter optimization for large-scale ODE models is still an open problem[26], and why research on parameter estimation is of major importance[27].

In the field of deep learning, where gradient-based local optimization methods are also in use[28–31], model training often involves large datasets, requiring many independent model evaluations[32,33]. Mini-batch optimization addresses the issue of an increase in computation time as the number of experimental conditions increases[34–37]: at each step of parameter optimization, a random subsample—a mini-batch—of the training data informs the optimization process[37,38]. Hence, the training requires simulation of only a fraction of the experimental conditions per optimization step, which leads to a drastic reduction of computation time[34,37], and may help to avoid convergence towards saddle points during optimization[39].

Sophisticated implementations of many mini-batch optimization algorithms are available in state-of-the-art toolboxes for neural nets, such as TensorFlow[36]. Conceptually, these frameworks can mimic simple ODE-solver schemes, e.g., a forward Euler integration[40]. However, it is well-known that ODE models in systems biology typically exhibit stiff dynamics. This makes it necessary to employ implicit solvers with adaptive time stepping[41]. Hence, it is essential to combine advanced methods from both fields, deep learning and ODE modelling. Furthermore, it is not clear how hyperparameters of mini-batch optimization methods, such as the mini-batch size, the learning rate, or the optimization algorithm affect the optimization process for ODE models.

We implement various mini-batch optimization algorithms for ODE models. We benchmark these algorithms on small- to medium-scale ODE models, identify the most important hyperparameters for successful parameter optimization, and introduce algorithmic improvements, which we tailor to ODE modelling. Then, we transfer the approach to a large-scale model of cancer signaling[10], which we train on a dataset comprising 13,000 experimental conditions—an unprecedented scale for training an ODE model. For this application example, we benchmark our approach against state-of-the-art methods[10], achieving better optimization results while reducing the computation time by more than an order of magnitude. To the best of our knowledge, this is the first study integrating advanced training algorithms from deep learning with established tools from ODE modelling.

## Results

**Implementation of mini-batch optimization algorithms for ODE models.** We assume the time evolution of a vector of state variables $x(t)$ to be given by the ODE system

$$\dot{x} = f(t, x(t, \theta, u^e), \theta, u^e), \quad x(0) = x_0(\theta, u^e), \quad (1)$$

where $\theta$ is the unknown model parameters and $u^e$ is a vector of known input parameters, which determine the simulated experiment indexed by $e$. Hence, $u^e$ encodes an experimental condition, i.e., a distinct initial value problem, such as a specific biological perturbation experiment. The inference of model parameters $\theta$ from experimental data is based on reducing a distance measure between simulated model outputs and measurements. In practice, the distance metric is often based on the assumption that the measurement noise is normally distributed and independent for each data point. The corresponding negative log-likelihood function $J$, which serves as objective or cost function, is (up to a constant, more details in the Methods section and in Supplementary Note 1) given by the sum of weighted least squares:

$$J(\theta) = \frac{1}{2} \sum_{e=1}^{M} \sum_{i=1}^{N_e} \frac{\left( \bar{y}_{e,i} - y_{e,i}(\theta) \right)^2}{\sigma_{e,i}^2} \quad (2)$$

Here, $M$ denotes the number of different experimental conditions, $N_e$ the number of measured data points for condition $e$, $\bar{y}_{e,i}$ are the measured data points, $y_{e,i}$ are the observables from the model simulation, and $\sigma_{e,i}$ denotes the standard deviation for the data point $\bar{y}_{e,i}$. If the system has $M$ experimental conditions, this means that the underlying ODE model must be solved $M$ times to evaluate the (full) objective function. A more detailed explanation of this aspect is given in the Methods section, an explanation in a more general context is given in Supplementary Note 1.

Classic (full-batch) optimization methods evaluate the full objective function, i.e., simulate all experimental conditions, in each iteration of parameter optimization (Fig. 1a). In contrast, mini-batch optimization methods evaluate only the contribution to the objective function coming from a randomly chosen subset, a mini-batch, of experimental conditions in each step[37,38]. The cycle until the whole dataset has been simulated, i.e., the computational equivalent of one iteration in full-batch optimization is called an epoch. Typically, each experimental condition is simulated only once per epoch, i.e., the experimental conditions are drawn in a random, but nonredundant fashion (Fig. 1b). In this way, mini-batch optimization allows to perform more—but less informed—optimization steps than classic full-batch approaches in the same computation time.

Various algorithms exist for full-batch and mini-batch optimization and each algorithm is influenced by different hyperparameters and optimizer settings. For full-batch optimization methods such as BFGS[42] and interior-point algorithms[43], many hyperparameters are associated with stopping conditions

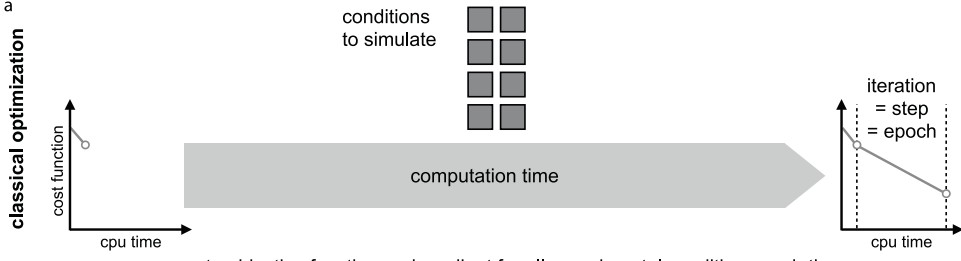

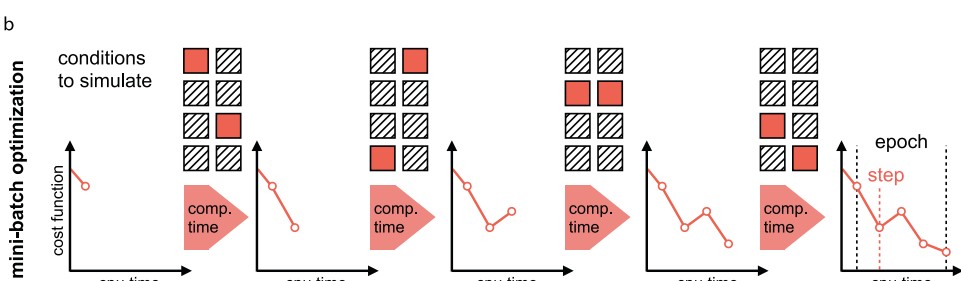

**Fig. 1 Visualization of full-batch and mini-batch optimization. a** Classic full-batch optimization methods evaluate the contribution of all data points—and thus all experimental conditions—to the objective function in each step. The computation time scales linearly with the number of independently evaluable experimental conditions (depicted as gray squares). **b** In mini-batch optimization, the independent experimental conditions are randomly divided into disjoint subsets, the mini-batches (depicted as squares). Per the optimization step, only the contribution of the chosen mini-batch is evaluated (red squares). Hence, possibly many optimization steps can be performed during one epoch, which is the time until the whole dataset has been evaluated.

**Table 1 Overview of ODE models for benchmarking mini-batch optimization.**

| Model name | State variables | Parameters | Conditions | Data points (synthetic) | Reference |
|---|---|---|---|---|---|
| Fujita | 9 | 19 | 600 | 6000 | [81] |
| Bachmann | 25 | 40 | 1200 | 12,000 | [16] |
| Lucarelli | 33 | 72 | 1500 | 60,000 | [82] |

and at least good rules-of-thumb exist for their choice. For mini-batch optimization, there are various critical and less studied hyperparameters, e.g., the learning rate, which controls—but is not identical to—the size of the optimization step in parameter space, and research on how to tune these hyperparameters is still ongoing[44]. In order to apply mini-batch optimization methods to ODE models and benchmark the influence of these hyperparameters, we implemented some of the most common algorithms in the parallelizable optimization framework parPE[14]: stochastic gradient descent (SGD)[38], stochastic gradient descent with momentum[31,45], RMSProp[46], and Adam[47] (see also Supplementary Note 2 and the Algorithms 1, 2, 3, and 4 provided therein). This allowed a direct comparison with the implemented full-batch optimizers when using multi-start local optimization. More importantly, our implementation in parPE combines state-of-the-art numerical integration methods available in the SUNDIALS solver package[48] and adjoint sensitivity analysis for scalable gradient evaluation[49], since simple schemes (such as Euler's method) cannot be expected to yield reliable results for this problem class.

**Mini-batch size and learning rate schedules have a strong influence on optimizer performance.** To evaluate the available mini-batch optimization algorithms for ODE models, we considered three benchmark problems (Table 1, models were taken from ref. 25 and adapted for the creation of synthetic data). To facilitate the analysis of the scaling behavior with respect to the number of experimental conditions, we generated artificial data

(Fig. 2a). Details on the three benchmark examples and on the artificial datasets are given in the Methods section.

We used a mini-batch size of 30 experimental conditions and 50 epochs of training—corresponding to roughly 50 iterations of a classic full-batch optimizer—which are typical hyperparameter choices in deep learning[37]. We benchmarked the four implemented optimization algorithms: SGD, SGD with momentum, RMSProp, and Adam (details are given in the Methods section). To assess the impact of the learning rate, we considered four learning rate schedules:

- Schedule 1: High learning rate, logarithmically decreasing
- Schedule 2: Medium learning rate, logarithmically decreasing
- Schedule 3: Low learning rate, logarithmically decreasing
- Schedule 4: Between low and medium learning rate, constant

Details on these choices are given in the Methods section. The well-established full-batch optimizer Ipopt[43] was used as a benchmark and was granted 50 iterations, so all tested methods had a similar computational budget. For each model, 100 randomly chosen initial parameter vectors were created, from which all optimizers were started. To assess the overall performance of each optimizer setting, we sorted the starts by their final objective function value and each of the 100 starts was ranked across the optimizer settings. Computing the mean of the 100 rankings for each setting led to an averaged rank, which we used as a proxy for overall optimization quality (Fig. 2a).

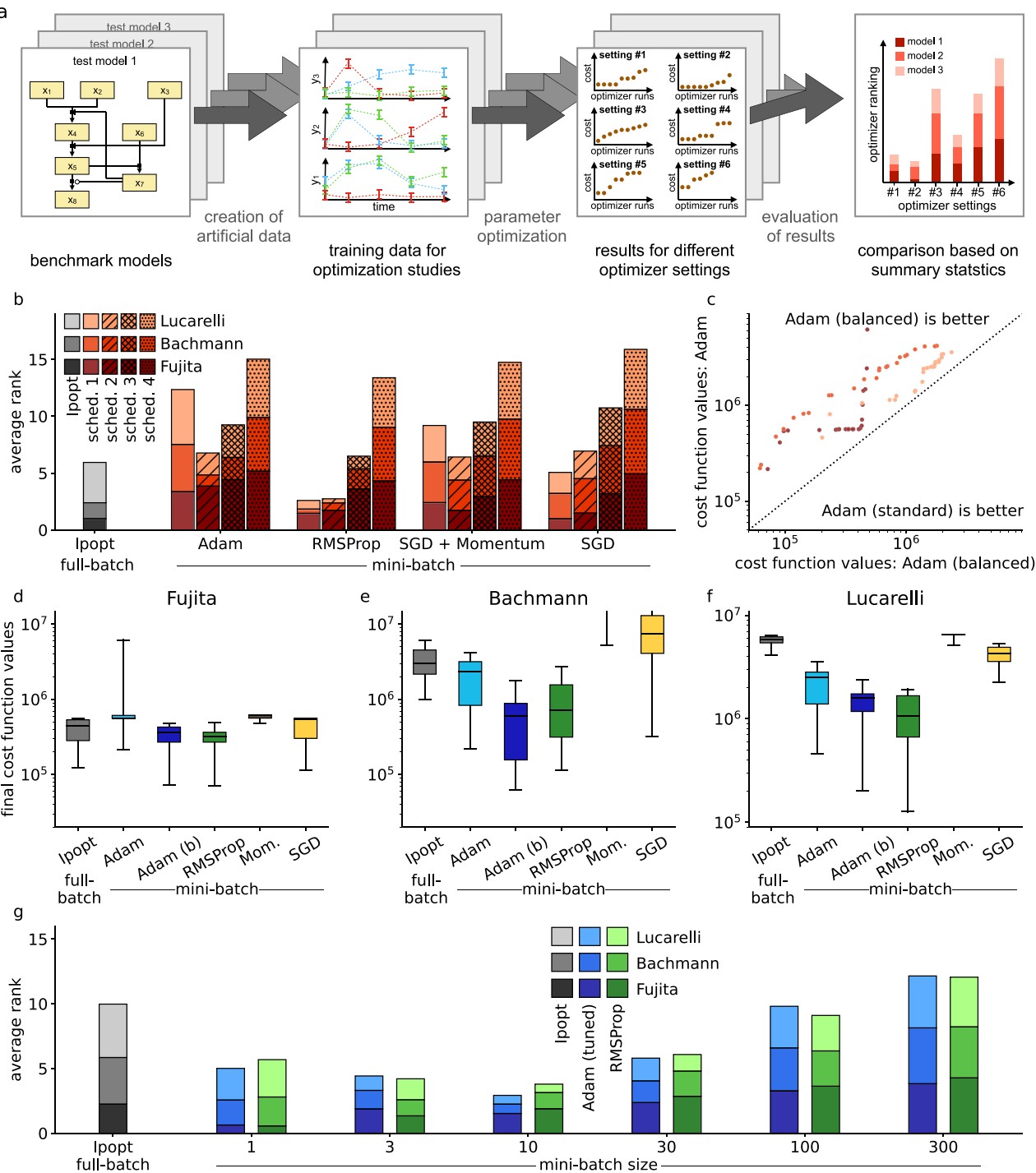

**Fig. 2 Benchmarking full-batch vs. mini-batch optimizers on small- to medium-scale models. a** Overview of optimizer comparison: Benchmark models were simulated, noisy artificial data created, 100 initial points were randomly sampled, and different local optimizers started, each start was ranked between optimizers, and an averaged score was computed. **b** Comparison of performance for different local optimizers with different learning rate schedules (lower rank implies better performance, ranks averaged over models). **c** Top 25 starts of the local optimizer Adam with tuning parameters taken from the literature (standard) vs. a simplified version (balanced). **d**–**f** Boxplots of final cost function values for the best 25 starts of the investigated mini-batch optimizers including the balanced version of Adam, denoted as Adam (b), compared against the lpopt (full-batch optimizer), for each model. Bold lines indicate medians, boxes extend from 25th to 75th percentiles, whiskers show the ranges of the data. **g** Comparison of all starts of the best two mini-batch optimizers given the learning rate Schedule 2, for different mini-batch sizes, compared against lpopt (ranks averaged over models).

For all algorithms except SGD, Schedule 2 was a reasonable choice. For SGD, Schedule 1 was better and for RMSProp, Schedules 1 and 2 performed approximately equally well. Schedule 4 was the worst choice (Fig. 2b). A higher learning rate in the beginning of the optimization process seemed to be beneficial for the mini-batch optimizers to progress quickly towards favorable regions of the parameter space (Supplementary Figs. 1, 2, and 3). Using Schedule 2, different algorithms were able

to compete with or even outperform the full-batch optimizer Ipopt, but the adaptive algorithm RMSProp performed particularly well. In most cases, the preferred learning rates led to step-sizes during optimization that were comparable or slightly lower than those which were chosen by classic (full-batch) optimization methods (Supplementary Fig. 4).

Given these findings, we compared the optimization algorithm Adam—which is maybe the most popular algorithm for training deep neural nets—with two different tuning variants: the tuning proposed in the original publication (called standard, see ref. [47]) and a simplified scheme (called balanced), which employs the same rate for both internally used decaying averages (see Methods for more details). The analysis of the best 25 starts for all models with Schedule 2 showed that the balanced version outperformed the original one for all cases on our benchmark examples (Fig. 2c and Supplementary Figs. 1, 2, and 3). When comparing the performance of the balanced version of Adam and RMSProp with Schedule 2, we see that they show very similar performance for the best 25 starts for all three tested models and outperform the remaining algorithms (Fig. 2d–f).

We then assessed the impact of the mini-batch size on the optimization result. Again, we used an average ranking, 100 starts, and investigated six mini-batch sizes for each model. We restricted our analysis to the two previously best-performing optimization algorithms, balanced Adam and RMSProp, with Schedule 2. We found that in general, small mini-batch sizes of about 0.1 to 1% of the whole dataset were preferred, but the optimal size seemed to be model-dependent (Fig. 2g and Supplementary Figs. 5, 6, and 7). Interestingly, the mini-batch size seemed to impact both optimization algorithms to the same degree. A more comprehensive analysis of the optimization results, which does not rely on summary statistics, is given in the Supplementary Information, including Supplementary Figs. 1 to 7.

**Combining mini-batch optimization with backtracking line-search improves the robustness of the optimization process.** A common challenge when performing parameter estimation for ODE models are regions in parameter space for which the numerical integration of the ODE is difficult or even fails. This may happen due to bad numerical conditioning of the problem or simply divergence of the solution[48]. Full-batch optimizers use line-search or trust-region approaches[50], which can deal with these non-evaluable points by adapting the step-size (Fig. 3a). We found these problems also present in our benchmark examples (Fig. 3b, left), leading to failure of local optimization processes as available mini-batch optimization methods cannot handle failures of the objective function evaluation (probably because it is not encountered in deep learning). Hence, we implemented a so-called rescue interceptor, which attempts to recover a local optimization by undoing the previous step and acting like a one-dimensional trust-region implementation (more details in the Methods section and in Supplementary Note 3, in particular, Algorithm 5). In some cases, these failures occurred at the initial points of optimization. These initial failures cannot be recovered and will lead to a failed local optimization, even when using the rescue interceptor. In all of the remaining cases, the rescue interceptor was able to successfully recover the respective local optimization (Fig. 3b).

In the previous batch of tests, the best optimization performance was achieved with the learning rate of Schedule 2, while a higher learning rate, i.e., Schedule 1, obstructed the optimization process (Fig. 3c). Overall, higher learning rates tended to be beneficial and as it is a priori not clear for a given model what a good learning rate would be, we additionally implemented a backtracking line-search. It reevaluates the objective function without gradient on the same mini-batch for different step-sizes, before accepting a proposed step (Fig. 3d). Details on the implementation can be found in the Methods section and in Supplementary Note 3, in particular Algorithm 6.

We evaluated these two algorithmic improvements for Adam and the learning rate Schedules 1 and 2 on the three benchmark models (Fig. 3e and Supplementary Fig. 8). Interestingly, we found the strongest improvement for the largest model, although it suffered only mildly from integration failure (Fig. 3b). The line-search substantially improved the optimization process at high learning rates, which can be seen in a direct comparison (Fig. 3f) and in the waterfall plot (Fig. 3g). Considering all three models, we saw that the rescue interceptor was generally helpful, whereas the line-search could also reduce the computational efficiency in case a good learning rate had been chosen (Fig. 3e). This is not surprising, as the line-search increased in a few cases the computation time by up to 9% and some optimization runs were stopped prematurely due to imposed wall-time limits (Supplementary Fig. 9). However, these negative effects at lower learning rates were mild when compared against the positive effects at high learning rates and as the selection of a good learning rate is currently a trial-and-error process, the adaptation is highly beneficial.

**Mini-batch optimization enables training of predictive models of the drug response of cancer cell lines.** Following the successful testing and improvement, we evaluated how mini-batch optimization performs when applied to the largest publicly available ODE model of cancer signaling[10]. The model comprises various pathways and their cross-talk and captures 1228 biochemical species and 2686 reactions and was originally developed and provided by Alacris Theranostics. The generic chemical reaction network can be adapted to cancer cell lines and treatment conditions using input parameter vectors. These vectors encode mutation and expression status (based on genome and transcriptome sequencing) and drug concentrations (Fig. 4a).

We extracted all available drug response data from the Cancer Cell Line Encyclopedia[19], which we could match to the model, yielding in total 16,308 data points of viability read-outs. We split the data 80:20 into a training set and an independent test set. The training data is taken from 21 tissues with seven different drugs at eight different concentrations, adding up to 13,000 of the 16,308 data points and experimental conditions (Fig. 4b). The test data comprises the same number of drugs and concentrations and is taken from 59 cell lines from 21 (partly different) tissues, yielding 3308 of the 16,308 data points and experimental conditions. To the best of our knowledge, this is the first time that an ODE model has been trained on such a large dataset derived from so many different experimental conditions.

We performed 100 local optimizations, in which we trained the model for 20 epochs and a mini-batch size of 100, using the optimization algorithm Adam (balanced) with rescue interceptor as well as line-search. As in the Adam algorithm, the step-size during optimization scales with the square root of the problem size, we adapted the learning rate schedule such that it yields a step-size comparable to those for Schedule 2 on the small- to medium-scale examples (details on these hyperparameter choices are given in the Methods section). We considered the best 10 optimization results for the creation of an ensemble model, similar to ref. [51]. Based on this approach, we found a Pearson correlation of 0.76 of the simulation of the trained ensemble model with the training data (Fig. 4c). We then used our trained model to classify treatments for specific cell lines into responding and non-responding situations. Therefore, we considered a cell line to be responsive to a particular treatment, if the viability of

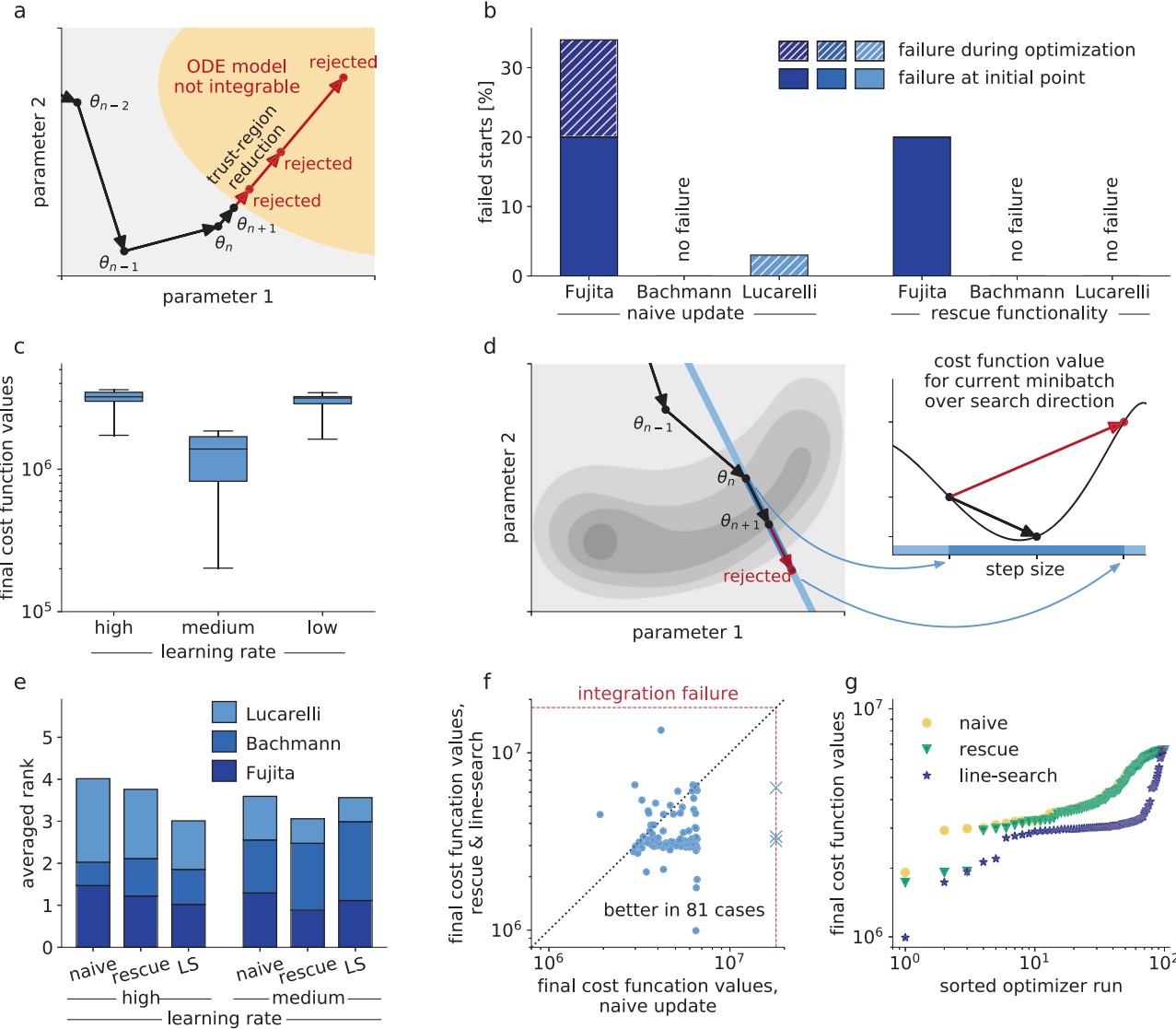

**Fig. 3 Influence of line-search methods on optimizer's performance and reliability. a** Schematic of the rescue interceptor, which tries to recover from failed model evaluations, based on one-dimensional trust regions. **b** Percentage of failed local optimizations per model (with optimizer Adam) due to non-integrability of the underlying ODE. Failure at the initial point of optimization cannot be recovered, but failure during the optimization process is prevented when applying the rescue interceptor. **c** Boxplots for the best 25 starts of mini-batch optimizers Adam for learning rate Schedules 1–3 for the largest example (Lucarelli), showing that too high learning rates obstruct the optimization process. Medians are indicated as thick lines, boxes extend from 25th to 75th percentiles, whiskers show the ranges of the data. **d** Line-search for mini-batch optimizers is implemented based on backtracking while keeping the mini-batch fixed during line-search. **e** Comparison of performance for optimizer Adam, given different learning rates, for naive implementation, with rescue interceptor, and rescue interceptor and line-search, denoted as LS (ranks of models averaged). **f** All starts of the local optimizer Adam for the largest of the three examples (Lucarelli), naive implementation compared against rescue interceptor and line-search, employing learning rate Schedule 1. **g** Waterfall plot for the largest of the three examples (Lucarelli), for naive implementation of Adam, with rescue functionality and with rescue interceptor and line-search.

the corresponding cell line was reduced by more than 50%. When computing the receiver-operating-characteristic (ROC)[52] based on the trained ensemble model, we achieved an AUC value of 0.96 (Fig. 4d). Interestingly, the AUC values when relying on single optimization runs instead of an ensemble were lower (between 0.91 and 0.94). After computing the ROC for all data, an ideal classification threshold was computed and used to classify whether a simulation is responsive. The classification accuracy was computed from this classification. This was repeated for each drug individually, as well as with all data (all drugs). On the training data, the ensemble model achieved a classification accuracy of 86% (Fig. 4e). Beyond the correlation and ROC analysis, we also analyzed the model fits to drug response data. The trained model was able to describe the response to different

drug treatments and also to capture the varying behavior between the 233 cell lines correctly (Fig. 4f and Supplementary Information).

In the next step, we validated predictions of the model on cell lines from the independent test set, where we still found a Pearson correlation between the data and the simulation of the ensemble model of 0.74 (Fig. 5a). The ROC analysis for the classification into responding and non-responding treatments on the test set yielded an AUC value of 0.94 for the ensemble model values between 0.90 and 0.92 for the ten best local optimizations (Fig. 5b), while the classification accuracy was around 85% (Fig. 5c). Importantly, the model did not only classify trivial cases correctly but was also able to capture the variability between cell lines and drugs (Fig. 5d).

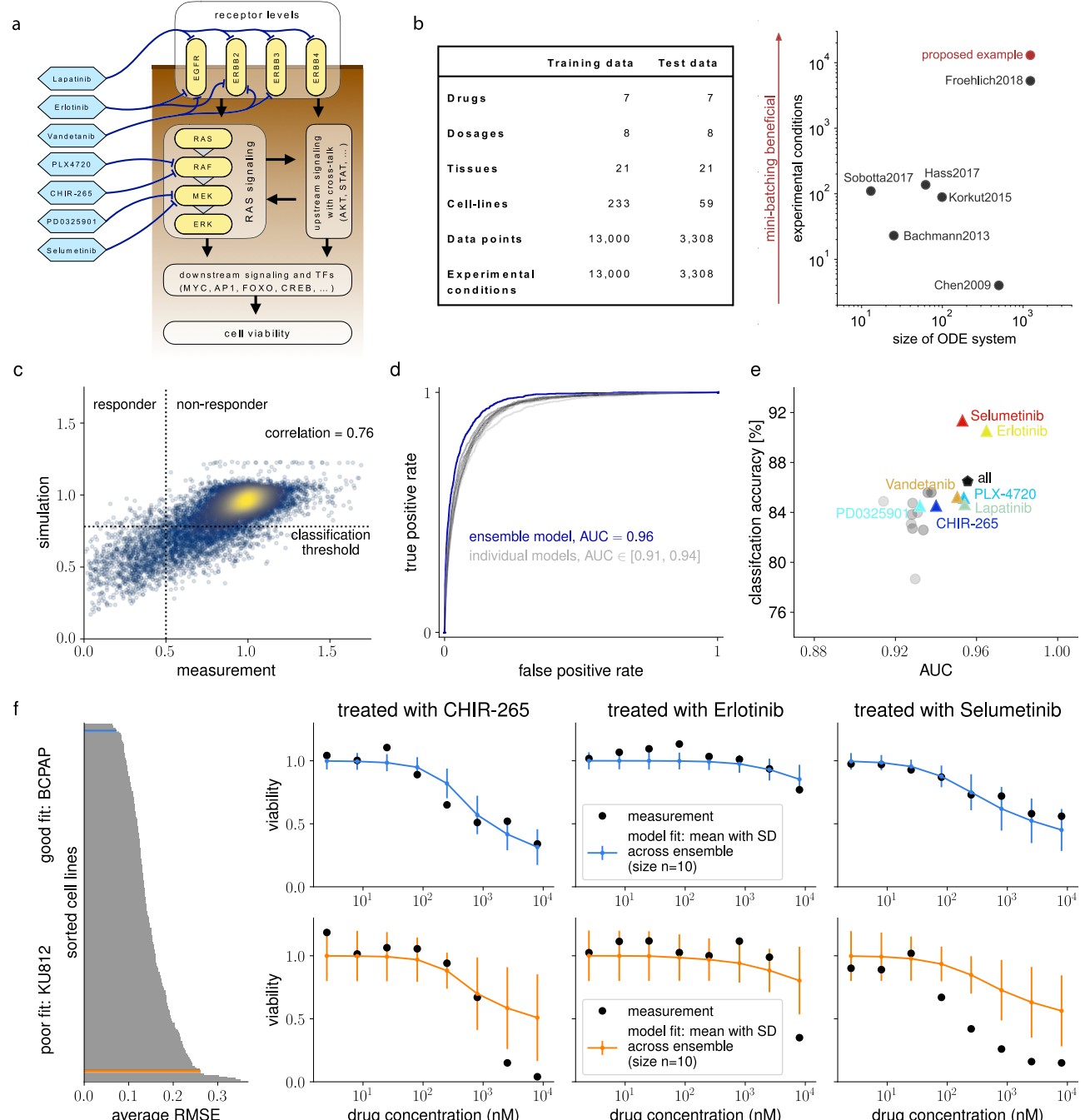

**Fig. 4 Description of application example, datasets, and model performance, when trained with mini-batch optimization. a** Simplifying illustration of the multi-pathway model of cancer signaling. **b** Left: Overview of the datasets used for training and model validation, taken from the Cancer Cell Line Encyclopedia. Right: Comparison of model sizes and experimental conditions used for model training of recently published ODE models. **c** Correlation of measured and simulated cell viability for all points of the training data, color-coding indicates density in scatter plot. **d** Receiver-operating characteristics for classification into responsive and nonresponsive combinations of cell lines and treatments on training data for the best ten optimization runs (gray) and an ensemble simulation (blue). **e** Area under ROC curve and classification accuracy on training data for the ten best optimization results (gray), for the ensemble model (black), and for the ensemble model on data for each drug individually (colored). **f** Simulated drug response. Left: Ranking of fit quality for cell lines by average root-mean-square error (RMSE). Right: Two out of 233 cell lines from the training data, error bars indicate the standard deviation across an ensemble of the $n = 10$ best optimization runs, for a cell line which the model was able to describe well (blue, BCPAP) and a cell line, which was less well captured by the model (orange, KU812).

**Mini-batch optimization renders iterative model refinement possible for large-scale ODE models.** Beyond predicting drug response, we also validated the model on experimental data from a recently published CRISPR screen: Behan et al. had analyzed the change in cell viability when knocking out various genes on a large

set of cell lines[53]. We found 107 genes and 18 cell lines in the knockout dataset, which were also included in our application example and the dataset used for model training, respectively, summing up to 1926 data points which we used for validation. Based on the gene knockout data, we classified a gene as

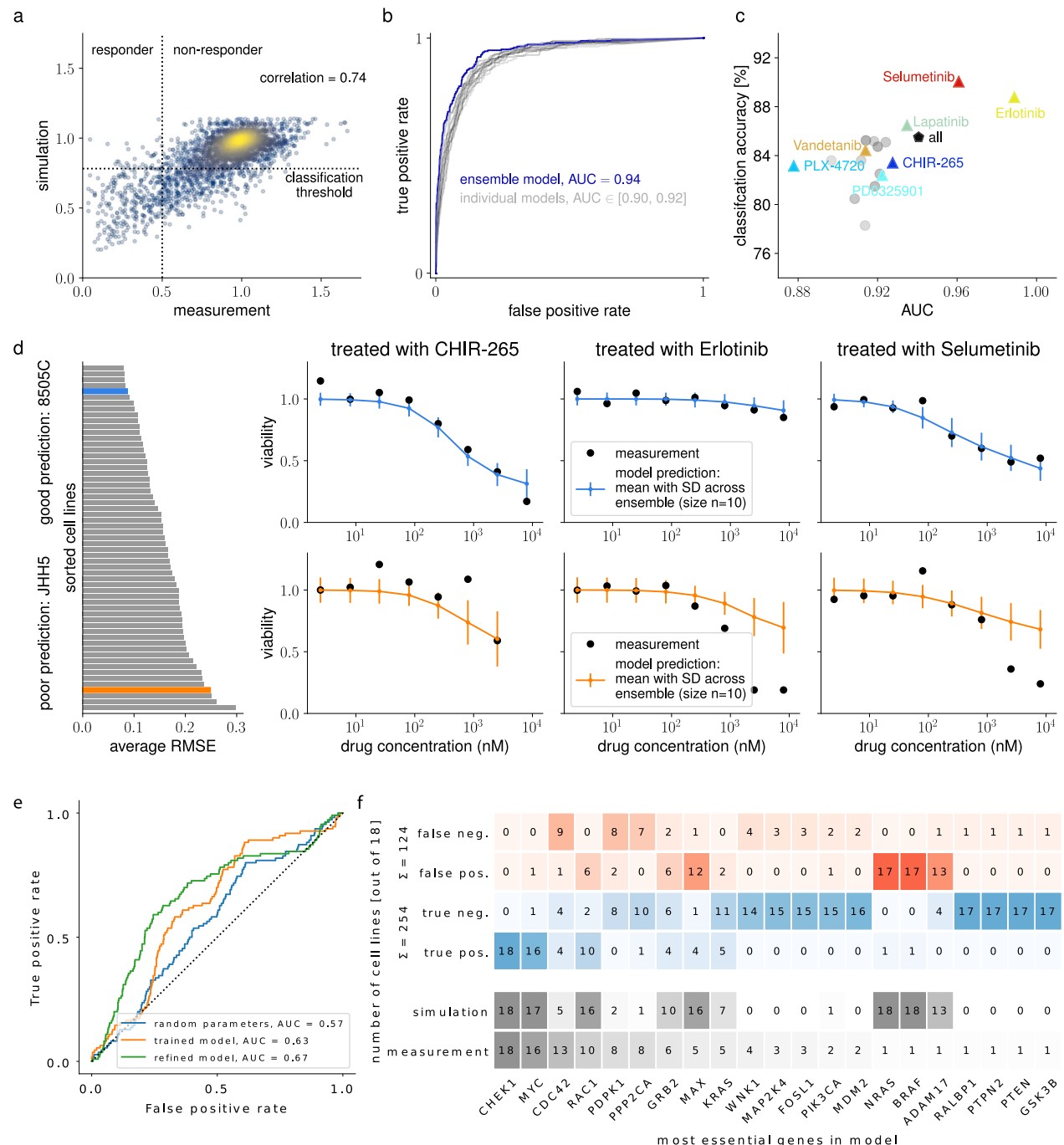

**Fig. 5 Validation of the fitted large-scale cancer model on independent test data. a** Correlation of test data and model prediction. Color-coding indicates density in a scatter plot. **b** ROC curves for classification of drug responses of cell lines on the test set. Classification thresholds from the training data were used. **c** Area under ROC and classification accuracy on test data for the ten best optimizations (gray), the ensemble model (black), and the ensemble model for each drug individually (colored). **d** Simulated drug response. Left: Ranking of fit quality for cell lines by average root-mean-square error. Right: Two out of 59 cell lines from the test data, error bars indicate the standard deviation across an ensemble of the $n = 10$ best optimization runs, for a cell line which the model was able to describe well (blue, 8505C) and a cell line, which was less well captured by the model (orange, JHH5). **e** ROC curves for classification of gene essentiality. Measurement data for 18 cell lines were taken from Behan et al., 2019. In silico knockouts are shown for the untrained model (blue), the trained model (orange), and the refined and trained model (green). **f** Measurement, prediction, and confusion matrix for essential genes for the refined and trained model. Numbers indicate how often a gene was found to be essential in experimental data and in silico knockout predictions for the 18 cell lines, sums show the number of true and false predictions over essential genes.

"essential" for a cell line, if the knockout led to a viability reduction of more than 50% and compared this essentiality classification with in silico knockouts from our application example (Fig. 5e). We computed receiver-operating characteristics for knockout simulations based on our trained model, and on random parameters as reference. We obtained better-than-random predictions (AUC of 0.59) from the untrained model and, as expected, better predictions for the trained model (AUC of 0.63).

Yet, the classification threshold for model simulation chosen by the ROC analysis was surprisingly high, leading to many false-

positive predictions (Supplementary Fig. 10). As this indicated a shortcoming of the model, we added further proteins, which should contribute to the viability readout of the model (see Methods section and Supplementary Note 4 for more details on how the viability readout is computed): First, we included the protein Checkpoint kinase 1, a major regulator of the cell cycle that is encoded by the gene *CHEK1*, as additional pro-proliferative readout. The gene *CHEK1* was shown to be among the most essential ones for cell viability in the validation data, which was however not accounted for in the original model of ref. [10]. Secondly, we also included doubly phosphorylated MAP3K1 as pro-proliferative readout, which was so far only acting in a feedback loop with upstream proteins. This modification allowed the model to partly circumvent the downstream bottleneck of the MAPK-cascade, and hence made it possible to explain the potential downstream effects of MAP3K1. After calibrating this refined model on the previously used drug response data, we obtained substantially improved predictions of gene essentiality (Fig. 5e, AUC of 0.67). The classification threshold for the model readout was strongly reduced, leading to substantially fewer false-positive predictions (Fig. 5f and Supplementary Fig. 11). A detailed analysis of the remaining false predictions is given in Supplementary Note 5.

As confirmation of the refined model topology, we refitted the original model on the drug response and gene knockout data, i.e., 14,926 data points, simultaneously. As expected, calibrating the model towards the knockout data led to substantially improved classification results (AUC of 0.74, vs. 0.63 for the model trained on drug response data only). However, also when being refitted on the knockout data, the model was still not able to describe the data when knocking out *CHEK1*, confirming the limitations of the original viability readout (Supplementary Fig. 10). In the last step, we refitted also the improved model on drug response and knockout data simultaneously, which again drastically improved the classification accuracy (AUC of 0.84, vs. 0.67 for the model trained on drug response data only). This suggested that a more general ODE model of cell signaling should encode at least a basic version of the cell cycle if the model is to explain viability. To the best of our knowledge, this is the first time that a systems-biology modelling loop (see Supplementary Fig. 12), i.e., iterative model refinement and recalibration, has been carried out for an ODE model of this size.

**Mini-batch optimization outperforms full-batch optimization by more than an order of magnitude.** As mechanistic modelling of biological processes relied so far on full-batch optimization techniques, we reevaluated our findings about hyperparameter tuning from the smaller models on the large-scale application example: in a first step, we compared 20 epochs of mini-batch optimization with Adam, mini-batch size 100, and two learning rate schedules—a "high" and a "low" learning rate—with and without line-search, always using the rescue interceptor, which enabled substantial improvements for this model (Supplementary Fig. 13). In a second step, we ran additional optimizations with mini-batch sizes 10, 100, 1000, and 13,000 (full-batch), granting 10, 20, 50, and 150 epochs of optimization time and 100, 100, 50, and 25 local optimizations, respectively. As a benchmark, we performed 150 iterations with Ipopt—which also employs a line-search algorithm—and restricted to 20 local optimizations due to the high computation time. All optimizers were initialized with the same parameter vectors. We also took snapshots of the optimization process with Ipopt at computation times that were as close as possible to those used by the mini-batch optimizations.

In the learning rate study, mini-batch optimization at low learning rates achieved slightly better results after 20 epochs than Ipopt after 150 iterations in terms of objective function values (Fig. 6a) and of correlation with measurement data (Fig. 6b). Moreover, the model also generalized slightly better to cell lines from the independent test data when trained with mini-batch optimization (Fig. 6c), while reducing the total computation time by a factor of 4.1 (Fig. 6d). Optimization was faster but less successful when using the high learning rate. However, it clearly improved when adding the line-search feature, which increased the computation time by less than 13%. For the lower learning rate schedule, line-search had almost no effect. We assessed the computation time until convergence was reached for the first time and the number of converged starts per computation time (see Methods for details on these convergence criteria) as additional performance measures. Now granting 100 starts to mini-batch optimization, i.e., a computational budget similar to Ipopt, we found that mini-batch optimization was up to 6.9-fold faster than full-batch optimization when comparing the number of converged starts per computation time. In terms of computation time, until first convergence was achieved, mini-batch optimization outperformed full-batch optimization by 27-fold (Supplementary Fig. 14). However, the latter was achieved using the higher learning rate, which was generally less effective. Overall, these results confirm that the learning rate is a crucial hyperparameter for mini-batch optimizers also when working with ODE models and that line-search can markedly improve results when working with high learning rates.

In the mini-batch size study, we found a strong benefit of smaller mini-batch sizes, which improved objective function and correlation values and clearly outperformed Ipopt (Fig. 6e, f). This result generalized to the independent test data, where mini-batch optimization with the smallest batch size 10 achieved better correlation values than all previously tested optimization approaches (Fig. 6g), possibly due the regularizing effect of small batch sizes and hence possibly less overfitting of the training data[54]. In addition to the markedly improved optimization results, which were best visible from the waterfall plot (Supplementary Fig. 15), the total computation time was reduced by more than a factor of 10 when compared to Ipopt (Fig. 6h). When comparing computation time to first convergence, mini-batch optimization was up to 52 times faster than Ipopt. In terms of converged starts per computation time, we found an 18-fold improvement when using mini-batch optimization. As an additional test, we assessed the influence of the optimization algorithm on the optimization result (Supplementary Fig. 16). This indicated again that the chosen algorithm was less important than the choice of the learning rate or the mini-batch size.

**Mini-batch optimization improves parameter space exploration and uncertainty analysis.** In a last set of tests, we further analyzed the optimization result with the smallest batch size, i.e., 10. For this purpose, we created a large parameter ensemble from the optimization history based on an objective function threshold, similar to the ideas of ref. [55] (more details on the ensemble generation can be found in the Methods section). This ensemble contained 8450 parameter vectors from 52 out of the 100 local optimization runs. We also generated a second, smaller ensemble using only the final results of the ten best optimization runs (Fig. 7a). In addition, we investigated the coverage of the feasible interval across the different local optimization runs for each model parameter at three different checkpoints during optimization (Fig. 7a). Comparing these parameter interval coverages from mini-batch optimization with those from full-batch optimization revealed that mini-batch optimization substantially increased coverage, and thus, yielded a better exploration of the parameter space (Fig. 7b).

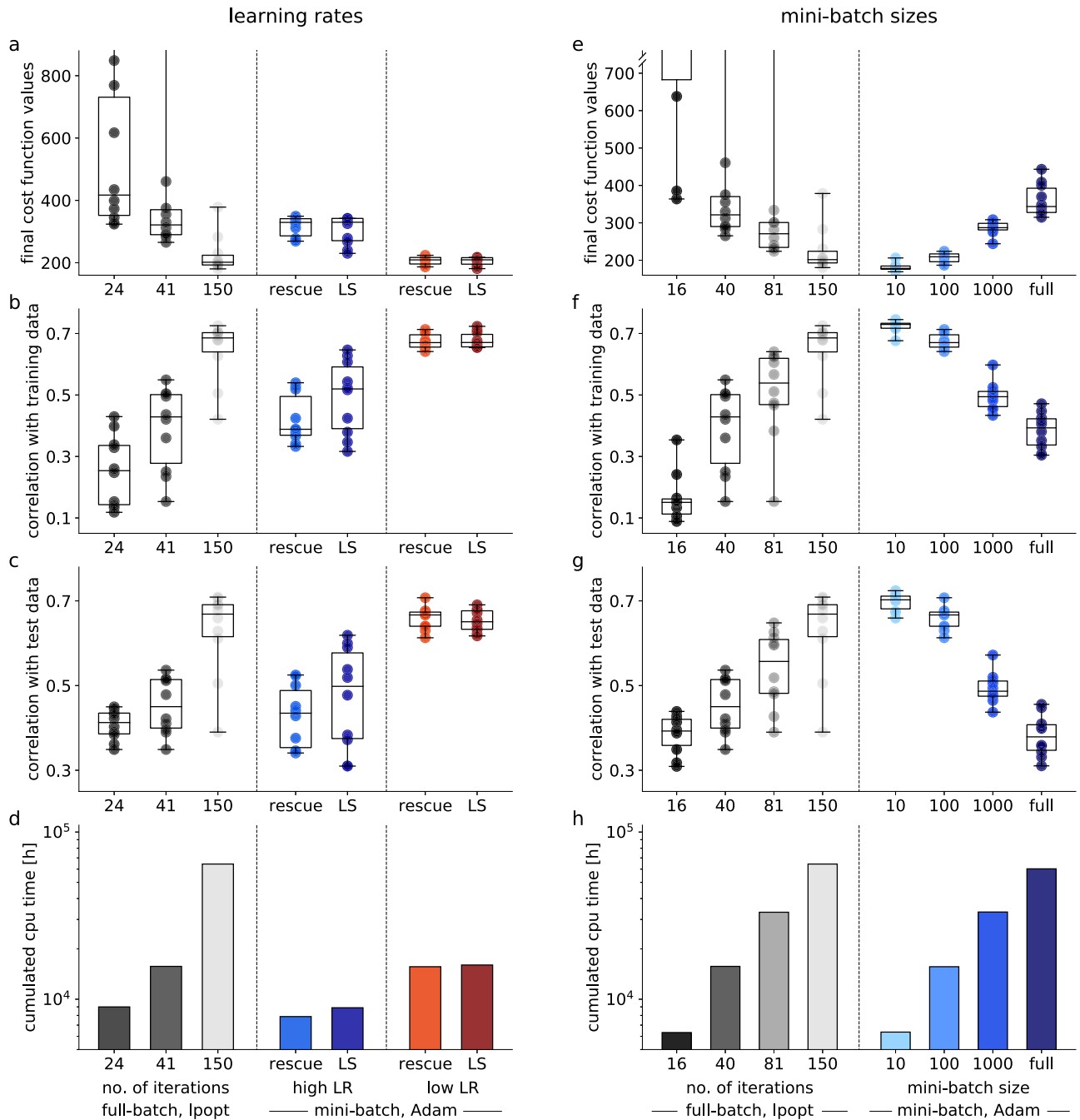

**Fig. 6 Comparison of optimization results for the large-scale cancer model, for different hyperparameters. a–c** Boxplots of the ten best optimization runs out of 20, started at the same random parameters, for Ipopt (at three different stages of the optimization process to compare performance over computation time) and for mini-batch optimization with different learning rates (LR), with rescue interceptor only (rescue) and with additional line-search (LS). Boxes extend from 25th to 75th percentiles, whiskers show the ranges of the data, and thick lines indicate medians. **a** Final objective functions values. **b** Correlation of model simulation with measurement data (training set). **c** Correlation of model simulation with measurement data (test set). **d** Total computation time for all 20 optimization runs (lower panel). **e–g** Boxplots of the ten best optimization runs out of 20, started at the same random parameters, for Ipopt (at four different stages of the optimization process to compare performance over computation time) and for mini-batch optimization with mini-batch sizes (10, 100, 1000 and full-batch, i.e., 13,000). Specifications as described for subfigures **a–c**. **e** Final objective functions values. **f** Correlation of model simulation with measurement data (training set). **g** Correlation of model simulation with measurement data (test set). **h** Total computation time for all 20 optimization runs (lower panel).

To assess the uncertainty of the calibrated model parameters, we analyzed the parametric sensitivities of the model output on the ten best optimization results. A singular value decomposition of this sensitivity matrix showed that 3029 directions in parameter space were numerically nonzero, indicating that the model could acquire information about more than 3000 degrees of freedom. Yet, many directions in parameter space were poorly determined (Fig. 7c). Complementary to this sensitivity analysis, we analyzed the covariance structure of the large parameter ensemble. A principal component analysis (PCA) revealed that

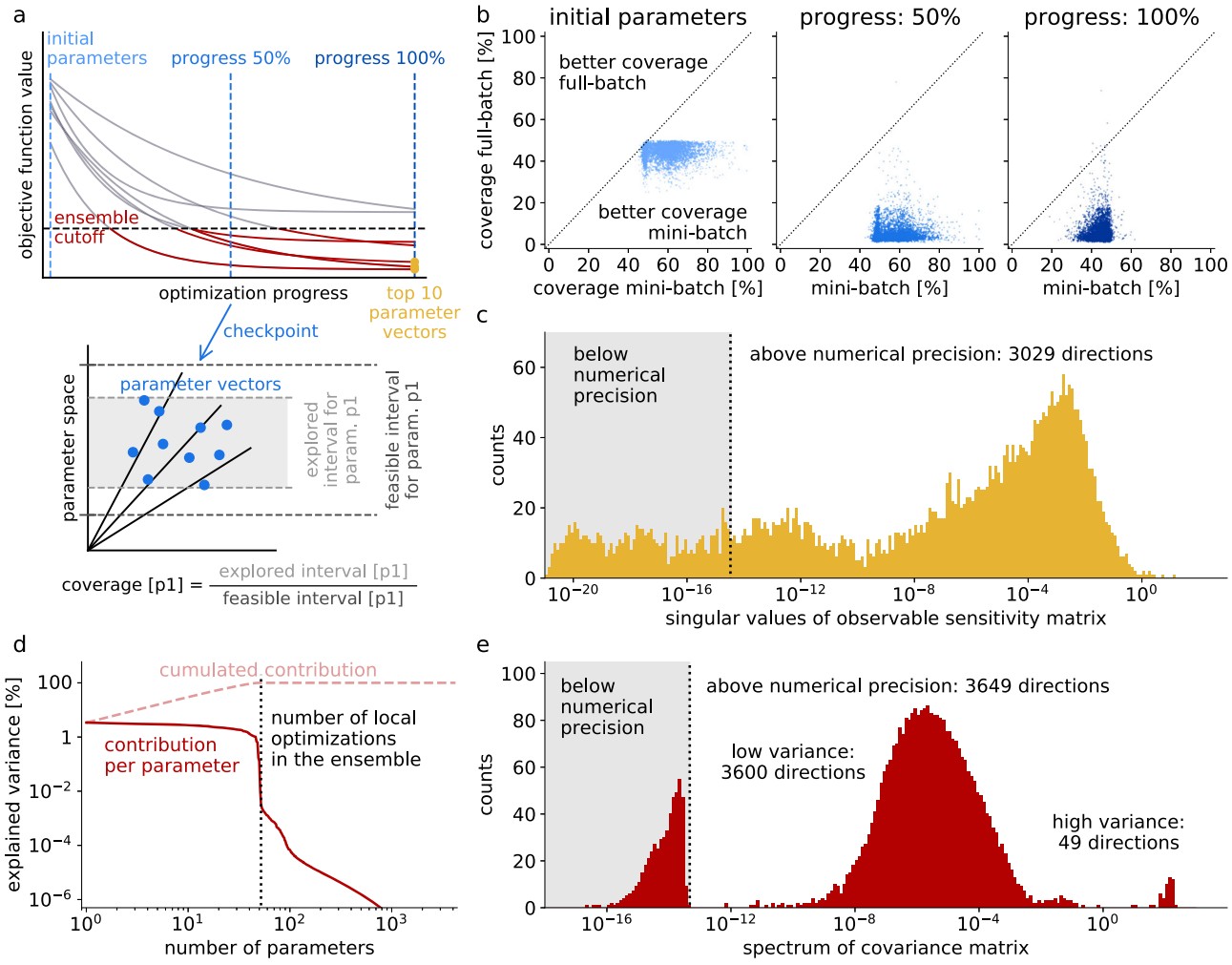

**Fig. 7 Analysis of parameter uncertainty based on an ensemble of parameter vectors from the the best mini-batch optimization, i.e., with batch size 10. a** Generation of a large parameter ensemble (8,450 vectors) based on the history of multi-start local mini-batch optimization. A cutoff value was chosen and parameter vectors added to the ensemble if the objective function value was below the cutoff (red). A smaller ensemble was created from the ten best optimization results (yellow). At three checkpoints (blue), the coverage of parameter intervals was computed as a ratio of the interval covered by the optimization runs over the interval between the parameter bounds. **b** Computation of parameter interval coverage based on multi-start local optimization with mini-batch optimization and Ipopt (full-batch optimization). **c** Sensitivity analysis for model output (cell viability) w.r.t. the 4232 model parameters. Singular values of the sensitivity matrix computed on the small parameter ensemble. **d** Principal component analysis of the large parameter ensemble created from mini-batch optimization, indicating the contribution (individual and summed) of the different PCA directions to the total observed variance in the ensemble. **e** Spectrum of the covariance matrix (for the 4232 model parameters) created from the large parameter ensemble from (**d**).

most of the total variance was spread across a low-dimensional subspace, showing a sharp drop in the explained variance after about 50 PCA directions, which coincides well with the number of local optimizations present in the parameter ensemble (Fig. 7d). This indicates that the parameter space was not yet sufficiently explored, although mini-batch optimization allowed to afford more local optimization runs than full-batch optimization. This was confirmed by a UMAP embedding of the parameter ensemble, which showed that the local optimization runs did not converge to a common optimum, but remained separated in parameter space (Supplementary Fig. 17). The spectrum of the covariance matrix indicated 49 directions with high variability, a second mode containing about 3600 directions with lower, but clearly present variability (Fig. 7e), and the third mode of probably non-explored directions, which had a variability below floating point precision. The overall 3600 explored PCA directions in this large ensemble are in line with the results from the sensitivity analysis (roughly 3000), as the discrepancy between

the two results can be explained with additional information being present in the larger ensemble.

## Discussion

We presented a framework for using mini-batch optimization in combination with advanced methods from dynamic modelling for the parameter estimation of ODE models in systems biology. We introduced algorithmic improvements (tailored to ODE models), benchmarked different methods and their hyperparameters on published models (with artificial data), and identified the most important factors for successful optimization. Then, we applied mini-batch optimization to a particularly large model of cancer signaling and trained it on measured cell line drug response data from a public database. The trained model provided accurate predictions of whether a certain treatment would reduce cell viability by more than 50% for a chosen cell line in more than 85% of the cases, even on cell lines, which had not been used for model training. Furthermore, we performed *in*

*silico* gene knockouts and validated them on a large experimental CRISPR screen from literature, which helped us to improve the model topology and to obtain well-matching knockout predictions. In the last step, we performed uncertainty analysis based on ensemble methods and assessed the exploration of the parameter space during optimization, where mini-batch optimization enabled results, which had not been achievable with established optimization methods.

A closer analysis of the in silico knockout study allowed us to pinpoint deficiencies of the large-scale cancer signaling model, e.g., concerning the implementation of cell cycle genes, and a subsequent model refinement yielded substantially improved model predictions. To the best of our knowledge, this is the first time that a full modelling cycle, including model refinement and retraining, has been performed for an ODE model of this size, as the cost of model calibration renders iterative refinement usually prohibitive for large ODE models. Yet, a set of incorrect predictions remained: in particular, the essentiality of genes that encode master kinases such as PDPK1 or phosphatases such as PPP2CA was underestimated. At the same time, the importance of some genes encoding proteins involved in the AKT and RAS signaling pathways was overestimated, especially those which either interact with the EGF receptor, such as SRC, VAV2, or EGFR itself, or those which acts as transcription factors, such as CREB1, ELK1, or STAT3. Hence, we assume that a more comprehensive model, trained on a larger dataset, would be needed to, e.g., reliably discover new drug targets or to reject possible drug candidates before entering a clinical trial. Possible approaches for an also refined proliferation model are outlined in Supplementary Note 6. The datasets to train such a more comprehensive, whole-cell signaling model have been made available in the last years, and mini-batch optimization renders the calibration of such a model computationally feasible.

In a previous study relying on full-batch optimization[10], it was observed that the error of model predictions resulted rather from overfitting than from prediction uncertainty. It seemed that the fewer passes through the dataset and the stochasticity introduced by the mini-batching in combination with the large dataset have addressed this overfitting. Ensemble modelling, for which mini-batch optimization is particularly well-suited, improved the prediction accuracy further. Overall, our implementation reduced computation time by more than one order of magnitude while providing better fitting results and predictions than established methods. The improved scaling characteristics of mini-batch optimization should render problems with even larger datasets feasible. In addition, the ensemble analysis indicated that a better exploration of the parameter space would be necessary to obtain more reliable uncertainty estimates—a goal, which needs further research on mini-batch optimization methods and which is unlikely to be achievable with full-batch optimization methods.

We identified the choices of learning rate and mini-batch size to be the most influential hyperparameters for optimization and made the three following observations: Firstly, learning rates for mini-batch optimization which yield step-sizes slightly smaller than those used by established optimization techniques—see Supplementary Fig. 4—are a good choice. Secondly, surprisingly small mini-batch sizes were preferred in all of our application examples. Thirdly, the choice of the optimization algorithms seems to be less important, as at least Adam and RMSProp performed equally well on all examples. Therefore, we suggest the following approach for hyperparameter tuning when using mini-batch optimization with ODE models:

- Optimization algorithm: Choose an "adaptive" optimization algorithm: Beyond the here tested RMSProp and (balanced) Adam algorithms, many more algorithms with similar behavior exist, such as recently discussed in ref. [44].

- Learning rate: The learning rate is probably the most problematic, because of strongly model-dependent hyperparameter: Many full-batch optimizers use choices such as $\sqrt{n_\theta}$ as initial step size, with $n_\theta$ being the number of model parameters. Our results suggest that using a learning rate, which results in initial step sizes in the range of $\kappa \cdot \sqrt{n_\theta}$, with $\kappa \in [0.01, 0.1]$, is a reasonable first choice for mini-batch optimization of ODE models. For particularly large models, small values of $\kappa$ were more successful. Furthermore, decreasing learning rate schemes tended to be more helpful for optimizer convergence than constant learning rates. However, at the moment, it seems most helpful to test two or three different learning rate schedules.

- Mini-batch size: Start using very small mini-batch sizes first, and then test the effect of increasing the mini-batch sizes. From our experience, this will lead to finding an appropriate mini-batch size faster than starting with large mini-batch sizes.

- Number of epochs: A recent study has come to the conclusion that the overall number of optimization steps is roughly conserved for full-batch optimizers[25]: for a large group of models, it was typically in the range of a few hundred to at most a one or two thousand optimization steps. Since mini-batch optimization cannot be expected to converge in fewer steps but should do so in fewer epochs, choose the number of epochs accordingly, e.g., in the order of a few dozen epochs at most, depending on the number of optimization steps per epoch.

Overall, learning rates and mini-batch sizes would be promising candidates for auto-tuning schemes. There are various known methods for auto-tuning of step-sizes during full-batch optimization[50,56,57]. Combining those with mini-batch optimization may lead to substantial improvements. We proposed and tested the implementation of a line-search method for mini-batch optimization, which may serve as a starting point. For the mini-batch size, auto-tuning maybe less straightforward, but also here, first approaches exist, which are based on assessing the variance of the objective function gradient across a chosen mini-batch and possibly enlarging the mini-batch size[58]. Other algorithmic improvements—more specific to ODE models—would be combining mini-batch optimization with hierarchical optimization for observation-specific parameters, such as scaling factors or parameters for measurement noise[14,59]. This approach allowed substantial improvements in parameter optimization for ODE models and it is to be expected that also mini-batch optimization would benefit from it. A complementary approach would be to implement variance reduction techniques[60,61]. Some of these methods enjoy good theoretical properties but are demanding in terms of memory consumption, which might make them prohibitive for applications in deep learning, but possibly well-suited for the training of ODE models. Hierarchical optimization, as well as variance reduction, should be combined with methods for early-stopping, to avoid overfitting and to further reduce computation time[62,63]. Other improvements, which would not impact the parameter estimation procedure itself, but could further reduce computation time, might be based on GPU computing[64] or direct methods for steady-state computation[65].

Since mini-batch optimization is computationally more efficient than full-batch optimization when working with large datasets, it is also a promising approach to drastically improve the exploration of parameter space. Especially when using methods such as multi-start local or hybrid local-global optimization[23,24,66], much

more local optimizations can be performed. In our large-scale application example, we confirmed that ensemble modelling leads to better predictions than point estimates[51,67]. Furthermore, there have been recent advances when using ensembles created from the optimization history of an ODE model[55]. Mini-batch optimization is particularly well-suited for these approaches, as it creates more comprehensive optimization histories.

In summary, we showed that combining mini-batch optimization with advanced methods from ODE modelling can help to overcome some major limitations in the field. We hope it will become an actively developed and applied group of methods in systems biology. We think and hope that our work can serve as a foundation for other research groups to further push the boundaries of what is computationally feasible and lead to new, fruitful applications.

## Methods

### Modelling of chemical reaction networks with ordinary differential equations (ODEs).

We considered ODE models with state vector $x(t) \in \mathbb{R}^{n_x}$, describing the dynamics of the concentrations of $n_x \in \mathbb{N}$ biochemical species, e.g., (phospho-) proteins or mRNA levels in a time interval $t \in [0, T]$. The time evolution of $x$ was given by a vector field $f$, depending on unknown parameters $\theta \in \mathbb{R}^{n_\theta}$, e.g., reaction rate constants, and a vector of known input parameters $u \in \mathbb{R}^{n_u}$:

$$\frac{d}{dt}x(t, \theta, u) = f(x(t, \theta, u), \theta, u), \quad \text{with } x(0) = x_0(\theta, u) \quad (3)$$

In our case, input parameters were drug treatments and, for the large-scale application example, also differences between cell lines assessed by mRNA expression levels and genetic profiles. As the ODEs had no closed-form solutions, we used numerical integration methods to solve/integrate Eq. (3). As ODEs in systems biology applications must be assumed to be stiff[41,68,69], we employed an implicit multi-step backward differential formula scheme of variable order. This allowed adaptive time stepping and automated error control, helping to ensure the desired accuracy of the computed results[41,68,69].

To match the model to the data, we used observable functions, which describe (phospho-)protein concentrations for the small- to medium-scale models. For the large-scale model, there is only one observable function (cell viability), described by a combination of downstream signaling activities, which can either act pro- or anti-proliferative[10]:

$$y(\theta, u) = h(x(t, \theta, u), \theta) \quad (4)$$

For the small- to medium-scale models, artificial data $D = \{\bar{y}_{e,i}\}_{e=1,\dots,M, i=1,\dots,N_e}$ was simulated as a time-course and Gaussian noise was added. For the large-scale model, measurements were taken at steady-state, yielding only one data point per experimental condition $e$. The steady-state was inferred by simulating the model to time-point $t = 10^8$ (seconds). Afterwards, we verified whether the model trajectories had reached a steady-state by assessing the $L^2$-norm of the right-hand side of the ODE, using the absolute tolerance of $10^{-16}$ and relative tolerance of $10^{-8}$ as convergence threshold. Distinct experimental conditions differed through their vectors of input parameters $u_e$, $e = 1, \dots, M$. Those input parameters captured all the differences between the different experimental setups, i.e., drug treatments and mRNA expression levels of the cell lines. Hence, to simulate the whole dataset $D$ once, $M$ different initial value problems had to be solved.

To account for the fact that experimental data are noise-corrupted, we chose an additive Gaussian noise model with standard deviation $\sigma_{e,i}$ for experimental condition $e$ and measurement index $i$. For the large-scale application example, we used the same $\sigma_{e,i}$ all experiments, as no prior knowledge on the standard deviation was available.

$$\bar{y}_{e,i} = y_i(\theta, u_e) + \varepsilon_{e,i}, \quad \text{with } \varepsilon_{e,i} \sim \mathcal{N}(0, \sigma_{e,i}^2) \quad (5)$$

A more detailed explanation of ODE modelling, in general, is given in the Supplementary Information.

### Parameter optimization.

This statistical observation model allowed us to compute the likelihood of an observed value $y(x(t, \theta, u), \theta)$ given a parameter vector $\theta$, assuming independence of the measurement noise terms[23]. Due to its better numerical properties, we took its negative logarithm, which yielded:

$$\bar{J}(\theta) = \frac{1}{2}\sum_{e=1}^{M}\sum_{i=1}^{N_e}\left(\frac{(\bar{y}_{e,i} - y_i(\theta, u_e))^2}{\sigma_{e,i}^2} + \log\left(2\pi\sigma_{e,i}^2\right)\right) = J(\theta) + \text{const.} \quad (6)$$

Assuming fixed measurement noise, the logarithmic term was just a constant offset. By neglecting it and identifying $y_i(\theta, u_e)$ with $y_{e,i}(\theta)$, we arrived at the objective or

cost function $J(\theta)$, which was given in Eq. (2).

$$J(\theta) = \frac{1}{2}\sum_{e=1}^{M}\sum_{i=1}^{N_e}\frac{(\bar{y}_{e,i} - y_{e,i}(\theta))^2}{\sigma_{e,i}^2} \quad (2)$$

For global optimization of $\theta$, we restricted the feasible parameter space to a region $\Omega = [10^{-5}, 10^3]^{n_\theta}$, which was assumed to be biologically plausible. Parameters were transformed and optimized on a logarithmic scale, i.e., in the box $\Omega' = [-5, 3]^{n_\theta}$. We used multi-start local optimization, i.e., we randomly sampled many parameter vectors, from which we initialized local optimizations. This approach has repeatedly been shown to be among the most competitive methods[23,24], if high-performing local optimization methods with accurate gradient information of the objective function are used. In order to compute accurate gradients, we employed adjoint sensitivity analysis (see ref. [70] for a review on the method), which is currently the most scalable method for gradient computation of high-dimensional ODE systems (see refs. [49,71] for method comparisons with focuses on systems biology and[49] for the implementation that was used).

### Full-batch optimization.

To benchmark our local optimization methods, we used the interior-point optimizer Ipopt[43], which combines a limited-memory BFGS scheme with a line-search approach[72] and solves the linear system in the inner problem via the solver package COINHSL[73] (see the section on implementation for the precise versions that were used). In previous studies[14], such interior-point optimizers have shown to be among the most competitive methods for local optimization of large-scale ODE systems[14,24].

More information on the formulation of the (log-)likelihood function and global parameter optimization of ODE models in a more general context is given in the Supplementary Information.

### Mini-batch optimization algorithms.

Mini-batch optimization is a particular type of local optimization, which exploits the sum structure of the objective function[38]. In our case, we could rewrite the objective function in the following form:

$$J(\theta) = \sum_{e=1}^{M}\underbrace{\frac{1}{2}\sum_{i=1}^{N_e}\left(\frac{\bar{y}_{e,i} - y_i(\theta, u_e)}{\sigma_{e,i}}\right)^2}_{= J_e(\theta)} = \sum_{e=1}^{M}J_e(\theta) \quad (7)$$

In the beginning of each epoch, the dataset was randomly shuffled and then divided into mini-batches, which are random subsets of the same size $S \subseteq \{1, \dots, n_e\}$. Hence, no data point was used redundantly within one epoch. In each optimization step $r$, only the contribution to the gradient, sometimes also called gradient estimate[37], based on the mini-batch $S_r$ was used. The exact way how a parameter update was executed, i.e., how $\theta^{(r+1)}$ was computed from $\theta^{(r)}$ and the gradient estimate $\sum_{e \in S_r} \nabla_\theta J_e(\theta^{(r)})$, was dependent on the chosen algorithm. We investigated the following common mini-batch optimization algorithms in our study (see Supplementary Information for more details and[37,74] for a more comprehensive summary of mini-batch optimization algorithms, as well as[44] for a detailed comparison in the context of Deep Learning):

- Stochastic gradient descent (SGD)[38], which is the simplest possible algorithm, using only the negative gradient of the objective function as an update direction (Supplementary Information, Algorithm 1).
- Stochastic gradient descent with momentum[31,45], a common variant, which uses a decaying average of negative gradients as direction instead of the negative objective function gradient alone (Supplementary Information, Algorithm 2).
- RMSProp[46], a so-called adaptive algorithm, which rescales/preconditions the current gradient by a decaying average over root-mean-squares of the previous objective function gradients (Supplementary Information, Algorithm 3).
- Adam[47], another adaptive algorithm, attempts to combine the benefits of RMSProp with the momentum approach by using two decaying averages (Supplementary Information, Algorithm 4).

For Adam, we tested two different settings: as the two decaying averages in the algorithm are controlled by two tuning parameters $\rho_1$ and $\rho_2$, we set them first—according to the original publication—to 0.9 and 0.999, respectively, and then, based on some non-exhaustive testing, both to 0.9. We denoted the first setting as Adam (standard), the second as Adam (balanced).

### Learning rates and optimizer step-sizes.

All the considered mini-batch algorithms rescale the computed parameter update with a factor called learning rate $\eta$, which can either be fixed over the optimization process, prescheduled, or adapted according to the optimization process. In our study, we tested—based on the literature[37,74] and our experience with local optimization—in total four learning rate schedules, which refer to the following numerical values for the small- to medium-scale models:

- Schedule 1: High learning rate, logarithmically decreasing from $10^0$ to $10^{-3}$.

- Schedule 2: Medium learning rate, logarithmically decreasing from $10^{-1}$ to $10^{-4}$.
- Schedule 3: Low learning rate, logarithmically decreasing from $10^{-2}$ to $10^{-5}$.
- Schedule 4: Constant learning rate, fixed to the value $10^{-3}$.

Assuming a given algorithm in optimization step $r$ produced a parameter update $\delta_r$, then the next proposed parameter vector would be

$$\theta^{(r+1)} = \theta^{(r)} + \eta_r \cdot \delta_r, \tag{8}$$

with $\eta_r$ being the learning rate at step $r$. Obviously, the learning rate influences the step size of the optimization algorithm in parameter space. However, many of the algorithms we investigated yielded parameter updates with $||\delta_r|| \neq 1$, thus $\delta_r$ does not only provide the direction of the parameter update but may also contribute to the step size. For, e.g., Adam, we obtained step sizes scaling with $\sqrt{n_\theta}$, with $n_\theta$ being the dimension of the unknown parameter vector (see Supplementary Information for more details and the corresponding calculation). When transferring the results of our study from the small- and medium-scale models to the large-scale model, we tried to conserve the actual step-sizes of the optimizers rather than the learning rates themselves, assuming the step-sizes to be the more fundamental quantities. On the large-scale model, we used two learning rate schedules with the following names and values:

- High learning rate, logarithmically decreasing from $10^{-1}$ to $10^{-4}$.
- Low learning rate, logarithmically decreasing from $10^{-2}$ to $10^{-4}$.

**Rescue interceptor and line-search for mini-batch optimization**. The rescue interceptor was implemented to mimic the behavior of a one-dimensional trust-region algorithm (Fig. 8). It works similar to an iterative backtracking line-search algorithm, which also reduces the step-length of the subsequent optimization steps, performing at most ten iterations. It is triggered if the objective function and its gradient cannot be evaluated. In this case, it keeps the current mini-batch, but undoes the previous parameter update, and reduces the current step length. In the next steps, the step length is gradually increased again until it reaches its original, unmodified value. If ten repetitions of this procedure are not sufficient to restore the optimization process, the corresponding local optimization run is stopped. More details on the method and its pseudo-code are given in the Supplementary Information (Algorithm 5 for the pseudo-code).

An additional line-search was implemented according to the interpolation method, described in Chapter 3 of ref. [50], and limited to at most three iterations. In each optimization step, the objective function value is checked on the same mini-batch after the parameter update. The parameter update is accepted if the objective function decreases, otherwise, the step size is reduced and the update repeated (Fig. 7). More details on the implementation and its pseudo-code are given in the Supplementary Information (Algorithm 6 for the pseudo-code).

**Computation of final objective function and correlation values**. To ensure an unbiased comparison of objective function values, we computed the final objective function and correlation values after optimization for all methods (full-batch and mini-batch) on the whole dataset. As we performed this comparison also on a set of independent test data and since the model was using scaling parameters to match the model output to the measurement data[23], we computed those scaling parameters analytically[14,75].

**In silico gene knockouts**. When working with the gene knockout data, we used all 18 cell lines, which were available in the dataset of ref. [53] and in our training data. The model from the large-scale application contains a total of 107 implemented genes and mutated forms of those. For an *in silico* knockout, we assumed the gene expression to be zero for the knocked-out gene. Furthermore, we considered a gene knockout to also affect all mutated variants. The change in cell viability was always computed as fold-change with respect to the untreated cell line.

**Threshold-dependent convergence criteria**. To assess the quality of parameter optimization, we investigated (beyond the final objective function and correlation values)

1. the computation time until convergence was reached for the first time
2. the number of converged starts per computation time

Both criteria are common metrics for assessing optimization performance[24]. Since the full-batch optimizer Ipopt was used as a benchmark throughout the study, we also fixed the convergence criterion based on optimization results from Ipopt: based on its ten best optimization results, we defined a value-to-reach as the mean plus one standard deviation over these ten final objective function values.

**Computation of receiver-operating-characteristics and classification accuracy**. When working with the drug response data, we used 13,000 of the 16,308 data points from 233 cell lines for model training and 3308 of the 16,308 data

points from 59 cell lines as a test set. However, only for 198 cell lines from the training set and 49 cell lines from the test set, all treatment conditions were available. In order to compute unbiased receiver-operating characteristics (ROCs), we used only those cell lines, yielding 11,088 data points for the training set and 2744 data points for the test set. Experimental conditions were grouped into two groups: Those, in which cell viability was reduced by more than 50% when compared to the untreated condition, were defined as responsive, the rest as non-responsive. We then computed classification thresholds to be those model output values, which corresponded to the points on the ROC being tangential to an affine function with slope $1 \, \widehat{=} \, 45°$. The inferred classification thresholds for model simulation were then used for classification on the independent test data.

Receiver-operating characteristics for the gene knockout data were computed analogously, by assuming a gene to be essential for a specific cell line if its knockout led to a viability reduction of more than 50%, according to the data from ref. [53]. Hence, the ROCs allowed us to compute classification threshold analogous to the ones obtained from drug response data. Prediction quality was assessed by the predicted essentiality of each gene for the 18 cell lines.

**Generation of the parameter ensemble from mini-batch optimization history**. The generation of the large parameter ensemble from the mini-batch optimization run with step size 10 was done in four steps: First, a smoothed history of the objective function value was computed, by averaging over up to one epoch (half an epoch prior and past the current point in optimization, when possible). Second, a cutoff value was determined, by the RMSE value of the best-found optimization result plus a tolerance of 10%, yielding a value which was still better than the best found by full-batch optimization. Third, all parameter vectors with smoothed objective function values better than this threshold were added to the ensemble, yielding parameter vectors from 52 optimization runs in total. Forth, this ensemble was thinned by a factor of 26, i.e., allowing at most 50 parameter vectors per epoch to be included in the ensemble. This procedure resulted in an ensemble with 8450 parameter vectors, which is about twice as much as the number of model parameters.

**Implementation of parameter estimation using the toolboxes AMICI and parPE**. Parameter estimation was performed using the parPE C++ library[14], which

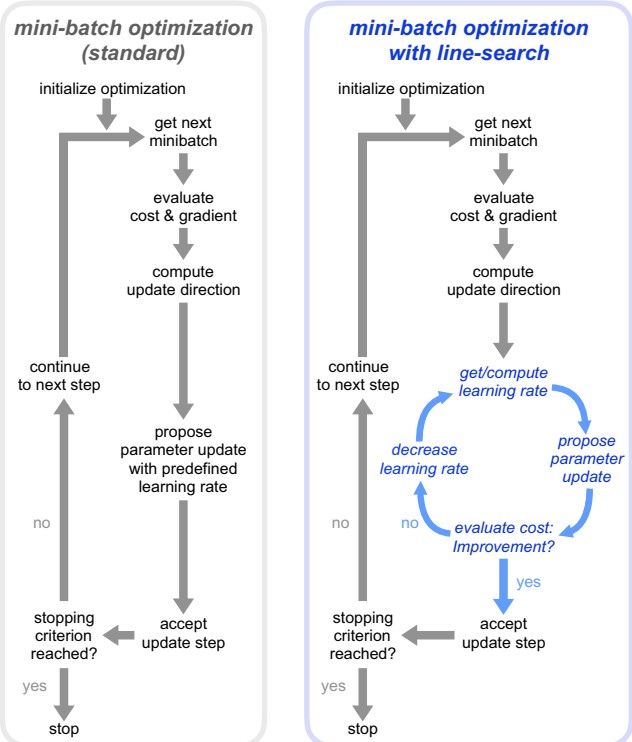

**Fig. 8 Comparison of standard mini-batch optimization and mini-batch optimization with line-search.** Left panel: Standard mini-batch optimization uses a prescheduled learning rate, which determines the step size during optimization regardless of whether an optimization step leads to an improvement or not. Right panel: If line-search is enabled, the objective function is reevaluated on the same mini-batch and checked for improvement. If no improvement is achieved, the learning rate is reduced until either improvement is achieved or until the maximum number of line-search steps is reached.

provides the means for parallelized objective function evaluation and optimization of ODE models generated by the AMICI ODE-solver toolbox[76]. parPE is specifically designed for computing clusters, employing dynamic scheduling across multiple compute nodes via MPI, and uses optimizers such as Ipopt[43]. In our studies, we used Ipopt version 3.12.9 in combination with COINHSL[73], running with linear solver ma27 and L-BFGS approximation of the Hessian matrix and extended parPE with the mini-batch algorithms described above.

For numerical integration of the ODEs, we used AMICI[76], which provides a high-level interface to the CVODES solver[77] from the SUNDIALS package[48] and generates model-specific C++ code for model evaluation and likelihood computation to ensure computational efficiency. In our applications, we used AMICI default settings with adjoint sensitivity analysis, employing a backward differential formula (BDF) scheme of variable order, with adaptive time stepping and error tolerances of $10^{-8}$ for the relative and $10^{-16}$ for the absolute integration error per step, allowing at most $10^4$ integration steps.

Optimizations were run on the SuperMUC phase 2 and SuperMUC-NG supercomputers (Leibniz Supercomputing Centre, Garching, Germany). Compute nodes were equipped with two Haswell Xeon Processor E5-2697 v3 CPUs (28 cores per node) and 64GB of RAM. For the large-scale model, mini-batch optimization multi-starts comprising 100 local optimizations were run on 65 nodes (1820 cores) with a wall-time limit of 48 h. The 20 local optimizations of Ipopt were separated into single runs with 12 nodes (336 cores), and 35 h of wall-time were granted. For the small to medium-scale models, each of the multi-starts comprising 100 local optimizations was run on 1, 2, and 3 nodes for the Fujita, Bachmann, and the Lucarelli model, respectively, always exploiting all 28 cores per node. Wall-times were fixed to 15, 32, and 40 h, respectively. Computations for the refined model were carried out on SuperMUC-NG (Leibniz Supercomputing Centre, Garching, Germany). Compute nodes were equipped with Intel Skylake Xeon Platinum 8174 CPUs (48 cores per node) and 96GB of RAM.

**Adaptation of benchmark models and creation of artificial data**. The small- to medium-scale examples for the benchmark study were chosen based on a collection of benchmark models[25]. We chose models with different system sizes, which allowed the generation of large artificial datasets that were sufficiently heterogeneous. This should ensure clear differences in objective function values and gradients when different mini-batches were used. In order to allow the creation of heterogeneous datasets, the SBML files and input parameters were slightly altered. The precise model versions are made freely available in the SBML/PEtab[78,79] format at Zenodo, under 10.5281/zenodo.4949641[80].

Artificial data was created by simulating the models with the parameter vectors reported in ref. [25]. Additive Gaussian noise was added to the model simulations, using the noise levels that were reported in ref. [25] for each observable.

**Data analysis and visualization**. Plots were created with Matplotlib, version 3.3.2, boxplots were drawn using boxplot from matplotlib.pyplot, with the following settings: whiskers show the whole span of the data, boxes show the range from the 25th to the 75th percentile, medians are highlighted as bold lines.

**Reporting Summary**. Further information on research design is available in the Nature Research Reporting Summary linked to this article.

## Data availability

Source data are provided with this paper. The models and data used in (and produced by) this study are available in the Zenodo database under accession code 4949641 (https://doi.org/10.5281/zenodo.4949641)[80]. This includes model implementations in SBML format[78], as well as implementations of the parameter estimation problems for all models in PEtab format[79]. The SBML implementation of the large-scale model is additionally available on GitHub (https://github.com/ICB-DCM/CS_Signalling_ERBB_RAS_AKT). The Zenodo archive also includes artificial data that was created and used for a benchmark study, and condensed results of the parameter estimation in HDF5 format. The data used for model training were taken from the Cancer Cell Line Encyclopedia (https://sites.broadinstitute.org/ccle/) and had previously been published by Barretina et al. in 2012[19]. The data used for *in silico* gene knockout validation were taken from the DepMap database (https://score.depmap.sanger.ac.uk) and had previously been published by Behan et al. in 2019[53]. Source data are provided with this paper.

## Code availability

Open-source software packages were used and further developed for this study. Parallelized parameter estimation was performed with parPE, and we implemented our mini-batch algorithms there directly. The AMICI package was used for the simulation of ODE systems, as well as log-likelihood and gradient computation with adjoint sensitivity analysis. The following open-source software packages were also used: the Python package for PEtab, the parameter estimation package pyPESTO, the scientific computing package SciPy, and the dimension reduction package umap-learn. The optimization package Ipopt was used with the linear solver package CoinHSL[73], which is free for academic use.

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

## Acknowledgements

This work was supported by the European Union's Horizon 2020 research and inno-vation program under grant agreement no. 686282 (B.M.H.L., C.W., D.W., J.H., L.S., S.M., and P.S.), CanPathPro. Computer resources for this project have been provided by the Gauss Centre for Supercomputing/Leibniz Supercomputing Centre under the grants pr62li and pn72go. Furthermore, this work was funded by the Deutsche For-schungsgemeinschaft (DFG, German Research Foundation) under Germany's Excellence Strategy—EXC 2151—390873048 (J.H.), MoKoCo19 (BMBF-DLR, grant no. 01KI20271) (D.P.), and the Federal Ministry of Economic Affairs and Energy (Grant no. 16KN074236) (D.P.).

## Author contributions

D.W. designed and implemented the parallelized optimization framework. P.S. imple-mented the mini-batch optimization algorithms, designed and implemented their algo-rithmic improvements. C.W. and B.M.H.L. retrieved and processed the data for the large-scale application example. J.H. and P.S. conceived the computational studies for mini-

batch optimization, P.S., L.S., J.H., and D.W. conceived the studies for full-batch optimization and the in silico knockout study. P.S. and L.S. performed the studies, P.S., L.S., and S.M. performed the validation studies. P.S. and D.P. performed the uncertainty analysis. P.S., L.S., J.H., and D.W. analyzed the results. All authors wrote and approved the final manuscript.

## Competing interests

The authors B.M.H.L. and C.W. are employees of Alacris Theranostics GmbH. The company did not, however, influence the interpretation of the data, or the data reported, or financially profit from the publication of the results. The remaining authors declare no competing interests.
