## [Peer Review File · Nature Communications]

Mini-batch optimization enables training of ODE models on large-scale datasetsReviewers' comments:

Reviewer #1 (Remarks to the Author):

This report from Stapor et al. is about an evaluation of selected "mini-batch optimization" methods for use in biological ODE modeling. The classic method in this class is called "stochastic gradient descent" or SGD, which is a very old method - the authors cite a 1951 paper. SGD fairly recently rose to prominence in the machine learning field. The reason is that this method provides a means for reducing the computational cost of objective function evaluation in optimization by using only a fraction of the available data for training in each function evaluation step. In many machine learning applications, this is important, because the dataset sizes being used for training are "large." The report at hand is interesting IMHO because SGD and related methods have, to date, not been used in biological (ODE) modeling, and the authors provide an example where mini-batch optimization offers a notable benefit (an order-of-magnitude reduction or so in the computational cost of optimization/curve fitting). A known benefit of SGD, which doesn't seem to be mentioned in the manuscript, is that SGD is less likely than, say, gradient descent, to become mired down near saddle points. This is because the gradient is calculated only approximately, which helps in escaping from/avoiding convergence to saddle points. For details, see the following preprint from Michael Jordan and colleagues:
<https://arxiv.org/pdf/1902.04811.pdf>

To the editor: I think the main point to ponder when evaluating the potential impact of this manuscript is future opportunities to profitably apply mini-batch optimization in biological ODE modeling. Modelers haven't used this method much if at all very probably because they typically don't have all that much data to use for model parameterization. Here, the authors provide an example involving a large amount of training data where mini-batch optimization is helpful. So... if you believe that large datasets are going to become more commonly available for parameterizing ODE models, this report is likely to be impactful. I would expect gradual increasing use of larger-than-normal datasets in the future, and thus, I'm skeptical that the manuscript will have an immediate high impact, but as noted above, I find the manuscript interesting (for the reasons stated above).

Somewhat disappointingly, the example fitting problem solved using mini-batch optimization in the study being reported didn't seem to generate biological insights, or those insights are not being reported here. The manuscript is purely about evaluation of selected mini-batch optimization methods and the example model/fitting problem is considered as a demonstration, not evidently as part of a modeling study aimed at generating insights/interesting testable predictions.

The rest of my comments are all minor, and can presumably be addressed by the authors with minor revisions.

The authors may want to cite this recent paper from Clemens Kreutz: "Guidelines for benchmarking of optimization-based approaches for fitting mathematical models" in *Genome Biology*.

John Tyson and co-workers have a long history of using viability data to parameterize ODE models. The authors may want to cite some of this work along with their paper (Ref. 14) in which they also used viability data for ODE model parameterization.

In making their case that large models are becoming more important, the authors may want to cite the recent paper from Marc Birtwistle and co-workers: "A mechanistic pan-cancer pathway model..." in *PLOS Computational Biology*.

The wording in the Introduction about numerical integration of stiff ODEs suggests that the authors are doing something innovative. I would suggest a revision of the wording to better indicate that they are just using standard methods, developed by others long ago.

Are there applications of SGD and related methods to ODE model parameterization in fields outside biology? I would think so. If so, perhaps this work should be discussed in the Introduction and/or Discussion.

With the introduction of Eq. (1) in Results, the authors use the term "experimental condition." It could

be helpful if the authors provided a definition of "condition" here.

The authors apparently used computer clusters, but it's not discussed how this helped in the main text. To what degree are the methods being evaluated capable of leveraging parallel computing resources? If the algorithms are parallelized, are they synchronous or asynchronous?

In Table 1 itself, it should be made clear that the data points listed are all synthetic data points.

In Figure 2, the authors use relative rankings vs. an absolute measure of algorithm/method performance. I would much rather see the latter. The protocols being considered in this figure should probably be given easy-to-reference names, such as Protocol 1. The protocols are referred to using long (somewhat confusing) descriptive phrases. The possibility for confusion is illustrated best when the authors use the phrase "the highest, i.e., the medium".

The authors compare mini-batch optimization methods against Ipopt, which is a commonly used software package for constrained optimization. Without constraints, this package isn't really necessary. Why do the authors not consider comparison against standard unconstrained optimization methods? An answer to this question should be given.

For adjoint sensitivity analysis, the authors cite one of their recent papers. This method (from the 50's) is classical and widely used in diverse fields, and perhaps citation of other papers would be appropriate, to give readers unfamiliar with the method a better indication of its long history.

SGD is referred to as "vanilla SGD." The method's accepted name is just SGD.

The authors refer to a variation of Adam as "tuned." This descriptor is not very descriptive. Better descriptors might be "balanced" or "with equal decay rates."

Could the authors comment on why the Fujita model is having problems with successful numerical integration?

Confusingly, the authors discuss two distinct methodological innovations that both involve use of line search. These methods should be given more distinctive names. It was difficult to figure out which of the two methods the authors were discussing at points in the text.

It might be helpful to replace "13,000 data points" with "13,000 of the 16,308 data points" and to replace "3,308 data points" with "3,308 of the 16,308 data points."

The ROC analysis reported in Figure 4 could be briefly described in the main text for greater clarity.

One question that arose in my mind is the following. The authors are using an ODE model to make binary classification decisions. How does the model perform relative to a more standard machine learning approach in which a classifier is trained based on the available features?

Could the authors summarize their recommendations in the form of a flow chart? I'd appreciate a clearer statement of the bottom-line recommendations.

The authors remark that their early study (Ref. 14) suffered from overfitting in the first paragraph of the Discussion section. Could the authors elaborate to better explain what problem is being solved with the new better mini-batch optimization approach? At present, a reader would have to consult Ref. 14 to fully grasp the point being made.

A claim is made that early stopping avoids overfitting? This is not obvious to me. Could the authors explain their thinking a little bit better?

Is choosing the same standard deviation for all data points (in the discussion of Eq. (4)) the same as just omitting the standard deviation weighting from the optimization problem?

How was approach to steady state determined for the large-scale model simulations? Nothing is said in

this version of the manuscript about that.

Just below Eq. (6), it is said that mini batch sizes are all the same. This doesn't seem possible except for special cases. Rather, they are all nearly the same?

The vector Δ_r is described as provided a direction, just above Eq. (7). I think the stepsize is given by the product of Δ_r and the scalar learning rate η_r , so... Δ_r is providing not just a direction but also contributing to the stepsize?

What happens in the optimization procedure if rescue fails (say, on the first step)?

The authors selected a handful of mini-batch optimization methods to evaluate. In the Discussion, it might be helpful to briefly acknowledge some of the other methods in this class that were not evaluated.

Reviewer #2 (Remarks to the Author):

The manuscript by Stapor et al 'Mini-batch optimization enables training of ODE models on large-scale datasets' presents an application of mini-batch based learning of parameters of large ODE-based models of chemical networks. The study is based on the previously published model and the fast learning methodology was also previously published. However, the current study increases the size of the dataset on which the training has been performed which is possible due to learning optimization thanks to the application of mini-batch trick.

The manuscript is well written and technically correct as far as one can judge because the description is rather technical and specific and requires the good knowledge of the used industrial scale optimizers and machine learning terminology. I am convinced that using mini-batches is an excellent idea to further scale up the learning of ODE parameters in large models.

However, I am not sure that the message of the manuscript represents a sufficient breakthrough with respect to already published work and that it can be interesting to a wide readership. The model and the accelerated parameter fitting formalism has been published by the authors of this manuscript before (PLoS Comp Biol, 2017; Cell Systems, 2018) which was a quite remarkable achievement. Mini-batch approach and its application in the current study also looks quite known and widely exploited idea even if in a different field (AI).

I do not feel that the current manuscript promises a similar advance, representing rather an important technical improvement to already established framework. The scale of the used dataset is indeed impressive. However, it remains unclear how relevant are the resulting predictions, since there is no real validation part in the manuscript besides the standard machine-learning based statistical validation. The correlation plot and the ROC curves in Figure 4 look quite promising but it is difficult to judge the actual value of the fit performed under so many conditions merged together. In order to prove the value of the undertaken large scale parameter fitting exercise, I would generally expect more insights into what is the level of biological relevance of the trained model predictions. I would also like to see a clear demonstration that the reported predictive power is not of artefactual nature (such as absence of artefacts similar to the Simpson's paradox resulting from merging many

heterogeneous sources of data together). I would also expect placing the current study in the context of other mathematical modeling studies aimed at exploiting the same datasets in order to unravel the mechanism of drugs action.

In other words, I feel that the focus of the manuscript should be completely changed in order to make the paper interesting for the wide computational biology community. In the current form, it better fits to a journal specialized on machine learning and optimization. Therefore, I can not recommend this manuscript for publication in Nature Communications.

Reviewer #3 (Remarks to the Author):

The authors present a novel approach for the calibration of large-scale models by means of mini-batch optimization, typically used in machine learning approach.

The method proposed by the authors is promising, as shown by the application to the large scale model of cancer signaling.

I have some comments that I hope will help to improve the manuscript, making it suitable for publication on Nature Communication journal.

Major comments:

- I believe that the computational complexity and the increase in running time in the case of the application of the rescue functionality should be discussed by the authors. To this regard, I believe that the remark reported at lines 187-189 should be extended and better clarified.
- I do not understand why the results concerning high learning rate values are only presented in Fig 3C considering only Adam optimization algorithm applied to Lucarelli model. I believe that it would be interesting to have in the paper the results obtained with different algorithms on different models (as shown in Figure 2B).
- Considering the results presented in Figure 4, I would like to see some additional comments about the correlation value achieved (0.76-0.74) and speculations on possible strategies to improve such results.
- Comparing the results on Figure 5A and B (top panels), I missed the reason why there is a remarkable difference in the final cost function values obtained with training and test data. Please explain.
- I am skeptical about the conclusion of the authors concerning the choice of hyperparameters, especially the learning rate. It seems to me that this value is highly problem dependent. The authors should extend the discussion about this aspect and provide some heuristics to choose an appropriate value for this hyperparameter (if any).
- I believe that the authors should show the results of a sensitivity analysis on the parameters of the ODE model of cancer signaling, to highlight the importance of the obtained results and to better explain the results of the parameter estimation.
- To make the computational experiment reproducible, the authors should provide the parameter space limits used in their tests.
- Finally, a discussion about alternative methods to accelerate the parameter estimation task, such as those exploiting GPUs, should be included in the manuscript. The authors might consider this paper: <https://academic.oup.com/bib/article/18/5/870/2562773>

Minor comments:

- replace "classical" with "classic" throughout the manuscript
- I believe Figure 1 should be better explain, especially panel B. Moreover, Figure 1B is never referred to in the text
- lines 172, 276: replace "previous study" with "previous batch of tests"
- I suggest to replot the histogram of Fig 3B, since columns representing values close to zero are not visible
- line 211: replace "dimension" with "size"
- there is an inconsistency between Figure 4D E G H and the corresponding explanation in the text (AUC vs AUROC). Moreover, I suggest to add labels to Figure 4E

Response to Reviewer 1

This report from Stapor et al. is about an evaluation of selected "mini-batch optimization" methods for use in biological ODE modeling. The classic method in this class is called "stochastic gradient descent" or SGD, which is a very old method - the authors cite a 1951 paper. SGD fairly recently rose to prominence in the machine learning field. The reason is that this method provides a means for reducing the computational cost of objective function evaluation in optimization by using only a fraction of the available data for training in each function evaluation step. In many machine learning applications, this is important, because the dataset sizes being used for training are "large." The report at hand is interesting IMHO because SGD and related methods have, to date, not been used in biological (ODE) modeling, and the authors provide an example where mini-batch optimization offers a notable benefit (an order-of-magnitude reduction or so in the computational cost of optimization/curve fitting). A known benefit of SGD, which doesn't seem to be mentioned in the manuscript, is that SGD is less likely than, say, gradient descent, to become mired down near saddle points. This is because the gradient is calculated only approximately, which helps in escaping from/avoiding convergence to saddle points. For details, see the following preprint from Michael Jordan and colleagues: <https://arxiv.org/pdf/1902.04811.pdf>

We thank the reviewer for this positive evaluation. We did not mention the benefit of being less prone to converging towards saddle points, because this is very hard to verify for our applications in practice, unlike the computational speed up or the better optimization results we obtained. Furthermore, also state-of-the-art full-batch optimizers such as Ipopt should – at least in theory – not converge towards saddle points. For example, Ipopt uses a limited memory BFGS approach for choosing the step direction. However, to which degree this really avoids convergence towards saddle points is again hard to verify in practice, as computing the exact Hessian matrix at the final point of the optimization trajectory is computationally prohibitive. However, we decided to include the reference pointed out by the reviewer in our revised manuscript with a short remark.

Changes in the manuscript:

Initial submission:

Hence, the model is only evaluated on a fraction of the dataset per optimization step, which leads to a drastic reduction of computation time (21; 79).

Revised submission:

Hence, the model is only evaluated on a fraction of the dataset per optimization step, which leads to a drastic reduction of computation time (21; 79), **and may help to avoid convergence towards saddle-points during optimization (11).**

Somewhat disappointingly, the example fitting problem solved using mini-batch optimization in the study being reported didn't seem to generate biological insights, or those insights are not being reported here. The manuscript is purely about evaluation of selected mini-batch optimization methods and the example model/fitting problem is considered as a demonstration, not evidently as part of a modeling study aimed at generating insights/interesting testable predictions.

We agree that our original manuscript did not yield biological insights, as the main point of our manuscript focused on presenting the implementation of a novel method, rather than investigating our application model or its prediction in detail. For this reason, we revised our manuscript substantially: We now show that the method does not only speed up computations,

and improve the obtained optimization results, but added substantially more model validations: On one hand, we show the drug response predictions in much more detail, on the other hand, we perform in-silico gene knockouts, which we compare to experimental data from (5)). These validations indeed generated new insights about the biological relevance of certain (missing) parts of the large-scale ODE model, which are shown in the main manuscript and discussed in detail in the supplementary information. Based on these results, we even perform a full iteration in the classic modeling cycle, i.e., model training, validation of model predictions, model refinement, model re-training, and re-evaluation of model predictions. To the best of our knowledge, this has never been carried out for an ODE model of the considered size here, since such an iterative model refinement is usually much too expensive in terms of computation time for the models of this size. Beyond this, we added a part on uncertainty analysis of our trained model by using ensemble methods. Creating an ensemble with more parameter vectors than free parameters in the model has so far not yet been possible for such models. We hope that the reviewer agrees, that these additional validations and analyses underline the usefulness of our method for practical applications.

Changes in the manuscript (added validation of drug response):

Initial submission:

Based on the characteristics, we computed classification thresholds for the simulation, when a cell-line together with a treatment condition is to be classified as responsive and repeated the computations for each considered drug individually. On the training data, the ensemble model achieved a classification accuracy of 86% (Fig 4e).

Revised submission:

Based on the characteristics, we computed classification thresholds for the simulation, when a cell-line together with a treatment condition is to be classified as responsive and repeated the computations for each considered drug individually. On the training data, the ensemble model achieved a classification accuracy of 86% (Fig 4e). Beyond the correlation and ROC analysis, we also analyzed the model fits to drug response data. The trained model was able to describe the response to different drug treatments and also to capture the varying behavior between the 233 cell lines correctly (Fig. 4f and Supplementary Information).

In a next step, we validated predictions of the model on cell lines from the independent test set, where we still found a Pearson correlation between the data and the simulation of the ensemble model of 0.74 (Fig 5a). The ROC analysis for the classification into responding and non-responding treatments on the test set yielded an AUC value of 0.94 for the ensemble model and values between 0.90 and 0.92 for the ten best local optimizations (Fig 5b), while the classification accuracy was around 85% (Fig 5c). Importantly, the model did not only classify trivial cases correctly, but was also able to capture the variability between cell lines and drugs (Fig 5d).

Changes in the manuscript:

Added text on validation via in-silico knockouts:

Mini-batch optimization renders iterative model refinement possible for large-scale ODE models

Beyond predicting drug response, we also validated the model on experimental data from a recently published CRISPR screen: Behan et al. had analyzed the change in cell viability when knocking out various genes on a large set of cell lines (5). We found 107 genes and 18 cell lines in the knockout dataset, that were also included in our application example and the dataset used for model training, summing up to 1926 data points which we used for validation. Based on the gene knockout data, we classified a gene as "essential" for a cell line, if the knockout led to a viability reduction of more than 50% and compared this essentiality classification with in-silico knockouts from our application example (Fig 5e). We computed receiver-operator-characteristics (ROC) for knockout simulations based on our trained model, and on random parameters as reference. We obtained better-than-random predictions (AUC of 0.59) from the untrained model and, as expected, better predictions for the trained model (AUC of 0.63).

Yet, the classification threshold for model simulation chosen by the ROC analysis was surprisingly high, leading to many false positive predictions (Supplementary Fig. 10). As this indicated a shortcoming of the model, we added further proteins, which should contribute to the viability readout of the model (see Methods section for more details on how the viability readout is computed): First, we included the protein Checkpoint kinase 1, a major regulator of the cell cycle that is encoded by the gene CHEK1, as additional pro-proliferative readout. The gene CHEK1 was shown to be among the most essential ones for cell viability in the validation data, which was however not accounted for in the original model of (17). Secondly, we also included doubly phosphorylated MAP3K1 as pro-proliferative readout, which was so far only acting in a feedback loop with upstream proteins. This modification allowed the model to partly circumvent the downstream bottleneck of the MAPK-cascade, and hence made it possible to explain potential downstream effects of MAP3K1. After calibrating this refined model on the previously used drug response data, we obtained substantially improved predictions of gene essentiality (Fig 5e, AUC of 0.67). The classification threshold for the model readout was reduced, leading to substantially fewer false positive predictions (Fig 5f, Supplementary Fig. 11). A detailed analysis of the remaining false predictions is given in the Supplementary Information.

As confirmation of the refined model topology, we refitted the original model on the drug response and gene knockout data, i.e. 14,926 data points, simultaneously. As expected, calibrating the model towards the knockout data led to substantially improved classification results (AUC of 0.74, vs. 0.63 for the model trained on drug response data only). However, also when being refitted on the knockout data, the model was still not able to describe the data when knocking out CHEK1, confirming the limitations of the original viability readout (Supplementary Fig. 10). In a last step, we refitted also the improved model on drug response and knockout data simultaneously, which again drastically improved the classification accuracy (AUC of 0.84, vs. 0.67 for the model trained on drug response data only). This suggested that a more general ODE model of cell signaling should encode at least a basic version of the cell cycle, if the model is to explain viability. To the best of our knowledge, this is the first time that systems-biology modeling loop (see Supplementary Fig. 12), i.e., iterative model refinement and recalibration, has been carried out for an ODE model of this size.

Changes in the manuscript (Discussion of the in-silico knockouts):

Initial submission:

The trained model provided accurate predictions whether a certain treatment would reduce cell viability by more than 50% for a chosen cell-line in more than 85% of the cases, even on cell-lines which had not been used for model training. The stochasticity introduced by the mini-batching in combination with the large dataset seemed to have addressed the overfitting observed in previous studies (17) and ensemble modeling improved the prediction accuracy further. Overall, our implementation reduced computation time by more than one order of magnitude while providing better fitting results than established methods. The improved scaling characteristics should render problems with even larger datasets feasible.

Revised submission:

The trained model provided accurate predictions of whether a certain treatment would reduce cell viability by more than 50% for a chosen cell-line in more than 85% of the cases, even on cell-lines which had not been used for model training. Furthermore, we performed in-silico gene knockouts and validated them on a large experimental CRISPR screen from literature, which helped us to improve the model topology and to obtain well-matching knockout predictions. In a last step, we performed uncertainty analysis based on ensemble methods and assessed the exploration of the parameter space during optimization, where mini-batch optimization enabled results which had not been achievable with established optimization methods.

A closer analysis of the in-silico knockout study allowed us to pinpoint deficiencies of the large-scale cancer signaling model, e.g., concerning the implementation of cell cycle genes, and a subsequent model refinement yielded substantially improved model predictions. To the best of our knowledge, this is the first time that a full modeling cycle, including model refinement and re-training, has been performed for an ODE model of this size, as the cost of model calibration renders iterative refinement usually prohibitive for large ODE models. Yet, a set of incorrect predictions remained: In particular, the essentiality of genes that encode master kinases such as PDPK1 or phosphatases such as PPP2CA was underestimated. At the same time, the importance of some genes encoding proteins involved in the AKT and RAS signaling pathways was overestimated, especially those which either interact with the EGF receptor, such as SRC, VAV2, or EGFR itself, or those which acts as transcription factors, such as CREB1, ELK1, or STAT3. Hence, we assume that a more comprehensive model, trained on a larger dataset, would be needed to, e.g., reliably discover new drug targets or to reject possible drug candidates before entering a clinical trial. The datasets to train such a more comprehensive, whole-cell signaling model have been made available in the last years, and mini-batch optimization renders the calibration of such a model computationally feasible.

Changes in the manuscript:

Added text on uncertainty analysis:

Mini-batch optimization improves parameter space exploration and uncertainty analysis

In a last set of tests, we further analyzed the optimization result with the smallest batch-size, i.e., 10. For this purpose, we created a large parameter ensemble from the optimization history based on an objective function threshold, similar to the ideas of (76) (more details on the ensemble generation can be found in the Methods section). This ensemble contained 8,450 parameter vectors from 52 out of the 100 local optimization runs. We also generated a second, smaller ensemble using only the final results of the ten best optimization runs (Fig. 7a). In addition, we investigated the coverage of the feasible interval across the different local optimization runs for each model parameter at three different checkpoints during optimization (Fig. 7a). Comparing these parameter interval coverages from mini-batch optimization with those from full-batch optimization revealed that mini-batch optimization substantially increased coverage, and thus, yielded a better exploration of the parameter space (Fig. 7b).

To assess the uncertainty of the calibrated model parameters, we analyzed the parametric sensitivities of the model output on the ten best optimization results. A singular value decomposition of this sensitivity matrix showed that 3,029 directions in parameter space were numerically non-zero, indicating that the model could acquire information about more than 3,000 degrees of freedom. Yet, many directions in parameter space were poorly determined (Fig. 7c). Complementary to this sensitivity analysis, we analyzed the covariance structure of the large parameter ensemble. A principal component analysis (PCA) revealed that most of the total variance was spread across a low-dimensional subspace, showing a sharp drop in the explained variance after about 50 PCA directions, which coincides well with the number of local optimizations present in the parameter ensemble (Fig. 7d). This indicates that the parameter space was not yet sufficiently explored, although mini-batch optimization allowed to afford more local optimization runs than full-batch optimization. This was confirmed by a UMAP embedding of the parameter ensemble, which showed that the local optimization runs did not converge to a common optimum, but remained separated in parameter space (Supplementary Fig. 16). The spectrum of the covariance matrix indicated 49 directions with high variability, a second mode containing about 3,600 directions with lower, but clearly present variability (Fig. 7e), and a third mode of probably non-explored directions, which had a variability below floating point precision. The overall 3,600 explored PCA directions in this large ensemble are in line with the results from the sensitivity analysis (roughly 3,000), as the discrepancy between the two results can be explained with additional information being present in the larger ensemble.

Moreover, we revised Figure 4, and added a new Figure on model validation (Fig. 5 in the revised submission) and a new Figure on uncertainty analysis (Fig. 7 in the revised submission):

Changes in the manuscript:

Revised Figure 4:

See next page

Changes in the manuscript:

Added Figure 5:

See next page

Figure 4: Description of application example, datasets and model performance, when trained with mini-batch optimization. **a** Simplifying illustration of the multi-pathway model of cancer signaling. **b** Left: Overview of the datasets used for training and model validation, taken from the Cancer Cell Line Encyclopedia. Right: Comparison of model sizes and experimental conditions used for model training of recently published ODE models. **c** Correlation of measured simulated cell viability for all points of the training data, color-coding indicates density in scatter plot. **d** Receiver-operating-characteristics for classification into responsive and non-responsive combinations of cell-lines and treatments on training data for the best ten optimization runs (gray) and an ensemble simulation (blue). **e** Area under ROC and classification accuracy on training data for the ten best optimization results (gray), for the ensemble model (black), and for the ensemble model on data for each drug individually (colored). **f** Simulated drug response. Left: Ranking of fit quality for cell lines by average root-mean-square error (RMSE). Right: Two out of 233 cell lines from the training data, error bars indicate standard deviation across the ensemble, for a cell line which the model was able to describe well (blue, BCPAP) and a cell line, which was less well captured by the model (orange, KU812).

Figure 5: Validation of the fitted large-scale cancer model on independent test data. **a** Correlation of test data and model prediction. Color-coding indicates density in scatter plot. **b** ROC-curves for classification of drug responses of cell lines on the test set. Classification thresholds from the training data were used. **c** Area under ROC-curve and classification accuracy on test data for the ten best optimizations (gray), the ensemble model (black), and the ensemble model for each drug individually (colored). **d** Simulated drug response. Left: Ranking of fit quality for cell lines by average root-mean-square error. Right: Two out of 59 cell lines from the test data, error bars indicate standard deviation across the ensemble, for a cell line which the model was able to describe well (blue, 8505C) and a cell line, which was less well captured by the model (orange, JHH5). **e** ROC-curves for classification of gene essentiality. Measurement data for 18 cell lines was taken from Behan et al, 2019. In-silico knockouts are shown for the untrained model (blue), the trained model (orange), and the refined and trained model (green). **f** Measurement, prediction, and confusion matrix for essential genes for the refined and trained model. Numbers indicate how often a gene was found to be essential in experimental data and in-silico knockout predictions for the 18 cell lines, sums show the number of true and false predictions over essential genes.

Changes in the manuscript:

Added Figure 7:

See next page

The rest of my comments are all minor, and can presumably be addressed by the authors with minor revisions.

The authors may want to cite this recent paper from Clemens Kreutz: "Guidelines for benchmarking of optimization-based approaches for fitting mathematical models" in *Genome Biology*.

John Tyson and co-workers have a long history of using viability data to parameterize ODE models. The authors may want to cite some of this work along with their paper (Ref. 14) in which they also used viability data for ODE model parameterization.

In making their case that large models are becoming more important, the authors may want to cite the recent paper from Marc Birtwistle and co-workers: "A mechanistic pan-cancer pathway model..." in *PLOS Computational Biology*.

The three suggested references were added to our manuscript in the appropriate places. In the first case, we added also some explanatory text:

Changes in the manuscript:

Initial submission:

This is one of the main reasons why satisfactory parameter optimization for large-scale ODE models is still an open problem (31).

Revised submission:

This is one of the main reasons why satisfactory parameter optimization for large-scale ODE models is still an open problem (31), and why research on parameter estimation is of major importance (37).

The wording in the Introduction about numerical integration of stiff ODEs suggests that the authors are doing something innovative. I would suggest a revision of the wording to better indicate that they are just using standard methods, developed by others long ago.

Indeed, the wording in the Introduction might have been confusing. However, we wanted to put emphasis on this point, as many studies still disregard the importance of stiffness in ODE models. We added a reference which provides evidence for this stiffness and adapted the wording accordingly.

Changes in the manuscript:

Initial submission:

However, it is well-known that ODE models in systems biology typically exhibit stiff dynamics. This makes it necessary to employ advanced numerical integration methods, such as implicit solvers with adaptive time stepping (68). This implies that it is essential to combine advanced methods from both fields, deep learning and ODE modeling.

Revised submission:

However, it is well-known that ODE models in systems biology typically exhibit stiff dynamics. This makes it necessary to employ implicit solvers with adaptive time stepping (68). This implies that it is essential to combine advanced methods from both fields, deep learning and ODE modeling.

Figure 7: Analysis of parameter uncertainty based on an ensemble of parameter vectors from the the best mini-batch optimization, i.e., with batch size 10. **a** Generation of a large parameter ensemble (8,450 vectors) based on the history of multi-start local mini-batch optimization. A cutoff value was chosen and parameter vectors added to the ensemble if the objective function value was below the cutoff (red). A smaller ensemble was created from the ten best optimization results (yellow). At three checkpoints (blue), the coverage of parameter intervals was computed as a ratio of the interval covered by the optimization runs over the interval between the parameter bounds. **b** Computation of parameter interval coverage based on multi-start local optimization with mini-batch optimization and Ipopt (full-batch optimization). **c** Sensitivity analysis for model output (cell viability) w.r.t. the 4232 model parameters. Singular values of the sensitivity matrix, computed on the small parameter ensemble. **d** Principal component analysis of the large parameter ensemble created from mini-batch optimization, indicating the contribution (individual and summed) of the different PCA directions to the total observed variance in the ensemble. **e** Spectrum of the covariance matrix (for the 4232 model parameters) created from the large parameter ensemble from (d).

Changes in the manuscript:

Initial submission:

To the best of our knowledge, this is the first study integrating advanced training algorithms from deep learning with sophisticated tools from ODE modeling.

Revised submission:

To the best of our knowledge, this is the first study integrating advanced training algorithms from deep learning with established tools from ODE modeling.

Are there applications of SGD and related methods to ODE model parameterization in fields outside biology? I would think so. If so, perhaps this work should be discussed in the Introduction and/or Discussion.

We agree that it is possible that mini-batch optimization methods have been applied to ODE models in research contexts other than biology or machine learning. However, despite performing a literature research on the topic, we are not aware of any such applications. However, if the reviewer is aware of any other usage of mini-batch-like methods for ODE models in other fields of application, we would obviously be happy to reference and discuss those in our manuscript.

With the introduction of Eq. (1) in Results, the authors use the term "experimental condition." It could be helpful if the authors provided a definition of "condition" here.

As the term "experimental condition" is crucial to the understanding of the concept of mini-batching applied to ODE models, we followed the reviewer's remark and now give a more detailed explanation of its meaning not only in the Method section, but also in the main manuscript and put more emphasis on it.

Changes in the manuscript:

Initial submission:

Yet, if the data used to calibrate the model is derived from different experimental conditions, those constitute independent initial value problems, which need to be simulated repeatedly during model training (56). This causes a linear scaling of the computation time with the number of experimental conditions to be simulated. For large-scale ODE models ...

Revised submission:

Yet, if the data used to calibrate the model is derived from different perturbation experiments, these must be modeled as different initial value problems and simulated repeatedly during model training (56). We refer to these distinct initial value problems as "experimental conditions". Each of them is encoded by a different vector of input parameters to the ODE system. As they must be simulated separately, the computation time scales linearly with the number of experimental conditions during model training. For large-scale ODE models ...

Changes in the manuscript:

Initial submission:

The inference of model parameters θ from experimental data is based on reducing a distance measure between simulated model outputs and measurements. In practice, the distance metric is often based on the assumption that the measurement noise is normally distributed and independent for each data point. The corresponding negative log-likelihood function J , which serves as objective or cost function, is (up to a constant, more details in the Methods section and the Supplementary Information) given by the sum of weighted least squares:

$$J(\theta) = \frac{1}{2} \sum_{e=1}^M \sum_{i=1}^{N_e} \frac{(\bar{y}_{e,i} - y_{e,i}(\theta))^2}{\sigma_{e,i}^2} \quad (1)$$

Here, M denotes the number of different experimental conditions, N_e the number of measured data points for condition e , $\bar{y}_{e,i}$ are the measured data points, $y_{e,i}$ are the observables from the model simulation, and $\sigma_{e,i}$ denotes the standard deviation for data point $\bar{y}_{e,i}$. An experimental condition is the setting of a specific biological perturbation experiment, such as a stimulation with a drug and its dosage, but it can also comprise differences in the experiment which are due to working with different cell-lines. In the ODE model, these experimental settings are given by a vector of input parameters, which define the initial value problem and hence determine the time-evolution of the studied system. If the system has M experimental conditions, ...

Revised submission:

We assume the time evolution of the state variables x of the ODE system is given as

$$\dot{x} = f(t, x(t, \theta, u^e), \theta, u^e), \quad x(0) = x_0(\theta, u^e), \quad (2)$$

where θ are the unknown model parameters and u^e is a vector of known input parameters, which determine the simulated experiment indexed by e . Hence, u^e encodes an experimental condition, i.e., a distinct initial value problem. In an application context, this can encode a specific biological perturbation experiment, such as a stimulation with a drug or different cell-lines. The inference of model parameters θ from experimental data is based on reducing a distance measure between simulated model outputs and measurements. In practice, the distance metric is often based on the assumption that the measurement noise is normally distributed and independent for each data point. The corresponding negative log-likelihood function J , which serves as objective or cost function, is (up to a constant, more details in the Methods section and the Supplementary Information) given by the sum of weighted least squares:

$$J(\theta) = \frac{1}{2} \sum_{e=1}^M \sum_{i=1}^{N_e} \frac{(\bar{y}_{e,i} - y_{e,i}(\theta))^2}{\sigma_{e,i}^2} \quad (3)$$

Here, M denotes the number of different experimental conditions, N_e the number of measured data points for condition e , $\bar{y}_{e,i}$ are the measured data points, $y_{e,i}$ are the observables from the model simulation, and $\sigma_{e,i}$ denotes the standard deviation for data point $\bar{y}_{e,i}$. If the system has M experimental conditions, ...

The authors apparently used computer clusters, but it's not discussed how this helped in the main text. To what degree are the methods being evaluated capable of leveraging parallel computing resources? If the algorithms are parallelized, are they synchronous or asynchronous?

As stated in the main manuscript, we implemented our functionality in the parPE library and refer to its first publication (64). parPE is designed to be used with large computing clusters

and uses dynamic scheduling and asynchronous parallelization of local optimization runs across multiple compute nodes based on OpenMP and MPI. This way, we achieved a parallel efficiency of about 95% when using about 1,500 CPUs during optimization with Ipopt, for the large-scale cancer signaling model and 13,000 simulation conditions. This parallelization was crucial, as multi-start local optimizations with Ipopt took more than 70,000 CPU hours. Even assuming the best scenario, in which each of the 20 starts needs the same computation time, parallelization just over local optimization would have taken 3,500 hours. As Ipopt employed 150 optimization steps, each step took more than 23 hours. Even if all 13,000 independent simulations of each objective function evaluation had been parallelized with perfect efficiency across the 28 CPUs on one compute node, these 150 optimization steps would have led to a wall time of more than 120 hours. Dynamic scheduling and parallelization across multiple compute nodes allowed us to reduce this wall time to roughly 40 hours. Hence, the dynamic scheduling within parPE was necessary to produce the shown results and to enable a comparison of full-batch and mini-batch optimization. The explanatory text on parPE was extended in the Methods section. Interestingly, as mini-batch optimization performs fewer consecutive simulations than full-batch optimization, it would allow for less complex parallelization schemes, which is another benefit of the proposed methodology. However, as this aspect is of purely technical nature, we did not mention it in our manuscript.

Changes in the manuscript:

Initial submission:

Parameter estimation was performed using the parPE C++ library (64), which provides the means for parallelized objective function evaluation and optimization of ODE models generated by the AMICI ODE-solver toolbox (16) using optimizers such as Ipopt (77). In our studies, we used Ipopt version 3.12.9, running with linear solver ma27 and L-BFGS approximation of the Hessian matrix and extended parPE with the mini-batch algorithms described above.

Revised submission:

Parameter estimation was performed using the parPE C++ library (64), which provides the means for parallelized objective function evaluation and optimization of ODE models generated by the AMICI ODE-solver toolbox (18). **parPE is specifically designed for computing clusters, employing dynamic scheduling across multiple compute nodes via MPI and uses** optimizers such as Ipopt (77). In our studies, we used Ipopt version 3.12.9 **in combination with COINHSL (29)**, running with linear solver ma27 and L-BFGS approximation of the Hessian matrix and extended parPE with the mini-batch algorithms described above.

In Table 1 itself, it should be made clear that the data points listed are all synthetic data points.

Changes in the manuscript:

Initial submission:

Table 1 Overview of ODE models for benchmarking mini-batch optimization					
Model name	State variables	Parameters	Conditions	Data points	Reference
Fujita	9	19	600	6,000	(19)
Bachmann	25	40	1,200	12,000	(3)
Lucarelli	33	72	1,500	60,000	(44)

Revised submission:

Table 1 Overview of ODE models for benchmarking mini-batch optimization					
Model name	State variables	Parameters	Conditions	Data points (synthetic)	Reference
Fujita	9	19	600	6,000	(19)
Bachmann	25	40	1,200	12,000	(3)
Lucarelli	33	72	1,500	60,000	(44)

In Figure 2, the authors use relative rankings vs. an absolute measure of algorithm/method performance. I would much rather see the latter. The protocols being considered in this figure should probably be given easy-to-reference names, such as Protocol 1. The protocols are referred to using long (somewhat confusing) descriptive phrases. The possibility for confusion is illustrated best when the authors use the phrase "the highest, i.e., the medium".

We agree with the reviewer that the naming of the optimization protocols left room for improvement. Hence, we changed the naming of the learning rate schedules and included the highest learning rate already at an earlier point of the study. This should substantially simplify the understanding for readers, as suggested by the reviewer.

Changes in the manuscript:

Initial submission:

We benchmarked the four implemented optimization algorithms: Vanilla stochastic gradient descent (SGD), stochastic gradient descent with momentum, RMSProp, and Adam (details are given in the Methods section). [...]

We found across all algorithms that the highest, i.e., the medium, but decreasing learning rate was preferred, the low but decreasing learning rate was second and the constant, medium learning rate resulted in the worst performance (Fig. 2B). A higher learning rate in the beginning of the optimization process seemed to be crucial for the mini-batch optimizers to progress quickly towards favorable regions of the parameter space (Supplementary Fig. 1, 2, and 3). Given the medium learning rate, different algorithms were able to compete with or even outperform the full-batch optimizer Ipopt, but the adaptive algorithm RMSProp performed particularly well. In most cases, the preferred learning rates led to step-sizes during optimization which were comparable or slightly lower than those which were chosen by classical (full-batch) optimization methods (Supplementary Fig. 4).

Given these findings, we compared the optimization algorithm Adam – which is maybe the most popular algorithm for training deep neural nets – with two different tuning variants: the tuning proposed in the original publication (called standard, see (33)) and a simplified scheme (called tuned), which employs the same rate for both internally used decaying averages (see Methods for more details). The analysis of the best 25 starts for all models with medium, decreasing learning rate showed that the tuned version outperformed the original one for all cases on our benchmark examples (Fig. 2C, Supplementary Fig. 1, 2, and 3). When comparing the performance of the tuned version of Adam and RMSProp with medium learning rate, we see that they show a very similar performance for the best 25 starts for all three tested models and perform as good as Ipopt or even better (Fig. 2D-F).

We then assessed the impact of the mini-batch size on the optimization result. Again, we used an average ranking, 100 starts, and investigated 6 mini-batch sizes for each model. We restricted our analysis to the two previously best performing optimization algorithms, tuned Adam and RMSProp, with the medium but decreasing learning rate. We found that in general, small mini-batch sizes were preferred, but the optimal size seemed to be model dependent (Fig. 2G, Supplementary Fig. 5, 6, and 7). While a mini-batch size of only one experimental condition worked best for the smallest example (Fujita), a mini-batch size of 10 experimental conditions performed best for the other two examples, yielding about 0.1% to 1% of the whole dataset as mini-batch.

Changes in the manuscript (continued):

Revised submission:

We benchmarked the four implemented optimization algorithms: SGD, SGD with momentum, RMSProp, and Adam (details are given in the Methods section). To assess the impact of the learning rate, we considered **four learning rate schedules**:

- **Schedule 1: High learning rate, logarithmically decreasing**
- **Schedule 2: Medium learning rate, logarithmically decreasing**
- **Schedule 3: Low learning rate, logarithmically decreasing**
- **Schedule 4: Between low and medium learning rate, constant**

Details on these choices are given in the Methods section. [...]

For all algorithms except SGD, Schedule 2 was a reasonable choice. For SGD, Schedule 1 was better and for RMSProp, Schedules 1 and 2 performed approximately equally well. Schedule 4 was the worst choice (Fig. 2b). A higher learning rate in the beginning of the optimization process seemed to be **beneficial** for the mini-batch optimizers to progress quickly towards favorable regions of the parameter space (Supplementary Fig. 1, 2, and 3). Using **Schedule 2**, different algorithms were able to compete with or even outperform the full-batch optimizer Iopt, but the adaptive algorithm RMSProp performed particularly well. In most cases, the preferred learning rates led to step-sizes during optimization which were comparable or slightly lower than those which were chosen by classic (full-batch) optimization methods (Supplementary Fig. 4).

Given these findings, we compared the optimization algorithm Adam – which is maybe the most popular algorithm for training deep neural nets – with two different tuning variants: the tuning proposed in the original publication (called standard, see (33)) and a simplified scheme (called **balanced**), which employs the same rate for both internally used decaying averages (see Methods for more details). The analysis of the best 25 starts for all models with **Schedule 2** showed that the **balanced** version outperformed the original one for all cases on our benchmark examples (Fig. 2c, Supplementary Fig. 1, 2, and 3). When comparing the performance of the **balanced** version of Adam and RMSProp with **Schedule 2**, we see that they show a very similar performance for the best 25 starts for all three tested models **and outperform the remaining algorithms**(Fig. 2d-f).

We then assessed the impact of the mini-batch size on the optimization result. Again, we used an average ranking, 100 starts, and investigated 6 mini-batch sizes for each model. We restricted our analysis to the two previously best performing optimization algorithms, **balanced** Adam and RMSProp, with **Schedule 2**. We found that in general, small mini-batch sizes **of about 0.1% to 1% of the whole dataset** were preferred, but the optimal size seemed to be model dependent (Fig. 2g, Supplementary Fig. 5, 6, and 7).

Concerning how to best compare optimization results (using a relative or an absolute measure), we also agree that different approaches are possible. We strengthened a comparison on an absolute scale by including more optimizer setting into subfigures d-f of Figure 2. Overall however, we want to (and need to) compare too many different optimizer settings and optimization runs across a set of models, to show all these data in the main manuscript. Note that in Figure 2,

subfigure b, we compare in total the following settings and optimizations:

$$\begin{bmatrix} \text{SGD} \\ \text{Momentum} \\ \text{RMSProp} \\ \text{Adam (standard)} \\ \text{Adam (balanced)} \end{bmatrix} \times \begin{bmatrix} \text{Schedule 1} \\ \text{Schedule 2} \\ \text{Schedule 3} \\ \text{Schedule 4} \end{bmatrix} \times \begin{bmatrix} \text{Fujita} \\ \text{Bachmann} \\ \text{Lucarelli} \end{bmatrix} \times \begin{bmatrix} \text{optimization run \#1} \\ \text{optimization run \#2} \\ \vdots \\ \text{optimization run \#100} \end{bmatrix}$$

In subfigure g of Figure 2, we compare the following settings and optimizations:

$$\begin{bmatrix} \text{RMSProp} \\ \text{Adam (balanced)} \end{bmatrix} \times \begin{bmatrix} \text{mini-batch size 1} \\ \text{mini-batch size 3} \\ \text{mini-batch size 10} \\ \text{mini-batch size 30} \\ \text{mini-batch size 100} \\ \text{mini-batch size 300} \end{bmatrix} \times \begin{bmatrix} \text{Fujita} \\ \text{Bachmann} \\ \text{Lucarelli} \end{bmatrix} \times \begin{bmatrix} \text{optimization run \#1} \\ \text{optimization run \#2} \\ \vdots \\ \text{optimization run \#100} \end{bmatrix}$$

Showing all data without using some kind of summary statistic would make it hard to draw conclusions. Finding such an "absolute" summary statistic is hard, as this is strongly model dependent. Therefore, we included the full-batch optimizer Ipopt in the ranking, since this provides a benchmark to which one can compare. Interior-point optimizers are typical tools when performing parameter estimation for large-scale ODE models with box-constraints (see e.g., (75)), and Ipopt has, beyond being awarded to be "outstanding numerical software"¹, also shown in previous benchmarks to perform very well (64). Yet, we also show an optimizer comparison on an absolute scale, but this is part of the Supplementary information, i.e., Supplementary Figures 1-3 and 5-7 (6 page filling plots). We apologize for not making clear that a comparison on the absolute scale was included in the initial submission. In the revised manuscript, we stress this in the following text. In these supplementary figures, all information can be found in detail, without being condensed into two single subplots.

Changes in the manuscript:

Initial submission:

We found that in general, small mini-batch sizes of about 0.1% to 1% of the whole dataset were preferred, but the optimal size seemed to be model dependent (Fig. 2g, Supplementary Fig. 5, 6, and 7). Interestingly, the mini-batch size seemed to impact both optimization algorithms to the same degree.

Revised submission:

We found that in general, small mini-batch sizes of about 0.1% to 1% of the whole dataset were preferred, but the optimal size seemed to be model dependent (Fig. 2g, Supplementary Fig. 5, 6, and 7). Interestingly, the mini-batch size seemed to impact both optimization algorithms to the same degree. A more comprehensive analysis of the optimization results, which does not rely on summary statistics, is given in the Supplementary information, and there in the figures 1 to 7.

Changes in the manuscript:

Revised Figure 2:

See next page

¹Carl Laird and Andreas Wächter were awarded with the J. H. Wilkinson Prize for Numerical Software for their implementation of Ipopt. (84)

Figure 2: Benchmarking full-batch against mini-batch optimization methods on small- to medium-scale models. **a** Schematic overview of optimizer comparison: Benchmark models were chosen, noisy artificial data created, 100 initial points randomly sampled and different local optimizers started, each start was ranked between optimizers and an averaged score was computed. **b** Comparison of performance for different local optimizers with different learning rate schedules (lower rank implies better performance, ranks averaged over models). **c** Top 25 starts of the local optimizer Adam with tuning parameters taken from the literature (standard) vs. a simplified version (balanced). **d-f** Boxplots of final cost function values for the best 25 starts of the investigated mini-batch optimizers including the balanced version of Adam, denoted as Adam (b), compared against the Ipopt (full-batch optimizer), for each model. **g** Comparison of performance for all starts of the best two mini-batch optimizers given the learning rate Schedule 2, for different mini-batch sizes, compared against Ipopt (ranks averaged over models).

The authors compare mini-batch optimization methods against Ipopt, which is a commonly used software package for constrained optimization. Without constraints, this package isn't really necessary. Why do the authors not consider comparison against standard unconstrained optimization methods? An answer to this question should be given.

We consider box constraints on our model parameters (see also comment above), which makes it necessary to employ some kind of constrained optimization method. This kind of constraint is indeed very common for ODE models of biological processes, as box constraints allow to restrict model parameter to biologically plausible ranges (56). This is often useful, as it allows to avoid too small or large parameter values, which may lead to numerical problems, such as ODE integration failure. In principle, it would also be possible to add a penalty term for biologically implausible parameter values to the objective function and use unconstrained optimization methods. However, in our experience and also based on exchange with other groups that work in the field of ODE modeling, such as the group of Clemens Kreutz, unconstrained optimization algorithms have generally worked less reliably than algorithms implementing box-constraints, such as lsqnonlin or fmincon (from the MATLAB optimization package), or Ipopt. As also pointed out above, Ipopt – or more generally, interior-point methods – have shown in previous studies to be among the best optimization methods for these kind of problems (64; 75), which is why we consider them as Gold standards.

For adjoint sensitivity analysis, the authors cite one of their recent papers. This method (from the 50's) is classical and widely used in diverse fields, and perhaps citation of other papers would be appropriate, to give readers unfamiliar with the method a better indication of its long history.

Adjoint sensitivity analysis is an advanced, but already known method in the field of systems and computational biology and is not the actual novelty of our manuscript. Hence, citing the implementation (16) rather than its first appearance in theoretical papers seemed more reasonable to us, as it is more helpful for readers which would like to reproduce our results. For completeness, we decided to put an additional review on adjoint methods as additional reference on adjoint sensitivity analysis in the Methods section.

Changes in the manuscript:

Initial submission:

In order to compute accurate gradients, we employed adjoint sensitivity analysis, which is currently the most performing method for gradient computation of high-dimensional ODE systems (16; 17).

Revised submission:

In order to compute accurate gradients, we employed adjoint sensitivity analysis (see (54) for a review on the method), which is currently the most scalable method for gradient computation of high-dimensional ODE systems (16; 65).

SGD is referred to as "vanilla SGD." The method's accepted name is just SGD.

We changed all of the appearances of "vanilla SGD" in the manuscript and the figures to "SGD".

The authors refer to a variation of Adam as "tuned." This descriptor is not very descriptive. Better descriptors might be "balanced" or "with equal decay rates."

We changed the naming of the method in the Figures and the text to "Adam (balanced)".

Could the authors comment on why the Fujita model is having problems with successful numerical integration?

As we took this model from a benchmark collection of published ODE models (24) and did not create it ourselves, we cannot give a precise reason for this effect. However, such a behavior is not uncommon for ODE models. This is the main reason why we implemented the rescue interceptor in our optimization implementation. In general, the degree of problems with numerical integration is, from our experience based on empirical studies (68), highly model dependent. It might be that the the model structure of the Fujita model in combination with biologically meaningful parameter values leads to higher stiffness in the ODE system than for other models, or that parameter bounds for optimization provided by Fujita et al. are less appropriate than those provided by the authors of other models. For example, ODE integration failure was substantially more frequent for our large-scale application example, as shown in Supplementary Figure 9.

Confusingly, the authors discuss two distinct methodological innovations that both involve use of line search. These methods should be given more distinctive names. It was difficult to figure out which of the two methods the authors were discussing at points in the text.

We agree that, although having different purposes, both functionalities have strong similarities. As the discussion of both methods may indeed lead to confusion, we decided to use the following naming:

- The functionality which aims at recovering local optimizations after failed ODE integration is always called "rescue interceptor" in the revised manuscript and described as a (one-dimensional) trust-region implementation, not as line-search any more. This is correct, as it does not only affect the optimization step with ODE integration failure, but also the subsequent ones.
- The functionality, which aims at reducing the step sizes to more appropriate lengths, is just called "line-search" in the revised manuscript.

As both functionalities are explained in a more distinct way with more distinct names in the revised version, we hope that any possible confusion is now resolved.

Changes in the manuscript:

Initial submission:

Hence, we implemented a rescue functionality, which attempts to recover a local optimization by undoing the previous step and performing backtracking line-search. In some cases, these failures happened at the initial points of optimization, and could hence not be recovered. In all of the remaining cases, the rescue functionality was able to successfully recover the respective local optimization (Fig. 3B). More details are given in the Methods section and Supplementary Information, Algorithm 5.

In the previous study, best optimization performance was achieved with medium learning rates. In the following, we increased the learning rate to higher values, but found it obstructing the optimization process (Fig. 3C). As overall, higher learning rates were beneficial and as it is a priori not clear for a given model what a good learning rate would be, we implemented an additional backtracking line-search. It re-evaluates the objective function without gradient on the same mini-batch for different step-sizes, before accepting a proposed step (Fig. 3D). Details on the implementation can be found in the Methods section (Fig. 7) and Supplementary Information, Algorithm 6.

We evaluated these two algorithmic improvements for Adam and the medium and high learning rate on the three benchmark models (Fig. 3E, Supplementary Fig. 8). Interestingly, we found the strongest improvement for the largest model, although it suffers only little from integration failure (Fig. 3B). The line-search substantially improved the optimization process at high learning rates, which can be seen in a direct comparison (Fig. 3F) and in the waterfall plot (Fig. 3G). Considering all three models, we saw that the rescue functionality was generally helpful, whereas the line-search could also reduce the computational efficiency in case a good learning rate was chosen (Fig. 3E). This is not surprising, as the line-search needs additional computation time and some optimization runs were stopped prematurely due to imposed wall-time limits. However, these negative effects at lower learning rates were mild when compared against the positive effects at high learning rates and as the selection of a good learning rate is currently a trial-and-error process, the adaptation is highly beneficial.

Changes in the manuscript (continued):

Revised submission:

Hence, we implemented a so-called rescue **interceptor**, which attempts to recover a local optimization by undoing the previous step and acting like a one-dimensional trust-region implementation (more details in the Methods section and Supplementary Information, Algorithm 5). In some cases, these failures occurred at the initial points of optimization, and could hence not be recovered. In all of the remaining cases, the rescue **interceptor** was able to successfully recover the respective local optimization (Fig. 3b).

In the previous **batch of tests**, the best optimization performance was achieved with **learning rate Schedule 2**, while a higher learning rate, i.e., **Schedule 1**, obstructed the optimization process (Fig. 3c). As overall, higher learning rates **tended to be** beneficial and as it is a priori not clear for a given model what a good learning rate would be, we **additionally** implemented a backtracking line-search. It re-evaluates the objective function without gradient on the same mini-batch for different step-sizes, before accepting a proposed step (Fig. 3d). Details on the implementation can be found in the Methods section (Fig. 8) and Supplementary Information, Algorithm 6.

We evaluated these two algorithmic improvements for Adam and the **learning rate Schedules 1 and 2** on the three benchmark models (Fig. 3e, Supplementary Fig. 8). Interestingly, we found the strongest improvement for the largest model, although it suffered only mildly from integration failure (Fig. 3b). The line-search substantially improved the optimization process at high learning rates, which can be seen in a direct comparison (Fig. 3f) and in the waterfall plot (Fig. 3g). Considering all three models, we saw that the rescue **interceptor** was generally helpful, whereas the line-search could also reduce the computational efficiency in case a good learning rate **had been** chosen (Fig. 3e). This is not surprising, as the line-search **increased in a few cases the** computation time **by up to 9%** and some optimization runs were stopped prematurely due to imposed wall-time limits (Supplementary Fig. 9). However, these negative effects at lower learning rates were mild when compared against the positive effects at high learning rates and as the selection of a good learning rate is currently a trial-and-error process, the adaptation is highly beneficial.

It might be helpful to replace "13,000 data points" with "13,000 of the 16,308 data points" and to replace "3,308 data points" with "3,308 of the 16,308 data points."

We thank the reviewer for this suggestions and adhered to it in the main text and the method section for greater clarity.

Changes in the manuscript:

Initial submission:

The training data is taken from 21 tissues with 7 different mechanistic targeted drugs at 8 different concentrations, adding up to 13,000 data points and experimental conditions (Fig. 4B). The test data comprises the same number of drugs and concentrations and is taken from 59 cell-lines from 21 (partly different) tissues, yielding 3,308 data points and experimental conditions.

Revised submission:

The training data is taken from 21 tissues with 7 different mechanistic targeted drugs at 8 different concentrations, adding up to 13,000 **of the 16,308** data points and experimental conditions (Fig. 4b). The test data comprises the same number of drugs and concentrations and is taken from 59 cell-lines from 21 (partly different) tissues, yielding 3,308 **of the 16,308** data points and experimental conditions.

Changes in the manuscript:

Initial submission:

We used 13,000 datapoints from 233 cell-lines for model training and 3,308 datapoints from 59 cell-lines as test set.

Revised submission:

We used 13,000 **of the 16,308** datapoints from 233 cell-lines for model training and 3,308 **of the 16,308** datapoints from 59 cell-lines as test set.

The ROC analysis reported in Figure 4 could be briefly described in the main text for greater clarity.

We added a sentence in the main text to clarify the setting of our ROC analysis. However, as receiver-operator characteristics are commonly used tools for assessing the quality of classification criteria, we prefer to keep a more detailed explanation in the Methods section, to which we also refer at this point.

Changes in the manuscript:

Initial submission:

We then considered a cell-line to be responsive to a particular treatment, if the viability of the corresponding cell-line was reduced by more than a factor of two.

Revised submission:

We then **used our trained model to classify treatments for specific cell-lines into responding and non-responding situations**. Therefore, we considered a cell-line to be responsive to a particular treatment, if the viability of the corresponding cell-line was reduced by more than a factor of two.

One question that arose in my mind is the following. The authors are using an ODE model to make binary classification decisions. How does the model perform relative to a more standard machine learning approach in which a classifier is trained based on the available features?

We thank the reviewer for this interesting and important question. It has indeed already been addressed in the publication of the mentioned ODE model (17), where this ODE model was compared to a series of other classification techniques from machine learning, such as a random forest, a sparse linear and nonlinear regression model, and a network-constrained sparse regression model. Those methods were among the best performers in a previous drug sensitivity prediction DREAM challenge, making them particularly interesting competitors. The results showed that the trained ODE model was on par with the best performers. Furthermore, in this comparison, the ODE model was trained on less data and with full-batch optimizers, due to the high computation times involved in model calibration. Hence, we expect the results for a model trained with mini-batch optimization to be at least as good as the state-of-the-art machine learning techniques.

Could the authors summarize their recommendations in the form of a flow chart? I'd appreciate a clearer statement of the bottom-line recommendations.

We agree that clearer suggestions may be helpful for many readers. Therefore, we added a bullet point list summarizing our suggestions for mini-batch optimization of ODE models in the Discussion section.

Changes in the manuscript:

Initial submission:

Thirdly, the choice of the optimization algorithms seems to be less important, as at least Adam and RMSProp performed equally well on all examples.

Revised submission:

Thirdly, the choice of the optimization algorithms seems to be less important, as at least Adam and RMSProp performed equally well on all examples. **Therefore, we suggest the following approach for hyperparameter tuning when using mini-batch optimization with ODE models:**

- *Optimization algorithm:* Choose an "adaptive" optimization algorithm: Beyond the here tested RMSProp and (balanced) Adam algorithms, many more algorithms with similar behavior exist, such as recently discussed in (62).
- *Learning rate:* The learning rate is probably the most problematic, because strongly model dependent hyperparameter: Many full-batch optimizers use choices such as $\sqrt{n_\theta}$ as initial step size, with n_θ being the number of model parameters. Our results suggest that using a learning rate which results in initial step sizes in the range of $\kappa \cdot \sqrt{n_\theta}$, with $\kappa \in [0.01, 0.1]$, is a reasonable first choice for mini-batch optimization of ODE models. For particularly large models, small values of κ were more successful. Furthermore, decreasing learning rate schemes tended to be more helpful for optimizer convergence than constant learning rates. However, at the moment, it seems most helpful to test two or three different learning rate schedules.
- *Mini-batch size:* Start using very small mini-batch sizes first, and then test the effect of increasing the mini-batch sizes. From our experience, this will lead to finding an appropriate mini-batch size faster than starting with large mini-batch sizes.
- *Number of epochs:* A recent study has come to the conclusion that the overall number of optimization steps is roughly conserved for full-batch optimizers (24): For a large group of models, it was typically in the range of a few hundred to at most a one or two thousand optimization steps. Since mini-batch optimization cannot be expected to converge in fewer steps, but should do so in fewer epochs, choose the number of epochs accordingly, e.g., in the order of a few dozen epochs at most, depending on the number of optimization steps per epoch.

The authors remark that their early study (Ref. 14) suffered from overfitting in the first paragraph of the Discussion section. Could the authors elaborate to better explain what problem is being solved with the new better mini-batch optimization approach? At present, a reader would have to consult Ref. 14 to fully grasp the point being made.

It has been reported that the stochasticity in the objective function and gradient evaluation in mini-batch optimization help to avoid overfitting (58). The theoretical justification for this effect is that the dataset used for model calibration is presented less often to the computational model than in full-batch optimization. In Figure 6 of the revised manuscript, the best performing mini-batch optimization strategy is trained for only 10 epochs (i.e., 10 passes through the dataset), while full-batch optimization requires 150 of such passes through the dataset. This figure also shows that the model trained with mini-batch optimization generalizes better to the (unseen) test data than the model trained with full-batch optimization.

In addition, we clarified what was meant with the overfitting we had observed in (17):

Changes in the manuscript:

Initial submission:

The stochasticity introduced by the mini-batching in combination with the large dataset seemed to have addressed the overfitting observed in previous studies (17) and ensemble modeling improved the prediction accuracy further.

Revised submission:

In a previous study relying on full-batch optimization (17), it was observed that the error of model predictions resulted rather from overfitting than from prediction uncertainty. It seemed that the fewer passes through the dataset and the stochasticity introduced by the mini-batching in combination with the large dataset have addressed this overfitting. Ensemble modeling, for which mini-batch optimization is particularly well suited, improved the prediction accuracy further.

A claim is made that early stopping avoids overfitting? This is not obvious to me. Could the authors explain their thinking a little bit better?

Early-stopping methods are commonly employed in order to reduce overfitting and to regularize the training (see e.g. (21), Chapter 8.1.1 and 8.1.2, or (23)). Again, the theoretical justification for this lies in the fact that fewer passes through the data set during model training will lead to a smaller bias of model predictions towards the training data. Furthermore, stopping earlier, the estimates are effectively regularized towards the starting point of the optimization. We added a reference supporting this claim in the main manuscript.

Is choosing the same standard deviation for all data points (in the discussion of Eq. (4)) the same as just omitting the standard deviation weighting from the optimization problem?

Mathematically, this is indeed equivalent and the inferred optimum will be identical. However, our implementation allows to directly use different standard deviations (mathematically equivalent to weighted least-squares fitting), or to also fit parameters for the standard deviations based on the measured data, as done in (56). However, we decided to use the same identical standard deviation terms in our application, as no reliable values for these terms were available from measurements for our application example.

How was approach to steady state determined for the large-scale model simulations? Nothing is said in this version of the manuscript about that.

We simulated the model until steady-state was reached (by default to a time point of $t = 10^8$), and checked the norm of the right hand side afterwards, such as also done in (17). We added an explanation clarifying this approach in the Methods section. Although our ODE solving framework AMICI also allows the usage of direct steady-state computation methods such as a dampened Newton-method (22; 85), applying such direct techniques is non-trivial as they will only converge if started in proximity of a steady-state. Furthermore, any improvement on the method of inferring a steady-state is independent of and can be combined with mini-batch optimization.

Changes in the manuscript:

Initial submission:

The data $D = \{\bar{y}_{e,i}\}_{e=1,\dots,M, i=1,\dots,N_e}$ was simulated as a time-course for the small- to medium-scale models. For the large-scale model, it was taken at steady-state, yielding only one data-point per experimental condition e .

Revised submission:

The data $D = \{\bar{y}_{e,i}\}_{e=1,\dots,M, i=1,\dots,N_e}$ was simulated as a time-course for the small- to medium-scale models. For the large-scale model, it was taken at steady-state, yielding only one data-point per experimental condition e . **The steady-state was inferred by simulating the model to time-point $t = 10^8$ (seconds). Afterwards, we verified whether the model trajectories had reached steady-state by assessing the L^2 -norm of the right hand side of the ODE, using the an absolute tolerance of 10^{-16} and relative tolerance of 10^{-8} as convergence threshold.**

Just below Eq. (6), it is said that mini batch sizes are all the same. This doesn't seem possible except for special cases. Rather, they are all nearly the same?

The reviewer is correct that in general, it will not be possible to guarantee exactly equally sized mini-batches for all cases. However, in our case, the training set (for the main application example) has the size of 13,000 independent experimental conditions, all mini-batch sizes which divide 13,000 will yield exactly equally sized mini-batches. As we used mini-batch sizes of 10, 20, 100, and 1000 experimental conditions, all of our mini-batches were indeed equally sized. If the mini-batch size does not divide the data set size, our implementation would automatically use equally sized mini batches for all optimization steps except the last one per epoch. The last step would use a smaller mini batch size with the remaining data points.

The vector δ_r is described as provided a direction, just above Eq. (7). I think the stepsize is given by the product of δ_r and the scalar learning rate η_r , so... δ_r is providing not just a direction but also contributing to the stepsize?

Indeed, δ_r provides the direction of the parameter update, but may (depending on the optimization algorithm) also contribute to the step size. This implicit coupling of parameter update and step size is one of the reasons why it is hard to suggest good learning rate schemes. We already tried to explain this problem in the initial submission. However, we decided to put more emphasis on it and to clarify it further in the revised version of the manuscript.

Changes in the manuscript:

Initial submission:

Assuming a given algorithm in optimization step r produced a parameter update (direction) δ_r , then the next proposed parameter vector would be

$$\theta^{(r+1)} = \theta^{(r)} + \eta_r \cdot \delta_r, \quad (4)$$

with η_r being the learning rate at step r . Obviously, the learning rate influences the step-size of the optimization algorithm in parameter space. However, many of the algorithms we investigated yielded parameter updates with $\|\delta_r\| \neq 1$.

Revised submission:

Assuming a given algorithm in optimization step r produced a parameter update δ_r , then the next proposed parameter vector would be

$$\theta^{(r+1)} = \theta^{(r)} + \eta_r \cdot \delta_r, \quad (5)$$

with η_r being the learning rate at step r . Obviously, the learning rate influences the step-size of the optimization algorithm in parameter space. However, many of the algorithms we investigated yielded parameter updates with $\|\delta_r\| \neq 1$, thus δ_r does not only provide the direction of the parameter update but may also contribute to the step size.

What happens in the optimization procedure if rescue fails (say, on the first step)?

The rescue interceptor is applied repeatedly, until a maximum number of repetitions (10 in our case) is reached. As seen in the evaluation of the rescue interceptor in Figure 3, this occurred very rarely during optimization: For the small to medium-sized benchmark models, it did not happen at all, for the large-scale application example, it happened only once in 100 optimization runs (Supplementary Figure 12). In this case however, the respective local optimization run is stopped and not pursued any further. We think this implementation is reasonable, as it is analogous to the one of classic full-batch optimization algorithms and also shows a comparable performance. However, if optimization fails at the initial point, not much can be done, as there is no previous step to which the optimizer could return and redo the parameter update. Hence, failure at the initial point of optimization directly leads to a cancelled local optimization run. Also this behavior is in line with state-of-the-art implementations of current parameter estimation frameworks for ODE models using full-batch optimizers.

Changes in the manuscript:

Initial submission:

In some cases, these failures occurred at the initial points of optimization, and could hence not be recovered. In all of the remaining cases, the rescue functionality was able to successfully recover the respective local optimization (Fig. 3b).

Revised submission:

In some cases, these failures occurred at the initial points of optimization. These initial failures cannot be recovered and will lead to a failed local optimization, even when using the rescue interceptor. In all of the remaining cases, the rescue interceptor was able to successfully recover the respective local optimization (Fig. 3b).

Changes in the manuscript:

Initial submission:

The rescue functionality was implemented as an iterative backtracking line-search algorithm, performing at most ten iterations. It is triggered, if the objective function and its gradient cannot be evaluated. In this case, it keeps the current mini-batch, but undoes the previous parameter update, reducing the step length in each line-search iteration. More details on the method and its pseudo-code are given in the Supplementary Information (Algorithm 5 for the pseudo-code).

Revised submission:

The rescue interceptor was implemented to mimic the behavior of a one-dimensional trust-region algorithm. It works similar to an iterative backtracking line-search algorithm, which also reduces the step-length of the subsequent optimization steps, performing at most ten iterations. It is triggered, if the objective function and its gradient cannot be evaluated. In this case, it keeps the current mini-batch, but undoes the previous parameter update, and reduces the current step length. In the next steps, the step length is gradually increased again until it reaches its original, unmodified value. If ten repetitions of this procedure are not sufficient to restore the optimization process, the corresponding local optimization run is stopped. More details on the method and its pseudo-code are given in the Supplementary Information (Algorithm 5 for the pseudo-code).

The authors selected a handful of mini-batch optimization methods to evaluate. In the Discussion, it might be helpful to briefly acknowledge some of the other methods in this class that were not evaluated.

We agree that there exists a variety of mini-batch optimization methods in the literature. However, as we are limited in the number of references for articles in Nature Communications, we decided to rather cite a recent preprint, which summarizes more than 100 such optimization algorithms and benchmarks them (on deep learning problems). Interestingly, some of the main findings in this study coincide with ours, even if in a different field of application. The additional algorithms were added in the bullet point list of the discussion:

Changes in the manuscript:

Initial submission:

Thirdly, the choice of the optimization algorithms seems to be less important, as at least Adam and RMSProp performed equally well on all examples.

Revised submission:

Thirdly, the choice of the optimization algorithms seems to be less important, as at least Adam and RMSProp performed equally well on all examples. Therefore, we suggest the following approach for hyperparameter tuning when using mini-batch optimization with ODE models:

- Optimization algorithm: Choose an "adaptive" optimization algorithm: Beyond the here tested RMSProp and (balanced) Adam algorithms, many more algorithms with similar behavior exist, such as recently discussed in (62).
- ...

Response to Reviewer 2

The manuscript by Stapor et al 'Mini-batch optimization enables training of ODE models on large-scale datasets' presents an application of mini-batch based learning of parameters of large ODE-based models of chemical networks. The study is based on the previously published model and the fast learning methodology was also previously published. However, the current study increases the size of the dataset on which the training has been performed which is possible due to learning optimization thanks to the application of mini-batch trick.

The manuscript is well written and technically correct as far as one can judge because the description is rather technical and specific and requires the good knowledge of the used industrial scale optimizers and machine learning terminology. I am convinced that using mini-batches is an excellent idea to further scale up the learning of ODE parameters in large models.

We thank the reviewer for receiving our work positively.

However, I am not sure that the message of the manuscript represents a sufficient breakthrough with respect to already published work and that it can be interesting to a wide readership. The model and the accelerated parameter fitting formalism has been published by the authors of this manuscript before (PLoS Comp Biol, 2017; Cell Systems, 2018) which was a quite remarkable achievement. Mini-batch approach and its application in the current study also looks quite known and widely exploited idea even if in a different field (AI).

I do not feel that the current manuscript promises a similar advance, representing rather an important technical improvement to already established framework.

We agree that some of the past methodological advances led to speed-ups which were even higher. However, the previous advances cited by the reviewer – adjoint sensitivity analysis (16) and exploiting the sparsity of the system of ordinary differential equations (17) – were "only" reducing the overall computation time by about two orders of magnitude. Unlike those, mini-batch optimization does not only speed up computations, but comes with substantial further improvements:

1. Mini-batch optimization led, if good choices of hyperparameters were used, to substantially better optimization results. This goes beyond a pure computational speed-up and is remarkable, as the benchmark we compared against was the well-established and award-winning² optimizer Ipopt, which has been used and tested in many projects since many years.
2. Mini-batch optimization creates a much richer optimization history, from which parameter ensembles can be created. These ensembles can be used for (lightweight) Bayesian approaches for model predictions and uncertainty quantification (e.g. ensemble methods), which have been inapplicable to problems of the size of the discussed application example: Similar parameter ensembles created from classic optimization approaches may contain a few dozens, maybe one hundred parameter vectors. However, for the considered problem size, this is substantially less than the number of free model parameters. Creating an ensemble which actually contains more parameter vectors than free model parameters was only possible due to using mini-batch optimization. This opens up new possibilities to analyse the mathematical structure of the parameter estimation problem.
3. Mini-batch optimization comes with a better exploration of the parameter space, which

²Carl Laird and Andreas Wächter were awarded with the J. H. Wilkinson Prize for Numerical Software for their implementation of Ipopt. (84)

again is important in terms of (global) optimization quality, but also for uncertainty quantification. Mini-batching techniques can be used to also explore the parameter space via Markov-chain Monte Carlo (MCMC) sampling, either by repurposing specific optimization techniques (82) or by implementing mini-batch versions of, e.g., the Metropolis-Hastings algorithm (83).

For these reasons, we think that introducing mini-batch techniques into the field of dynamic modeling may open up a door to new research, which is what we want to do with our contribution.

However, we agree with the reviewer that some of these advantages may have not become clear enough to the reader from the initial submission. Therefore, we extended our manuscript on these points. We revised the text on the large-scale optimization problem and added a new section, which is concerned with ensemble methods and uncertainty analysis and created an additional figure for this (Figure 7 in the revised manuscript), as well as an additional supplementary figure (Supplementary Figure 16 in the revised manuscript).

Changes in the manuscript (Discussion section):

Initial submission:

The trained model provided accurate predictions whether a certain treatment would reduce cell viability by more than 50% for a chosen cell-line in more than 85% of the cases, even on cell-lines which had not been used for model training. The stochasticity introduced by the mini-batching in combination with the large dataset seemed to have addressed the overfitting observed in previous studies (17) and ensemble modeling improved the prediction accuracy further. Overall, our implementation reduced computation time by more than one order of magnitude while providing better fitting results than established methods. The improved scaling characteristics should render problems with even larger datasets feasible.

Revised submission:

The trained model provided accurate predictions whether a certain treatment would reduce cell viability by more than 50% for a chosen cell-line in more than 85% of the cases, even on cell-lines which had not been used for model training. Furthermore, we performed in-silico gene knockouts and validated them on a large experimental CRISPR screen from literature, which helped us to improve the model topology and to obtain well-matching knockout predictions. In a last step, we performed an uncertainty analysis based on ensemble methods and assessed the exploration of the parameter space during optimization, where mini-batch optimization enabled results which had not been achievable with established optimization methods.

A closer analysis of the in-silico knockout study allowed us to pinpoint deficiencies of the large-scale cancer signaling model, e.g., concerning the implementation of cell cycle genes, and a subsequent model refinement yielded substantially improved model predictions. To the best of our knowledge, this is the first time that a full modeling cycle, including model refinement and re-training, has been performed for an ODE model of this size, as the cost of model calibration renders iterative refinement usually prohibitive for large ODE models. Yet, a set of incorrect predictions remained: In particular, the essentiality of genes that encode master kinases such as PDPK1 or phosphatases such as PPP2CA was underestimated. At the same time, the importance of some genes encoding proteins involved in the AKT and RAS signaling pathways was overestimated, especially those which either interact with the EGF receptor, such as SRC, VAV2, or EGFR itself, or those which acts as transcription factors, such as CREB1, ELK1, or STAT3. Hence, we assume that a more comprehensive model, trained on a larger dataset, would be needed to, e.g., reliably discover new drug targets or to reject possible drug candidates before entering a clinical trial. The datasets to train such a more comprehensive, whole-cell signaling model have been made available in the last years, and mini-batch optimization renders the calibration of such a model computationally feasible.

Changes in the manuscript (continued):

In a previous study relying on full-batch optimization (17), it was observed that the error of model predictions resulted rather from overfitting than from prediction uncertainty. It seemed that the shorter training times and the stochasticity introduced by the mini-batching in combination with the large dataset have addressed this overfitting. Ensemble modeling, for which mini-batch optimization is particularly well suited, improved the prediction accuracy further. Overall, our implementation reduced computation time by more than one order of magnitude while providing better fitting results and predictions than established methods. The improved scaling characteristics of mini-batch optimization should render problems with even larger datasets feasible. In addition, the ensemble analysis indicated that a better exploration of the parameter space would be necessary to obtain more reliable uncertainty estimates – a goal, which needs further research on mini-batch optimization methods and which is unlikely to be achievable with full-batch optimization methods.

Changes in the manuscript:

Added Figure 7:

See next page

The scale of the used dataset is indeed impressive. However, it remains unclear how relevant are the resulting predictions, since there is no real validation part in the manuscript besides the standard machine-learning based statistical validation. The correlation plot and the ROC curves in Figure 4 look quite promising but it is difficult to judge the actual value of the fit performed under so many conditions merged together. In order to prove the value of the undertaken large scale parameter fitting exercise, I would generally expect more insights into what is the level of biological relevance of the trained model predictions. I would also like to see a clear demonstration that the reported predictive power is not of artefactual nature (such as absence of artefacts similar to the Simpson's paradox resulting from merging many heterogeneous sources of data together).

We agree with the reviewer that the validation of our calibrated model left room for improvement. We are convinced that the additional validations now included in the revised manuscript come indeed with interesting conclusions. We extended the model validation part significantly, showing on one hand model fits and drug response predictions for the training and test dataset. We show the model output for two cell lines in the main manuscript, for both datasets respectively (i.e., four cell lines in total): one well and one poorly described cell-line, each (Figures 4 and 5 in the revised manuscript). In addition, the model outputs for all cell lines on the training and the validation set (i.e. model output for all 16,308 data points) are now shown in the supplementary information (which are in total 292 additional page filling plots). On the other hand, we validate our model on a large screen of gene knockouts, which is strongly out-of-sample for a model validation and would not be easily possible with statistical methods, if being trained solely on drug responses. This validation led indeed to interesting insights also for biological modeling. We discuss the main conclusions in the main manuscript and added a more detailed discussion of these results to the supplementary information. In particular, this analysis showed that our application example, despite being one of the most complex published ODE models of cancer signaling to date, still lacks important features and is not yet comprehensive enough to correctly explain the role of master kinases and phosphatases, cell cycle regulators, and signaling pathways alternative to the AKT and RAS signaling cascades.

Changes in the manuscript:

Revised Figure 4:

See next page

Changes in the manuscript:

Added Figure 5:

See next page

Figure 7: Analysis of parameter uncertainty based on an ensemble of parameter vectors from the the best mini-batch optimization, i.e., with batch size 10. **a** Generation of a large parameter ensemble (8,450 vectors) based on the history of multi-start local mini-batch optimization. A cutoff value was fixed and parameter vectors added to the ensemble if the objective function value was below the cutoff (red). A smaller ensemble was created from the ten best optimization results (yellow). At three checkpoints (blue), the coverage of parameter intervals was computed as a ratio of the interval covered by the optimization runs over the interval between the parameter bounds. **b** Computation of parameter interval coverage based on multi-start local optimization with mini-batch optimization and Ipopt (full-batch optimization). **c** Sensitivity analysis for model output (cell viability) w.r.t. the 4232 model parameters. Singular values of the sensitivity matrix, computed on the small parameter ensemble. **d** Principal component analysis of the large parameter ensemble created from mini-batch optimization, indicating the contribution (individual and summed) of the different PCA directions to the total observed variance in the ensemble. **e** Spectrum of the covariance matrix (for the 4232 model parameters) created from the large parameter ensemble from (d).

Figure 4: Description of application example, datasets and model performance, when trained with mini-batch optimization. **a** Simplifying illustration of the multi-pathway model of cancer signaling. **b** Left: Overview of the datasets used for training and model validation, taken from the Cancer Cell Line Encyclopedia. Right: Comparison of model sizes and experimental conditions used for model training of recently published ODE models. **c** Correlation of measured simulated cell viability for all points of the training data, color-coding indicates density in scatter plot. **d** Receiver-operating-characteristics for classification into responsive and non-responsive combinations of cell-lines and treatments on training data for the best ten optimization runs (gray) and an ensemble simulation (blue). **e** Area under ROC and classification accuracy on training data for the ten best optimization results (gray), for the ensemble model (black), and for the ensemble model on data for each drug individually (colored). **f** Simulated drug response. Left: Ranking of fit quality for cell lines by average root-mean-square error (RMSE). Right: Two out of 233 cell lines from the training data, error bars indicate standard deviation across the ensemble, for a cell line which the model was able to describe well (blue, BCPAP) and a cell line, which was less well captured by the model (orange, KU812).

Figure 5: Validation of the fitted large-scale cancer model on independent test data. **a** Correlation of test data and model prediction. Color-coding indicates density in scatter plot. **b** ROC-curves for classification of drug responses of cell lines on the test set. Classification thresholds from the training data were used. **c** Area under ROC and classification accuracy on test data for the ten best optimizations (gray), the ensemble model (black), and the ensemble model for each drug individually (colored). **d** Simulated drug response. Left: Ranking of fit quality for cell lines by average root-mean-square error. Right: Two out of 59 cell lines from the test data, error bars indicate standard deviation across the ensemble, for a cell line which the model was able to describe well (blue, 8505C) and a cell line, which was less well captured by the model (orange, JHH5). **e** ROC-curves for classification of gene essentiality. Measurement data for 18 cell lines was taken from Behan et al, 2019. In-silico knockouts are shown for the untrained model (blue), the trained model (orange), and the refined and trained model (green). **f** Measurement, prediction, and confusion matrix for essential genes for the refined and trained model. Numbers indicate how often a gene was found to be essential in experimental data and in-silico knockout predictions for the 18 cell lines, sums show the number of true and false predictions over essential genes.

Changes in the manuscript:

Initial submission:

Based on the characteristics, we computed classification thresholds for the simulation, when a cell-line together with a treatment condition is to be classified as responsive and repeated the computations for each considered drug individually. On the training data, the ensemble model achieved a classification accuracy of 86% (Fig 4E). Simulated responses to the drugs Selumetinib and Erlotinib matched particularly well, yielding AUCs of 0.95 and 0.96, and classification accuracies of 91% each, respectively.

When considering the independent set of test data and comparing it to the simulation of the ensemble model, we still found a Pearson correlation of 0.74, and hence only a slight decrease in model performance (Fig 4F). This suggests that there is only little to no overfitting. Computing the receiver-operator-characteristics yielded an AUC value of 0.94 for the ensemble model and AUC values between 0.90 and 0.92 for the ten best local optimizations (Fig 4G). This indicates that the trained model generalizes well to unseen cell-lines. We computed the classification accuracy, relying on the thresholds inferred from the training data and found that 85% of all treatment conditions were classified correctly into responsive or non-responsive (Fig 4H). Again, the AUCs and the classification accuracies varied across the considered drug, with Selumetinib and Erlotinib performing best also on the test data, with AUCs of 0.96 and 0.99 and classification accuracies of 90% and 89%, respectively.

Revised submission:

Based on the characteristics, we computed classification thresholds for the simulation, when a cell-line together with a treatment condition is to be classified as responsive and repeated the computations for each considered drug individually. On the training data, the ensemble model achieved a classification accuracy of 86% (Fig 4e). Beyond the correlation and ROC analysis, we also analyzed the model fits to drug response data. The trained model was able to describe the response to different drug treatments and also to capture the varying behavior between the 233 cell lines correctly (Fig. 4f and Supplementary Information).

Changes in the manuscript (continued):

Revised submission:

In a next step, we validated predictions of the model on cell lines from the independent test set, where we still found a Pearson correlation between the data and the simulation of ensemble model of 0.74 (Fig 5a). The ROC analysis for the classification into responding and non-responding treatments on the test set yielded an AUC value of 0.94 for the ensemble model values between 0.90 and 0.92 for the ten best local optimizations (Fig 5b), while the classification accuracy was around 85% (Fig 5c). Importantly, the model did not only classify trivial cases correctly, but was also able to capture the variability between cell lines and drugs (Fig 5d).

Mini-batch optimization renders iterative model refinement possible for large-scale ODE models

Beyond drug response predictions, we also validated the model on experimental data from a recently published CRISPR screen: Behan et al. had analyzed the change in cell viability when knocking out various genes on a large set of cell lines (5). We found 107 genes and 18 cell lines in the knockout dataset, which were also included in our application example and the dataset used for model training, respectively, summing up to 1926 data points which we used for validation. Based on the gene knockout data, we classified a gene as "essential" for a cell line, if the knockout led to a viability reduction of more than 50% and compared this essentiality classification with in-silico knockouts from our application example (Fig 5e). We computed receiver-operator-characteristics for knockout simulations based on our trained model, and on random parameters as reference. We obtained better-than-random predictions (AUC of 0.59) from the untrained model and, as expected, better predictions for the trained model (AUC of 0.63).

Yet, the classification threshold for model simulation chosen by the ROC analysis was surprisingly high, leading to many false positive predictions (Supplementary Fig. 10). As this indicated a shortcoming of the model, we added further proteins, which should contribute to the viability readout of the model (see Methods section for more details on how the viability readout is computed): First, we included the protein Checkpoint kinase 1, a major regulator of the cell cycle that is encoded by the gene CHEK1, as additional pro-proliferative readout. The gene CHEK1 was shown to be among the most essential ones for cell viability in the validation data, which was however not accounted for in the original model of (17). Secondly, we also included doubly phosphorylated MAP3K1 as pro-proliferative readout, which was so far only acting in a feedback loop with upstream proteins. This modification allowed the model to partly circumvent the downstream bottleneck of the MAPK-cascade, and hence made it possible to explain potential downstream effects of MAP3K1. After calibrating this refined model on the previously used drug response data, we obtained substantially improved predictions of gene essentiality (Fig 5e, AUC of 0.67). The classification threshold for the model readout was reduced, leading to substantially fewer false positive predictions (Fig 5f, Supplementary Fig. 11). A detailed analysis of the remaining false predictions is given in the Supplementary Information.

As confirmation of the refined model topology, we refitted the original model on the drug response and gene knockout data, i.e. 14,926 data points, simultaneously. As expected, calibrating the model towards the knockout data led to substantially improved classification results (AUC of 0.74, vs. 0.63 for the model trained on drug response data only). However, also when being refitted on the knockout data, the model was still not able to describe the data when knocking out CHEK1, confirming the limitations of the original viability readout (Supplementary Fig. 10). In a last step, we refitted also the improved model on drug response and knockout data simultaneously, which again drastically improved the classification accuracy (AUC of 0.84, vs. 0.67 for the model trained on drug response data only). This suggested that a more general ODE model of cell signaling should encode at least a basic version of the cell cycle, if the model is to explain viability. To the best of our knowledge, this is the first

I would also expect placing the current study in the context of other mathematical modeling studies aimed at exploiting the same datasets in order to unravel the mechanism of drugs action.

A study which explored (at least a part of) the large dataset derived from the CCLE with the same dynamic model has been performed in (17). We also extended, as shown, the biological interpretation of our results substantially, and tried to link our results to other studies on the dataset in the Discussion section (see already shown changes in the Discussion section).

In other words, I feel that the focus of the manuscript should be completely changed in order to make the paper interesting for the wide computational biology community. In the current form, it better fits to a journal specialized on machine learning and optimization. Therefore, I can not recommend this manuscript for publication in Nature Communications.

Indeed, we have strengthened our manuscript concerning model validation and also in terms of interpretation of the biological results. Additionally, we show how to perform uncertainty quantification for models of the considered size. However, the main focus of our manuscript is a method contribution, which we think will be highly relevant for the development of more comprehensive and more complex computational models in the future. Moreover, ODE models are used in many scientific fields beyond biology, such as epidemiology or engineering and mini-batch optimization may be helpful in these fields as well. Furthermore, we show how to transfer and adapt a computational method from one field of application to another, which may be of interest for similar endeavors with other methods. Hence, we think that our manuscript is also interesting for readers, which perform demanding computations in other research fields. We sincerely hope that we changed the focus of the main result part of the paper as well as the supplementary information sufficiently for the reviewer to now find it acceptable.

Response to Reviewer 3

The authors present a novel approach for the calibration of large-scale models by means of mini-batch optimization, typically used in machine learning approach. The method proposed by the authors is promising, as shown by the application to the large scale model of cancer signaling. I have some comments that I hope will help to improve the manuscript, making it suitable for publication on Nature Communication journal.

We thank the reviewer for receiving our work positively.

Major comments: I believe that the computational complexity and the increase in running time in the case of the application of the rescue functionality should be discussed by the authors. To this regard, I believe that the remark reported at lines 187-189 should be extended and better clarified.

We think that the reviewer might have confused the rescue functionality (now called rescue interceptor in the revised manuscript, to make the distinction clearer) and the line-search functionality, as the text part he is referring to discussed the additional computational effort of the line-search: The rescue interceptor comes at no additional cost compared to a naive implementation of mini-batch optimization algorithms. It is only triggered upon failure of ODE integration – a scenario which would cause a naive implementation to abort the current optimization run. In these cases, the rescue interceptor allows to continue the optimization process.

However, we agree that there is a trade-off between the additional computational cost of the line-search and its positive effects on the optimization process. Therefore, we added more details on this aspect and also added a supplementary figure, which we reference in the main text. This analysis showed that the line-search may cost up to roughly 10% of additional computation time, but does not need to do so. It can even reduce computation time, if the line-search causes the optimization process to leave a "problematic region" in parameter space earlier. Such "problematic regions" may lead to stiff dynamics, which are hard to integrate and therefore often lead to high computation times when solving the ODE system. We assume that the cases, in which line-search led to a reduction of computation time, may stem from this effect.

Changes in the manuscript:

Initial submission: Considering all three models, we saw that the rescue functionality was generally helpful, whereas the line-search could also reduce the computational efficiency in case a good learning rate was chosen (Fig. 3E). This is not surprising, as the line-search needs additional computation time and some optimization runs were stopped prematurely due to imposed wall-time limits. However, these negative effects at lower learning rates were mild when compared against the positive effects at high learning rates and as the selection of a good learning rate is currently a trial-and-error process, the adaptation is highly beneficial.

Revised submission: Considering all three models, we saw that the rescue **interceptor** was generally helpful, whereas the line-search could also reduce the computational efficiency in case a good learning rate **had been** chosen (Fig. 3e). This is not surprising, as the line-search **increased in a few cases the** computation time **by up to 9%** and some optimization runs were stopped prematurely due to imposed wall-time limits (**Supplementary Fig. 9**). However, these negative effects at lower learning rates were mild when compared against the positive effects at high learning rates and as the selection of a good learning rate is currently a trial-and-error process, the adaptation is highly beneficial.

Changes in the manuscript:

Additional supplementary figure:

Supplementary Figure 9: Computation time comparison when using mini-batch optimization with and without the line-search feature. Left panel: Cumulated computation times for all 100 multi-start local optimizations for the three benchmark models, for the algorithms RmsProp and Adam (balanced), for learning rate Schedules 1 and 2, with and without line-search. Right panel: Computation time ratio of the respective total computation times.

I do not understand why the results concerning high learning rate values are only presented in Fig 3C considering only Adam optimization algorithm applied to Lucarelli model. I believe that it would be interesting to have in the paper the results obtained with different algorithms on different models (as shown in Figure 2B).

We initially presented the result with the high learning rate later to put more emphasize on the aspect of choosing a good learning rate. However, we agree with the reviewer that it may be confusing to present results in such split up fashion. We changed this presentation in the revised version of the manuscript and included the studies with high learning rate already earlier, where we present it alongside with the remaining studies (i.e., in Figure 2b, as suggested by the reviewer). We also changed the naming of the learning rate schedules, to make them easier to reference.

Changes in the manuscript:

Initial submission:

To assess the impact of the learning rate, we considered three schedules: a medium and a low learning rate, both logarithmically decreasing, and a fixed learning rate. Details on these choices are given in the Methods section. [...] Computing the mean of the 100 rankings for each setting led to an averaged rank, which we used as a proxy for overall optimization quality (Fig. 2A).

We found across all algorithms that the highest, i.e., the medium, but decreasing learning rate was preferred, the low but decreasing learning rate was second and the constant, medium learning rate resulted in the worst performance (Fig. 2B). A higher learning rate in the beginning of the optimization process seemed to be crucial for the mini-batch optimizers to progress quickly towards favorable regions of the parameter space (Supplementary Fig. 1, 2, and 3). Given the medium learning rate, different algorithms were able to compete with or even outperform the full-batch optimizer Ipopt, but the adaptive algorithm RMSProp performed particularly well. In most cases, the preferred learning rates led to step-sizes during optimization which were comparable or slightly lower than those which were chosen by classical (full-batch) optimization methods (Supplementary Fig. 4).

Given these findings, we compared the optimization algorithm Adam – which is maybe the most popular algorithm for training deep neural nets – with two different tuning variants: the tuning proposed in the original publication (called standard, see (33)) and a simplified scheme (called tuned), which employs the same rate for both internally used decaying averages (see Methods for more details). The analysis of the best 25 starts for all models with medium, decreasing learning rate showed that the tuned version outperformed the original one for all cases on our benchmark examples (Fig. 2C, Supplementary Fig. 1, 2, and 3). When comparing the performance of the tuned version of Adam and RMSProp with medium learning rate, we see that they show a very similar performance for the best 25 starts for all three tested models and perform as good as Ipopt or even better (Fig. 2D-F).

Changes in the manuscript (continued):

Revised submission:

To assess the impact of the learning rate, we considered four learning rate schedules:

- Schedule 1: High learning rate, logarithmically decreasing
- Schedule 2: Medium learning rate, logarithmically decreasing
- Schedule 3: Low learning rate, logarithmically decreasing
- Schedule 4: Between low and medium learning rate, constant

Details on these choices are given in the Methods section. [...] Computing the mean of the 100 rankings for each setting led to an averaged rank, which we used as a proxy for overall optimization quality (Fig. 2a).

For all algorithms except SGD, Schedule 2 was a reasonable choice. For SGD, Schedule 1 was better and for RMSProp, Schedules 1 and 2 performed approximately equally well. Schedule 4 was the worst choice (Fig. 2b). A higher learning rate in the beginning of the optimization process seemed to be beneficial for the mini-batch optimizers to progress quickly towards favorable regions of the parameter space (Supplementary Fig. 1, 2, and 3). Using Schedule 2, different algorithms were able to compete with or even outperform the full-batch optimizer Ipopt, but the adaptive algorithm RMSProp performed particularly well. In most cases, the preferred learning rates led to step-sizes during optimization which were comparable or slightly lower than those which were chosen by classic (full-batch) optimization methods (Supplementary Fig. 4).

Given these findings, we compared the optimization algorithm Adam – which is maybe the most popular algorithm for training deep neural nets – with two different tuning variants: the tuning proposed in the original publication (called standard, see (33)) and a simplified scheme (called balanced), which employs the same rate for both internally used decaying averages (see Methods for more details). The analysis of the best 25 starts for all models with Schedule 2 showed that the balanced version outperformed the original one for all cases on our benchmark examples (Fig. 2c, Supplementary Fig. 1, 2, and 3). When comparing the performance of the balanced version of Adam and RMSProp with Schedule 2, we see that they show a very similar performance for the best 25 starts for all three tested models and outperform the remaining algorithms(Fig. 2d-f).

Changes in the manuscript:

Revised figure:

(see next page)

Considering the results presented in Figure 4, I would like to see some additional comments about the correlation value achieved (0.76-0.74) and speculations on possible strategies to improve such results.

In the revised submission, we present not only the correlation values, but all predictions of drug response on the test set, as well as additional model validations in a newly created Figure 5. This provides much more details on the data set, the level of measurement noise, and the quality of the model predictions (see revised Figure 4 and 5, as well as the supplementary information). We also derive conclusions concerning the model topology and refine and refit the large-scale model (see more details on this in the response to the reviewer’s comment on the sensitivity analysis). Overall, we think that these results can still be improved (such as done

Figure 2: Benchmarking full-batch against mini-batch optimization methods on small- to medium-scale models. **a** Schematic overview of optimizer comparison: Benchmark models were chosen, noisy artificial data created, 100 initial points randomly sampled and different local optimizers started, each start was ranked between optimizers and an averaged score was computed. **b** Comparison of performance for different local optimizers with different learning rate schedules (lower rank implies better performance, ranks averaged over models). **c** Top 25 starts of the local optimizer Adam with tuning parameters taken from the literature (standard) vs. a simplified version (balanced). **d-f** Boxplots of final cost function values for the best 25 starts of the investigated mini-batch optimizers including the balanced version of Adam, denoted as Adam (b), compared against the Ipopt (full-batch optimizer), for each model. **g** Comparison of performance for all starts of the best two mini-batch optimizers given the learning rate Schedule 2, for different mini-batch sizes, compared against Ipopt (ranks averaged over models).

by the mini-batch optimization with mini-batch size 10), but that a close-to-perfect correlation will never be possible, due to the high amount of measurement noise, which is present in the dataset. This level of noise is visible in, e.g., the lower panel of the revised Figure 5d.

Comparing the results on Figure 5A and B (top panels), I missed the reason why there is a remarkable difference in the final cost function values obtained with training and test data. Please explain.

This discrepancy is due to the fact that the validation set contains less data points than the training data: Since the cost function is summed up over all datapoints, fewer points contribute to the cost function value on the validation set, which results in a lower cost function value. As this was confusing for the reader and as the pure cost function values themselves do not provide any insight (at least on the validation data set, where no optimization trajectories were computed), we decided to omit these values in the revised version of the manuscript. This allowed us to merge Figure 5 and Figure 6 (now both in Figure 6) with some of the main findings and to add a novel Figure 5 which contains more results on model validation.

I am skeptical about the conclusion of the authors concerning the choice of hyperparameters, especially the learning rate. It seems to me that this value is highly problem dependent. The authors should extend the discussion about this aspect and provide some heuristics to choose an appropriate value for this hyperparameter (if any).

We agree with the reviewer that especially the learning rate is a highly complicated hyperparameter, which is likely to be very model dependent. Yet, optimization algorithms have to provide default values for their hyperparameters, which also includes the learning rate. As suggested, we extended the discussion of the learning rate in the revised submission. Furthermore, we now motivate our reasoning on the learning rate more explicitly from heuristics which are used in full-batch optimization. We stress that these are suggestions, which should be helpful for a wide range of models, but may not solve the problem for all applications. To really address the issue of finding a good learning rate, self-tuning algorithms that adapt the learning rate in a flexible way are necessary. We think that this aspect is a very promising subject of further research and put more emphasize on its importance in the Discussion section.

Changes in the manuscript:

Added text in the Discussion section:

Therefore, we suggest the following approach for hyperparameter tuning when using mini-batch optimization with ODE models:

- *Optimization algorithm:* Choose an "adaptive" optimization algorithm: Beyond the here tested RMSProp and (balanced) Adam algorithms, many more algorithms with similar behavior exist, such as recently discussed in (62).
- *Learning rate:* The learning rate is probably the most problematic, because strongly model dependent hyperparameter: Many full-batch optimizers use choices such as $\sqrt{n_\theta}$ as initial optimization step size, with n_θ being the number of model parameters. Our results suggest that using a learning rate which results in initial step sizes in the range of $\kappa \cdot \sqrt{n_\theta}$, with $\kappa \in [0.01, 0.1]$, is a reasonable first choice for mini-batch optimization of ODE models. For particularly large models, small values of κ were more successful. Furthermore, decreasing learning rate schemes tended to be more helpful for optimizer convergence than constant learning rates. However, at the moment, it seems most helpful to test two or three different learning rate schedules.
- *Mini-batch size:* Start using very small mini-batch sizes first, and then test the effect of increasing the mini-batch sizes. From our experience, this will lead to finding an appropriate mini-batch size faster than starting with large mini-batch sizes.
- *Number of epochs:* A recent study has come to the conclusion that the overall number of optimization steps is roughly conserved for full-batch optimizers (24): For a large group of models, it was typically in the range of a few hundred to at most a one or two thousand optimization steps. Since mini-batch optimization cannot be expected to converge in fewer steps, but should do so in fewer epochs, choose the number of epochs accordingly, e.g., in the order of a few dozen epochs at most, depending on the number of optimization steps per epoch.

I believe that the authors should show the results of a sensitivity analysis on the parameters of the ODE model of cancer signaling, to highlight the importance of the obtained results and to better explain the results of the parameter estimation.

We added a part about uncertainty analysis and the singular value decomposition of an ensemble-based sensitivity analysis of the trained application model. Discussing the sensitivities of the 4232 parameter in detail may lead to interesting biological insights, but is rather problematic due to the size of the model. We think that this might shift the focus away from the method we want to present. However, we included an additional validation experiment, which is easier to interpret biologically and has also led to important insights: In-silico knockouts of the trained model were compared to a large gene knockout screen from recent literature. In the revised manuscript, biological conclusions from our results about the model and the importance of certain model modules, such as the cell cycle, are presented. A detailed discussion of the biology is presented in the supplementary information. Moreover, we perform a complete iteration of the classic modelling cycle: model training, model validation, model refinement, model re-training, and model re-validation. From our experience, these steps are commonly done for smaller ODE models, but have, as far as we know, never been carried out for large-scale ODE models due to the high computational cost. However, mini-batch optimization allowed us, for the first time, as far as we know, to indeed carry out all of these steps.

Changes in the manuscript:

Added text parts on model validation:

In a next step, we validated predictions of the model on cell lines from the independent test set, where we still found a Pearson correlation between the data and the simulation of ensemble model of 0.74 (Fig 5a). The ROC analysis for the classification into responding and non-responding treatments on the test set yielded an AUC value of 0.94 for the ensemble model values between 0.90 and 0.92 for the ten best local optimizations (Fig 5b), while the classification accuracy was around 85% (Fig 5c). Importantly, the model did not only classify trivial cases correctly, but was also able to capture the variability between cell lines and drugs (Fig 5d).

Mini-batch optimization renders iterative model refinement possible for large-scale ODE models

Beyond drug response predictions, we also validated the model on experimental data from a recently published CRISPR screen: Behan et al. had analyzed the change in cell viability when knocking out various genes on a large set of cell lines (5). We found 107 genes and 18 cell lines in the knockout dataset, which were also included in our application example and the dataset used for model training, respectively, summing up to 1926 data points which we used for validation. Based on the gene knockout data, we classified a gene as "essential" for a cell line, if the knockout led to a viability reduction of more than 50% and compared this essentiality classification with in-silico knockouts from our application example (Fig 5e). We computed receiver-operator-characteristics for knockout simulations based on our trained model, and on random parameters as reference. We obtained better-than-random predictions (AUC of 0.59) from the untrained model and, as expected, better predictions for the trained model (AUC of 0.63).

Yet, the classification threshold for model simulation chosen by the ROC analysis was surprisingly high, leading to many false positive predictions (Supplementary Fig. 10). As this indicated a shortcoming of the model, we added further proteins, which should contribute to the viability readout of the model (see Methods section for more details on how the viability readout is computed): First, we included the protein Checkpoint kinase 1, a major regulator of the cell cycle that is encoded by the gene CHEK1, as additional pro-proliferative readout. The gene CHEK1 was shown to be among the most essential ones for cell viability in the validation data, which was however not accounted for in the original model of (17). Secondly, we also included doubly phosphorylated MAP3K1 as pro-proliferative readout, which was so far only acting in a feedback loop with upstream proteins. This modification allowed the model to partly circumvent the downstream bottleneck of the MAPK-cascade, and hence made it possible to explain potential downstream effects of MAP3K1. After calibrating this refined model on the previously used drug response data, we obtained substantially improved predictions of gene essentiality (Fig 5e, AUC of 0.67). The classification threshold for the model readout was reduced, leading to substantially fewer false positive predictions (Fig 5f, Supplementary Fig. 11). A detailed analysis of the remaining false predictions is given in the Supplementary Information.

As confirmation of the refined model topology, we refitted the original model on the drug response and gene knockout data, i.e. 14,926 data points, simultaneously. As expected, calibrating the model towards the knockout data led to substantially improved classification results (AUC of 0.74, vs. 0.63 for the model trained on drug response data only). However, also when being refitted on the knockout data, the model was still not able to describe the data when knocking out CHEK1, confirming the limitations of the original viability readout (Supplementary Fig. 10). In a last step, we refitted also the improved model on drug response and knockout data simultaneously, which again drastically improved the classification accuracy (AUC of 0.84, vs. 0.67 for the model trained on drug response data only). This suggested that a more general ODE model of cell signaling should encode at least a basic version of the cell cycle, if the model is to explain viability. To the best of our knowledge, this is the first time that a model which includes a cell cycle (Supplementary Fig. 10) is able to predict

Changes in the manuscript:

Added text parts on uncertainty and sensitivity analysis:

Mini-batch optimization improves parameter space exploration and uncertainty analysis

In a last study, we further analyzed the optimization result with the smallest batch-size, i.e. 10. For this purpose, we created a large parameter ensemble from the optimization history based on a cutoff threshold, similar to the ideas of (76) (more details on the ensemble generation can be found in the Methods section). This ensemble contained 8,450 parameter vectors from 52 out of the 100 local optimization runs. We also generated a second, smaller ensemble using only the final results of the ten best optimization runs (Fig. 7a). In addition, we investigated the coverage of the feasible interval across the different local optimization runs for each model parameter at three different checkpoints during optimization (Fig. 7a). Comparing these parameter interval coverages from mini-batch optimization with those from full-batch optimization revealed that mini-batch optimization allowed to substantially increase the coverage and thus yields a better exploration of the parameter space (Fig. 7b).

To assess the uncertainty of the calibrated model parameters, we analyzed the parametric sensitivities of the model output on the ten best optimization results. A singular value decomposition of this sensitivity matrix showed that 3,029 directions in parameter space were numerically non-zero, indicating that the model could acquire information about more than 3,000 degrees of freedom. Yet, many directions in parameter space were poorly determined (Fig. 7c). Complementary to this sensitivity analysis, we analyzed the covariance structure of the large parameter ensemble. A principal component analysis (PCA) revealed that most of the total variance was spread across a low-dimensional subspace, showing a sharp drop in the explained variance after about 50 PCA directions, which coincides well with the number of local optimizations present in the parameter ensemble (Fig. 7d). This indicates that the parameter space was not yet sufficiently explored, although mini-batch optimization allowed to afford more local optimization runs than full-batch optimization. This was confirmed by a UMAP embedding of the parameter ensemble, which showed that the local optimization runs did not converge to a common optimum, but remained separated in parameter space (Supplementary Fig. 16). The spectrum of the covariance matrix indicated 49 directions with high variability, a second mode containing about 3,600 directions with lower, but clearly present variability (Fig. 7e), and a third mode of probably non-explored directions, which had a variability below floating point precision. The overall 3,600 explored PCA directions in this large ensemble are in line with the results from the sensitivity analysis (roughly 3,000), as the discrepancy between the two results can be explained with additional information being present in the larger ensemble.

Changes in the manuscript:

Added text parts model on gene knockouts in the Discussion section:

Furthermore, we performed in-silico gene knockouts and validated them on a large experimental CRISPR screen from literature, which helped us to improve the model topology and to obtain well-matching knockout predictions. In a last step, we performed an uncertainty analysis based on ensemble methods and assessed the exploration of the parameter space during optimization, where mini-batch optimization enabled results which had not been achievable with established optimization methods.

A closer analysis of the in-silico knockout study allowed us to pinpoint deficiencies of the large-scale cancer signaling model, e.g., concerning the implementation of cell cycle genes, and a subsequent model refinement yielded substantially improved model predictions. To the best of our knowledge, this is the first time that a full modeling cycle, including model refinement and re-training, has been performed for an ODE model of this size, as the cost of model calibration renders iterative refinement usually prohibitive for large ODE models. Yet, a set of incorrect predictions remained: In particular, the essentiality of genes that encode master kinases such as PDPK1 or phosphatases such as PPP2CA was underestimated. At the same time, the importance of some genes encoding proteins involved in the AKT and RAS signaling pathways was overestimated, especially those which either interact with the EGF receptor, such as SRC, VAV2, or EGFR itself, or those which acts as transcription factors, such as CREB1, ELK1, or STAT3. Hence, we assume that a more comprehensive model, trained on a larger dataset, would be needed to, e.g., reliably discover new drug targets or to reject possible drug candidates before entering a clinical trial. The datasets to train such a more comprehensive, whole-cell signaling model have been made available in the last years, and mini-batch optimization renders the calibration of such a model computationally feasible.

To make the computational experiment reproducible, the authors should provide the parameter space limits used in their tests.

We now provide the parameter bounds in the Methods section of the revised version of the manuscript.

Changes in the manuscript:

Initial submission:

For global optimization of θ , we used multi-start local optimization, i.e., we randomly sampled many parameter vectors, from which we initialized local optimizations.

Revised submission:

For global optimization of θ , we restricted the feasible parameter space to a region $\Omega = [10^{-5}, 10^3]^{n_\theta}$, which was assumed to be biologically plausible. Parameters were transformed and optimized on logarithmic scale, i.e., in the box $\Omega' = [-5, 3]^{n_\theta}$. We used multi-start local optimization, i.e., we randomly sampled many parameter vectors, from which we initialized local optimizations.

Finally, a discussion about alternative methods to accelerate the parameter estimation task, such as those exploiting GPUs, should be included in the manuscript. The authors might consider this paper: <https://academic.oup.com/bib/article/18/5/870/2562773>

We added a comment about further computational improvements, such as GPU computing, in the Discussion section of the manuscript. However, for our large-scale application example, we think that GPU computing will not be easily applicable due to the high memory requirements of the model (more than 2GB per model simulation, due to using adjoint sensitivity analysis,

which needs to store the model trajectories). Moreover, for leveraging the highly parallelized computing with GPUs, it would probably be necessary to integrate the ODE for different experimental conditions in parallel. As far as we understand, this would necessitate the same timestepping for the different experimental conditions, i.e., initial value problems, which we think may be problematic. Hence, in order to exploit GPU computing, a complete refactoring of at least our implementation would be necessary. Yet, for the long-term future, such a reimplementaion might indeed be promising.

Changes in the manuscript:

Initial submission:

Hierarchical optimization as well as variance reduction should be combined with methods for early-stopping, to avoid over-fitting and to further reduce computation time (23; 45).

Revised submission:

Hierarchical optimization as well as variance reduction should be combined with methods for early-stopping, to avoid over-fitting and to further reduce computation time (23; 45). **Other improvements, which would not impact the parameter estimation procedure itself, but could further reduce computation time, might be based on GPU computing (49) or direct methods for steady-state computation (22).**

Minor comments:

- replace "classical" with "classic" throughout the manuscript
- I believe Figure 1 should be better explain, especially panel B. Moreover, Figure 1B is never referred to in the text
- lines 172, 276: replace "previous study" with "previous batch of tests"
- I suggest to replot the histogram of Fig 3B, since columns representing values close to zero are not visible
- line 211: replace "dimension" with "size"
- there is an inconsistency between Figure 4D E G H and the corresponding explanation in the text (AUC vs AUROC). Moreover, I suggest to add labels to Figure 4E

We addressed all of the raised points, except the replotting of Figure 3B: The bars are not visible because they are identically zero. As we think that discerning zero values from small values in a bar plot is generally hard, we instead added a text snippets in this subfigure, which indicate that no (more) failure was observed. We also extended the explanation of the concept of experimental conditions and mini-batching in the manuscript, referenced subfigure 1b and extended the caption of Figure 1.

References

- [1] M. Abadi, A. Agarwal, P. Barham, E. Brevdo, Z. Chen, C. Citro, G. S. Corrado, A. Davis, J. Dean, M. Devin, S. Ghemawat, I. Goodfellow, A. Harp, G. Irving, M. Isard, Y. Jia, R. Jozefowicz, L. Kaiser, M. Kudlur, J. Levenberg, D. Mané, R. Monga, S. Moore, D. Murray, C. Olah, M. Schuster, J. Shlens, B. Steiner, I. Sutskever, K. Talwar, P. Tucker, V. Vanhoucke, V. Vasudevan, F. Viégas, O. Vinyals, P. Warden, M. Wattenberg, M. Wicke, Y. Yu, and X. Zheng. TensorFlow: Large-scale machine learning on heterogeneous systems, 2015. Software available from tensorflow.org.
- [2] B. B. Aldridge, J. M. Burke, D. A. Lauffenburger, and P. K. Sorger. Physicochemical modelling of cell signalling pathways. *Nat.*

- [3] J. Bachmann, A. Raue, M. Schilling, M. E. Böhm, C. Kreutz, D. Kaschek, H. Busch, N. Gretz, W. D. Lehmann, J. Timmer, and U. Klingmüller. Division of labor by dual feedback regulators controls JAK2/STAT5 signaling over broad ligand range. *Mol. Syst. Biol.*, 7(1):516, 2011.
- [4] J. Barretina, G. Caponigro, N. Stransky, K. Venkatesan, A. A. Margolin, S. Kim, C. J. Wilson, J. Lehár, G. V. Kryukov, D. Sonkin, A. Reddy, M. Liu, L. Murray, M. F. Berger, J. E. Monahan, P. Morais, J. Meltzer, A. Korejwa, J. Jané-Valbuena, F. A. Mapa, J. Thibault, E. Bric-Furlong, P. Raman, A. Shipway, I. H. Engels, J. Cheng, G. K. Yu, J. Yu, P. Aspesi, Jr, M. de Silva, K. Jagtap, M. D. Jones, L. Wang, C. Hatton, E. Palessandolo, S. Gupta, S. Mahan, C. Sougnez, R. C. Onofrio, T. Liefeld, L. MacConaill, W. Winckler, M. Reich, N. Li, J. P. Mesirov, S. B. Gabriel, G. Getz, K. Ardlie, V. Chan, V. E. Myer, B. L. Weber, J. Porter, M. Warmuth, P. Finan, J. L. Harris, M. Meyerson, T. R. Golub, M. P. Morrissey, W. R. Sellers, R. Schlegel, and L. A. Garraway. The Cancer Cell Line Encyclopedia enables predictive modelling of anticancer drug sensitivity. *Nature*, 483(7391):603–607, 2012.
- [5] F. M. Behan, F. Iorio, G. Picco, E. Gonçalves, C. M. Beaver, G. Migliardi, R. Santos, Y. Rao, F. Sassi, M. Pinnelli. Prioritization of cancer therapeutic targets using CRISPR–Cas9 screens. *Nature*, 568(7753), 2019.
- [6] M. Bouhaddou, A. M. Barrette, A. D. Stern, R. J. Koch, M. S. DiStefano, E. A. Riesel, L. C. Santos, A. L. Tan, A. E. Mertz, M. R. Birtwistle. A mechanistic pan-cancer pathway model informed by multi-omics data interprets stochastic cell fate responses to drugs and mitogens. *PLoS Comput. Biol.*, 14(3), page e1005985, 2018.
- [7] S. Boyd and L. Vandenberghe. *Convex Optimisation*. Cambridge University Press, UK, 2004.
- [8] R. H. Byrd, R. B. Schnabel, and G. A. Shultz. Approximate solution of the trust region problem by minimization over two-dimensional subspaces. *Math. Program.*, 40(1):247–263, 1988.
- [9] Z. Chaker, S. Aïd, H. Berry, and M. Holzenberger. Suppression of igf-i signals in neural stem cells enhances neurogenesis and olfactory function during aging. *Aging Cell*, 14(5):847–856, 2015.
- [10] W. W. Chen, B. Schoeberl, P. J. Jasper, M. Niepel, U. B. Nielsen, D. A. Lauffenburger, and P. K. Sorger. Input–output behavior of ErbB signaling pathways as revealed by a mass action model trained against dynamic data. *Mol. Syst. Biol.*, 5(1):239, 2009.
- [11] C. Jin, P. Netrapalli, R. Ge, S. M. Kakade, M. I. Jordan. On Nonconvex Optimization for Machine Learning: Gradients, Stochasticity, and Saddle Points. *preprint*, arXiv:1902.04811v2, 2021.
- [12] J. C. Costello, L. M. Heiser, E. Georgii, M. Gonen, M. P. Menden, N. J. Wang, M. Bansal, M. Ammad-ud din, P. Hintsanen, S. A. Khan, J.-P. Mpindi, O. Kallioniemi, A. Honkela, T. Aittokallio, K. Wennerberg, N. D. Community, J. J. Collins, D. Gallahan, D. Singer, J. Saez-Rodriguez, S. Kaski, J. W. Gray, and G. Stolovitzky. A community effort to assess and improve drug sensitivity prediction algorithms. *Nat. Biotech.*, 32(12):1202–1212, 2014.
- [13] F. Crauste, J. Mafille, L. Boucinha, S. Djebali, O. Gandrillon, J. Marvel, and C. Arpin. Identification of nascent memory CD8 T cells and modeling of their ontogeny. *Cell Syst.*, 4(3):306–317, 2017.
- [14] A. Defazio, F. Bach, and S. Lacoste-Julien. Saga: A fast incremental gradient method with support for non-strongly convex composite objectives. In Z. Ghahramani, M. Welling, C. Cortes, N. D. Lawrence, and K. Q. Weinberger, editors, *Advances in Neural Information Processing Systems (NIPS)*, volume 27, pages 1646–1654, 2014.
- [15] F. Eduati, V. Doldàn-Martelli, B. Klinger, T. Cokelaer, A. Sieber, F. Kogera, M. Dorel, M. J. Garnett, N. Blüthgen, and J. Saez-Rodriguez. Drug resistance mechanisms in colorectal cancer dissected with cell type-specific dynamic logic models. *Cancer Res.*, 77(12):3364–3375, 2017.
- [16] F. Fröhlich, B. Kaltenbacher, F. J. Theis, and J. Hasenauer. Scalable parameter estimation for genome-scale biochemical reaction networks. *PLoS Comput. Biol.*, 13(1):e1005331, 2017.
- [17] F. Fröhlich, T. Kessler, D. Weindl, A. Shadrin, L. Schmiester, H. Hache, A. Muradyan, M. Schütte, J.-H. Lim, M. Heinig, F. J. Theis, H. Lehrach, C. Wierling, B. Lange, and J. Hasenauer. Efficient parameter estimation enables the prediction of drug response using a mechanistic pan-cancer pathway model. *Cell Syst.*, 7(6):567–579.e6, 2018.
- [18] F. Fröhlich, D. Weindl, Y. Schälte, D. Pathirana, A. Paszkowski, G. T. Lines, P. Stapor, and J. Hasenauer. AMICI: high-performance sensitivity analysis for large ordinary differential equation models *Bioinformatics*, page btab227, 2021.
- [19] K. A. Fujita, Y. Toyoshima, S. Uda, Y.-i. Ozaki, H. Kubota, and S. Kuroda. Decoupling of receptor and downstream signals in the akt pathway by its low-pass filter characteristics. *Sci. Signal.*, 3(132):ra56, 2010.
- [20] D. Goldfarb. A family of variable-metric methods derived by variational means. *Math. Comput.*, 24(109):23, 1970.
- [21] I. Goodfellow, Y. Bengio, and A. Courville. *Deep Learning*. MIT Press, 2016.
- [22] S. Gopalakrishnan, S. Dash, C. Maranas. K-FIT: An accelerated kinetic parameterization algorithm using steady-state fluxomic data. *Metab. Eng.*, 61:197, 2020.
- [23] M. Hardt, B. Recht, Y. Singer. Train faster, generalize better: Stability of stochastic gradient descent. *PMLR.*, 48:1225, 2016.Â
- [24] H. Hass, C. Loos, E. Raimúndez-Álvarez, J. Timmer, J. Hasenauer, and C. Kreutz. Benchmark problems for dynamic modeling of intracellular processes. *Bioinformatics*, page btz020, 2019.
- [25] H. Hass, K. Masson, S. Wohlgemuth, V. Paragas, J. E. Allen, M. Sevecka, E. Pace, J. Timmer, J. Stelling, G. MacBeath, B. Schoeberl, and A. Raue. Predicting ligand-dependent tumors from multi-dimensional signaling features. *npj Syst. Biol. Appl.*, 3(1):27, 2017.

- [26] T. Hastie, R. Tibshirani, and J. H. Friedman. The Elements of Statistical Learning: data mining, inference, and prediction. Springer Series in Statistics. Springer-Verlag, New York, 2 edition, 2005.
- [27] D. Henriques, A. F. Villaverde, M. Rocha, J. Saez-Rodriguez, and J. R. Banga. Data-driven reverse engineering of signaling pathways using ensembles of dynamic models. PLoS Comput. Biol., 13(2):e1005379, 2017.
- [28] A. C. Hindmarsh, P. N. Brown, K. E. Grant, S. L. Lee, R. Serban, D. E. Shumaker, and C. S. Woodward. SUNDIALS: Suite of Nonlinear and Differential/Algebraic Equation Solvers. ACM T. Math. Software., 31(3):363–396, 2005.
- [29] HSL. A collection of Fortran codes for large scale scientific computation. <http://www.hsl.rl.ac.uk/>.
- [30] A. Janowczyk and A. Madabhushi. Deep learning for digital pathology image analysis: A comprehensive tutorial with selected use cases. J. Pathol. Inf., 7(29), 2016.
- [31] E.-M. Kapfer, P. Stapor, and J. Hasenauer. Challenges in the calibration of large-scale ordinary differential equation models. to appear in IFAC-PapersOnLine, 2019.
- [32] B. N. Kholodenko, O. V. Demin, G. Moehren, and J. B. Hoek. Quantification of short term signaling by the epidermal growth factor receptor. J Biol Chem, 274(42):30169–30181, 1999.
- [33] D. P. Kingma and L. J. Ba. Adam: A method for stochastic optimization. In International Conference on Learning Representations (ICLR) 2015 - accepted papers. Ithaca, 2015.
- [34] H. Kitano. Computational systems biology. Nature, 420(6912):206–210, 2002.
- [35] E. Klipp, R. Herwig, A. Kowald, C. Wierling, and H. Lehrach. Systems biology in practice. Wiley-VCH, Weinheim, 2005.
- [36] A. Korkut, W. Wang, E. Demir, B. A. Aksoy, X. Jing, E. J. Molinelli, Ö. Babur, D. L. Bemis, S. O. Sumer, D. B. Solit, et al. Perturbation biology nominates upstream–downstream drug combinations in raf inhibitor resistant melanoma cells. Elife, 4:e04640, 2015.
- [37] C. Kreutz. Guidelines for benchmarking of optimization-based approaches for fitting mathematical models. Genome Biology, 20(1):281, 2019.
- [38] Y. LeCun, L. Bottou, Y. Bengio, and P. Haffner. Gradient based learning applied to document recognition. P. IEEE, 86(11):2278–2323, 1998.
- [39] Y. LeCun, L. Bottou, G. B. Orr, and K.-R. Müller. Neural Networks: Tricks of the Trade, volume 1524 of Lecture Notes in Computer Science, chapter Efficient BackProp, pages 9–50. Springer Berlin Heidelberg, 2002.
- [40] L. Lei and M. I. Jordan. On the adaptivity of stochastic gradient-based optimization. preprint, arXiv:1904.04480v2 [math.OC], 2019.
- [41] J. Li, W. Zhao, R. Akbani, W. Liu, Z. Ju, S. Ling, C. P. Vellano, P. Roebuck, Q. Yu, A. K. Eterovic, L. A. Byers, M. A. Davies, W. Deng, Y. N. V. Gopal, G. Chen, E. M. von Euw, D. Slamon, D. Conklin, J. V. Heymach, A. F. Gazdar, J. D. Minna, J. N. Myers, Y. Lu, G. B. Mills, and H. Liang. Characterization of human cancer cell lines by reverse-phase protein arrays. Cancer Cell, 31(2):225–239, 2017.
- [42] A. C. Lloyd. The regulation of cell size. Cell, 154(6):1194–1205, 2013.
- [43] C. Loos, S. Krause, and J. Hasenauer. Hierarchical optimization for the efficient parametrization of ODE models. Bioinformatics, 34(24):4266–4273, 2018.
- [44] P. Lucarelli, M. Schilling, C. Kreutz, A. Vlasov, M. E. Boehm, N. Iwamoto, B. Steiert, S. Lattermann, M. Wösch, M. Stepath, M. S. Matter, M. Heikenwälder, K. Hoffmann, D. Deharde, G. Damm, D. Seehofer, M. Muciek, W. D. Gretz, Norbert Lehmann, J. Timmer, and U. Klingmüller. Resolving the combinatorial complexity of smad protein complex formation and its link to gene expression. Cell Systems, 6(1):75–89, 2018.
- [45] M. Mahsereci, L. Balles, C. Lassner, and P. Hennig. Early stopping without a validation set. preprint, arXiv:1703.09580 [cs.LG], 2017.
- [46] J. Martens. Deep learning via hessian-free optimization. In Proceedings of the 27th International Conference on Machine Learning, pages 735–742, 2010.
- [47] P. Mendes, S. Hoops, S. Sahle, R. Gauges, J. Dada, and U. Kummer. Computational Modeling of Biochemical Networks Using COPASI, chapter 2. Part of the Methods in Molecular Biology. Humana Press, 2009.
- [48] U. Münzner, E. Klipp, and M. Krantz. A comprehensive, mechanistically detailed, and executable model of the cell division cycle in *saccharomyces cerevisiae*. Nat. Commun., 10, 2019.
- [49] M. Nobile, P. Cazzaniga, A. Tangherloni, and D. Besozzi. Graphics processing units in bioinformatics, computational biology and systems biology. Brief. in Bioinf., 18(5):870, 2017.
- [50] J. Nocedal. Updating quasi-newton matrices with limited storage. Math. Comput., 35(151):773–782, 1980.
- [51] J. Nocedal and S. Wright. Numerical Optimization. Springer Science & Business Media, 2006.
- [52] C. Oguz, T. Laomettachtit, K. C. Chen, L. T. Watson, W. T. Baumann, and J. J. Tyson. Optimization and model reduction in the high dimensional parameter space of a budding yeast cell cycle model. BMC Syst. Biol., 7(53), 2013.

- [53] D. R. Penas, P. González, J. A. Egea, J. R. Banga, and R. Doallo. Parallel metaheuristics in computational biology: An asynchronous cooperative enhanced scatter search method. *Procedia Comput. Sci.*, 51:630–639, 2015.
- [54] R.-E. Plessix. A review of the adjoint-state method for computing the gradient of a functional with geophysical applications. *Geophys. J. Int.*, 167(2), 2006.
- [55] B. T. Polyak. Some methods of speeding up the convergence of iteration methods. *USSR Comp. Math. Math. Phys.*, 4(5):1–17, 1964.
- [56] A. Raue, M. Schilling, J. Bachmann, A. Matteson, M. Schelke, D. Kaschek, S. Hug, C. Kreutz, B. D. Harms, F. J. Theis, U. Klingmüller, and J. Timmer. Lessons learned from quantitative dynamical modeling in systems biology. *PLoS ONE*, 8(9):e74335, 2013.
- [57] H. Robbins and S. Monroe. A stochastic approximation method. *Ann. Math. Stat.*, 22(3):400–407, 1951.
- [58] D. A. Roberts. SGD Implicitly Regularizes Generalization Error. *preprint*, arXiv:2104.04874, 2021.
- [59] S. Ruder. An overview of gradient descent optimisation algorithms. arXiv:1609.04747.
- [60] D. E. Rumelhart, G. E. Hinton, and R. J. Williams. Learning representations by back-propagating errors. *Nature*, 323(6088):533–536, 1986.
- [61] Y. Schälte, P. Stapor, and J. Hasenauer. Evaluation of derivative-free optimizers for parameter estimation in systems biology. *IFAC-PapersOnLine*, 51(19):98–101, 2018.
- [62] R. M. Schmidt, F. Schneider, P. Hennig. Descending through a Crowded Valley – Benchmarking Deep Learning Optimizers. *preprint*, arXiv:2007.01547, 2020.
- [63] M. Schmidt, N. Le Roux, and F. Bach. Minimizing finite sums with the stochastic average gradient. *Math. Program. Ser. A*, 162:83–112, 2017.
- [64] L. Schmiester, Y. Schälte, F. Fröhlich, J. Hasenauer, and D. Weindl. Efficient parameterization of large-scale dynamic models based on relative measurements. *Bioinformatics*, btz581, 2019.
- [65] B. Sengupta, K. J. Friston, and W. D. Penny. Efficient gradient computation for dynamical models. *NeuroImage*, 98:521, 2014.
- [66] R. Serban and A. C. Hindmarsh. CVODES: The sensitivity-enabled ODE solver in SUNDIALS. In *ASME 2005 International Design Engineering Technical Conferences and Computers and Information in Engineering Conference*, pages 257–269. ASME, 2005.
- [67] S. L. Spencer and P. K. Sorger. Measuring and modeling apoptosis in single cells. *Cell*, 144(6):926–939, 2011.
- [68] P. Stäedter, Y. Schälte, L. Schmiester, J. Hasenauer, P. L. Stapor. Benchmarking of numerical integration methods for ODE models of biological systems. *Sci. Rep.*, 11(1):2696, 2021.
- [69] I. Sutskever. *Training recurrent neural networks*. PhD thesis, University of Toronto, Department of Computer Science, 2013.
- [70] I. Sutskever, J. Martens, G. Dahl, and G. Hinton. On the importance of initialization and momentum in deep learning. In *Proc. Int. Conf. Machine Learning*, pages 1139–1147, 2013.
- [71] P. Stapor, L. Schmiester, C. Wierling, B. Lange, D. Weindl, and J. Hasenauer. Supplementary material to Mini-batch optimization enables training of ODE models on large-scale datasets. Zenodo, 2019. <https://doi.org/10.5281/zenodo.3556429>.
- [72] I. Swameye, T. G. Müller, J. Timmer, O. Sandra, and U. Klingmüller. Identification of nucleocytoplasmic cycling as a remote sensor in cellular signaling by databased modeling. *Proc. Natl. Acad. Sci. USA*, 100(3):1028–1033, 2003.
- [73] TCGA Network. Comprehensive molecular portraits of human breast tumours. *Nature*, 490(7418):61–70, 2012.
- [74] T. Tieleman and G. Hinton. Lecture 6.5 – rmsprop: Divide the gradient by a running average of its recent magnitude. COURSERA: Neural Networks for Machine Learning, 2012.
- [75] A. F. Villaverde, F. Froehlich, D. Weindl, J. Hasenauer, and J. R. Banga. Benchmarking optimization methods for parameter estimation in large kinetic models. *Bioinformatics*, page bty736, 2018.
- [76] A. F. Villaverde, E. Raimúndez-Álvarez, J. Hasenauer, and J. R. Banga. A comparison of methods for quantifying prediction uncertainty in systems biology. *to appear in IFAC-PapersOnLine*, 2019.
- [77] A. Wächter and L. T. Biegler. On the implementation of an interior-point filter line-search algorithm for large-scale nonlinear programming. *Math. Program.*, 106(1):25–57, 2006.
- [78] P. Weber, J. Hasenauer, F. Allgöwer, and N. Radde. Parameter estimation and identifiability of biological networks using relative data. In S. Bittanti, A. Cenedese, and S. Zampieri, editors, *Proc. of the 18th IFAC World Congress*, volume 18, pages 11648–11653, Milano, Italy, 2011.
- [79] D. R. Wilson and T. R. Martinez. The general inefficiency of batch training for gradient descent learning. *Neural Networks*, 16:1429–1451, 2003.
- [80] B. Yuan, C. Shen, A. Luna, A. Korkut, D. S. Marks, J. Ingraham, C. Sander. CellBox: Interpretable Machine Learning for Perturbation Biology with Application to the Design of Cancer Combination Therapy. *Cell Systems.*, 12:128, 2021.

- [81] Y. Zheng, S. M. M. Sweet, R. Popovic, E. Martinez-Garcia, J. D. Tipton, P. M. Thomas, J. D. Licht, and N. L. Kelleher. Total kinetic analysis reveals how combinatorial methylation patterns are established on lysines 27 and 36 of histone H3. P. Natl. Acad. Sci. USA, 109(34):13549–13554, 2012.
- [82] S. Mandt, M. D. Hoffman, and D. M. Blei. Stochastic Gradient Descent as Approximate Bayesian Inference. preprint, arXiv:1704.04289v2, 2018.
- [83] D. Seita, X. Pan, H. Chen, and J. Canny. An Efficient Minibatch Acceptance Test for Metropolis-Hastings. In Proc. 27th IJCAI, 2018.
- [84] SIAM – Society for Industrial and Applied Mathematics James H. Wilkinson Prize for Numerical Software <https://www.siam.org/prizes-recognition/major-prizes-lectures/detail/james-h-wilkinson-prize-for-numerical-software>, on May 19, 2021
- [85] G. T. Lines, Å. Paszkowski, L. Schmiester, D. Weindl, P. Stapor, J. Hasenauer Efficient computation of steady states in large-scale ODE models of biochemical reaction networks. IFAC-PapersOnLine, 52(26):32–37, 2019.

REVIEWERS' COMMENTS

Reviewer #1 (Remarks to the Author):

The manuscript has been significantly improved. I'm impressed by the effort made to address concerns. My concerns have been adequately addressed.

Reviewer #3 (Remarks to the Author):

I am satisfied with the improvements introduced in the manuscript by the authors.
I believe the authors successfully addressed all points raised in the previous round of review.

Response to Reviewer 1

The manuscript has been significantly improved. I'm impressed by the effort made to address concerns. My concerns have been adequately addressed.

We thank this reviewer for this positive feedback.

Response to Reviewer 3

I am satisfied with the improvements introduced in the manuscript by the authors. I believe the authors successfully addressed all points raised in the previous round of review.

We thank this reviewer for this positive feedback.